

# CO$_2$ partial pressure and CO$_2$ emissions from the lower Red River (Vietnam)

Thi Phuong Quynh Le[1*], Cyril Marchand[2,3], Cuong Tu Ho[4], Thi Thuy Duong[4], Huong Thi Mai Nguyen[1], Lu XiXi[5], Duy An Vu[1], Phuong Kieu Doan[1] and Nhu Da Le[1]

[1]: Institute of Natural Product Chemistry, Vietnam Academy of Science and Technology, 18 Hoang Quoc Viet Road, Cau Giay, Hanoi, Vietnam.
[2]: IMPMC, Institut de Recherche pour le Développement (IRD), UPMC, CNRS, MNHN, Noumea, New Caledonia, France.
[3]: Faculty of Chemistry, University of Science – VNUHCM, 225 Nguyen Van Cu, Ho Chi Minh City, Vietnam
[4]: Institute of Environmental Technology, Vietnam Academy of Science and Technology, 18 Hoang Quoc Viet Road, Cau Giay, Hanoi, Vietnam.
[5]: Department of Geography, National University of Singapore, Arts Link 1, Singapore 117570, Singapore.

*Correspondence to*: Thi Phuong Quynh Le (quynhltp@yahoo.com or quynhltp@gmail.com)

**Abstract.** The Red River (Vietnam) is a good example of a South-East Asian river system, strongly affected by climate and human activities. This study aims to quantify the spatial and seasonal variability of carbon dynamic and CO$_2$ outgassing at the water-air interface of the lower Red River system. The monitoring of water quality and CO$_2$ emission were carried out for 24h cyclings at the five stations during the dry and monsoon seasons. The riverine water $p$CO$_2$ was supersaturated with CO$_2$ in contrast to the atmospheric equilibrium (400 ppm), averaging about 1588.6 ± 884.6 ppm, thus resulting in a water–air CO$_2$ flux of 26.9 ± 18.4 mmol m$^{-2}$ day$^{-1}$. The CO$_2$ outgassing rate was characterized by significant spatial variations, highest at Hoa Binh station (Da River) due to the dam impoundment and the highest river flow. Surprisingly, CO$_2$ outgassing was higher in the day time (30.4 ± 21.2 mmol m$^{-2}$ day$^{-1}$) than in the night time (23.3 ± 15.4 mmol m$^{-2}$ day$^{-1}$), probably as a result of the combined effect of higher wind speed and water temperature in the day time. Seasonal differences were also observed, higher in the wet season (30.7 ± 23.1 mmol m$^{-2}$ day$^{-1}$) than in the dry season (23.0 ± 12.2 mmol m$^{-2}$ day$^{-1}$), due to higher river discharges and higher external inputs of organic matters from watersheds. Conversely during dry season, temperature was among the main factors influencing C dynamic, with higher $p$CO$_2$ and fluxes, probably as a result of increased metabolic rates.

Keywords: carbon, CO$_2$ outgassing, Red River, Vietnam

## 1 Introduction

Natural hydrological processes and biogeochemistry of many rivers in the world have suffered from the influences of climate change and human activities in their drainage basins. Riverine carbon fluxes and outgassing are important parts of the carbon exchange among terrestrial, oceanic and atmospheric environment. Rivers and streams not only transfer various forms of carbon (dissolved and particulate) to oceans, but also evade a significant amount of carbon to the atmosphere (Battin et al., 2009; Richey





et al., 2002). Due to $CO_2$ evasion, the flux of carbon that leaves the terrestrial biosphere through global
fluvial network was suggested to be twice larger than the amount that ultimately reaches the coastal
ocean (Bauer et al., 2013; Regnier et al., 2013). The existence of hotspot of $CO_2$ evasion represented 70
% of the emission for only 20 % of the land surface (Raymond et al., 2013). Previous estimates
proposed that inland water bodies transport, mineralize and bury 2.7 Pg C yr$^{-1}$, which is similar to the
terrestrial carbon sink for anthropogenic emissions of 2.8 Pg C yr$^{-1}$ (Tranvik et al., 2009). However,
more recent studies proposed a lower value of global evasion rate from inland waters, being about 2.1
Pg C yr$^{-1}$ or even less (Raymond et al, 2013; Lauerward et al., 2015).
Carbon fluxes and emissions from rivers are impacted by both natural (plate margin tectonics,
volcanic deposits, high elevations, steep slopes, and high intensive rainfall) and anthropogenic factors
(high population density, deforestation, reservoir impoundment, intensive agricultures, and
urbanization). However, there is a limited understanding of spatial and temporal dynamics of carbon
exchange between terrestrial, oceanic and atmospheric environments for the large Asian rivers. In
southeast Asia, the river water discharge and sediment loads have been altered dramatically over the
past decades as a result of reservoir impoundment, land use, population, and climate changes (Walling
and Fang, 2003, 2006; Lu, 2004). Solid sediment loads not only directly contribute to increase the
organic carbon content, but also affect chemical weathering and hence carbon consumption and
possible carbon emission. Therefore, studies of carbon emission from the large Asian rivers are crucial
to quantify geochemical cycles accurately in the context of global change. Some studies noted that data
concerning $CO_2$ evasion (or $pCO_2$) of Southeast Asian rivers is a high priority in order to precise the
global evasion rate from inland water (Raymond et al., 2013; Lauerward et al., 2015).
The Red River with a basin area of 156,450 km$^2$ is a typical East Asia river that is strongly
affected by climate and human activities (Fig. 1). Previous studies reported about the hydrology and
suspended sediment load associated to some elements loads (N, P, C) of the Red River (Dang et al.,
2010; Lu et al., 2015; Le et al., 2015). Recently, the transfer of organic carbon of the Red River to
ocean has been studied (Dang et al., 2010; Le et al., 2017). However, the knowledge of carbon
exchange at the air-water interface of the Red River is still limited.
This study aims:   i) to investigate spatial and temporal (wet and dry, and day-night) variations of $CO_2$
partial pressure ($pCO_2$) and $CO_2$ flux  at the water-air surface of the lower Red River, from the
upstream to the downstream part;      and ii) to identify some of the factors that may control $pCO_2$ and
$CO_2$ outgassing rates of the lower Red River. To our knowledge, the present study introduced the first
measurement and estimation of $CO_2$ evasion from the lower Red River.
**2 Methods**
**2.1 Study sites**

Five stations were studied in the lower Red River (Vietnam): Yen Bai, Hoa Binh, Vu Quang, Hanoi
and Ba Lat (Fig. 1). Yen Bai station was located at the outlet of the Thao river; Hoa Binh station (after
Son La and Hoa Binh reservoirs) at the outlet of the Da River; Vu Quang at the outlet of the Lo River;





Hanoi and Ba Lat in the main course of the Red River downstream. The Ba Lat station is located in the
Red River mouth, about 50 km from the sea.
The climate in the Red River basin, of tropical East Asia monsoon type, is controlled by the
North East monsoon in winter and South West monsoon in summer. It is, thus, characterized by two
distinct seasons: rainy and dry season. The rainy season lasts from May to October and cumulates 85 –
90 % of the total annual rainfall in the Red River catchment, whereas the dry season covers the period
from November to next April. The monsoon climate weather results in a hydrologic regime
characterized by large runoffs during the wet season and low runoffs during the dry season (see below
for the detail river discharge in 2014). A series of dams-reservoirs are impounded in both Chinese and
Vietnamese territories of the Red River upstream part (Le et al., 2017). In the Da River, two large
dams Hoa Binh and Son La were constructed in the river main course whereas in the Lo River, two
large dams Thac Ba and Tuyen Quang were constructed in its tributaries.
The upstream part of the Red River in the Chinese part is dominated by mountain areas, which
are tectonically active and unstable, and this, combined with intense rainfall, causes high erosion
(Fullen et al., 1998) whereas in the Vietnamese part, soils are mostly (70 %) grey and alluvial soils (Le
et al., 2017).
Land use was quite different in the three upstream river basins Thao, Da and Lo: Industrial
crops dominate (58 %) in the Lo basin, forests and bare land (74 %) in the Da basin, and paddy rice
fields (66 %) in the delta area. The Thao basin is characterized by a larger diversity of land use
including forest, paddy rice fields, and industrial crops (85 %) (Le et al., 2015).
Population density varied from the upstream to the downstream part of the Red River basin.
The delta area, where the Hanoi and Ba Lat stations are located, is characterized by high population
density (> 1,000 inhabitant $km^2$). In the upstream part, where the Yen Bai station (in Thao River), Hoa
Binh station (in Da River) and Vu Quang station (in Lo River) situate, population density was much
lower, about 100 inhabitants $km^{-2}$ (Le et al., 2015).
**2.2 Sampling procedures and analysis**
Sampling campaigns were conducted in September (rainy season) and November (dry season) 2014 at
the five gauging stations.
Physico-chemical parameters were automatically recorded at the interval of 1 min during 24h
for each sampling campaign: pH, turbidity, salinity, chlorophyll *a* by the sensor YSI6920 (YSI, USA);
temperature and dissolved oxygen (DO) by a HOBO sensor (USA). These sensors have been calibrated
with standard solutions before each measurement campaign. All solutions used for calibration should
be at ambient temperature to ensure quality of calibration and all data must be entered on the
documents.
In parallel of in-situ measurement, river water samples were hourly collected for analysis of
other water quality variables (TSS, DOC, POC, and alkalinity) during 24h. A known volume of well-
mixed sample was filtered immediately by vacuum filtration through pre-combusted (at 450 $^\circ$C for 6 h)
glass fiber filters (Whatman GF/F, 47 mm diameter). The filters were then kept in a freezer (-20 $^\circ$C)
until analysis of TSS and POC. For the measurement of TSS, each filter was dried for 1h at 105 °C and



then weighed. Taking into account the filtered volume, the increase in weight of the filter represented
the total TSS per unit volume (mg L$^{-1}$).

POC concentrations were estimated on the same filters. Filters were then weighed before and
after calcination at 550 $^{\circ}$C for 4 hours. The difference in weight before and after calcination was
multiplied by 0.4 to provide an estimation of the POC content (Servais et al., 1995).

A volume of 30 ml sub-sample of filtrate was acidified with 35 µl 85 % $H_3PO_4$ acid and then
stored at 4 °C in amber glass bottles until measurement of the DOC concentrations using a TOC-V$_E$
(Shimadzu, Japan). The samples, standards and blank measurements were measured in triplicate.
Total alkalinity was hourly determined on non-filtered water samples in situ by titration method
(APHA, 1995).

Wind speed and air temperature were measured hourly during 24 h of all sampling campaigns
by a handle digital thermometer/anemometer (GM 8901, Total meter, Taiwan).
**2.3 Hydrological data collection**
Daily and hourly data of river water discharges in 2014 at the 5 hydrological stations studied were
collected from the Vietnam Ministry of Natural Resources and Environment (MONRE, 2016). The
daily data were collected for all days in 2014, whereas hourly data were obtained for the exact dates of
field measurements at the 5 sites (Table 1). The mean annual river flows in 2014 of the Thao, Da, Lo
Rivers and in the main axe of the Red River at the Hanoi and Ba Lat stations were: 527 ± 515; 1369 ±
833; 1302 ± 517; 1867 ± 1089; 615 ± 293 m$^3$ s$^{-1}$, respectively. Higher values of river discharges were
observed in wet season (May to October) than in dry season (January-April; November-December) at
all sites (Table 1).
**2.4 CO$_2$ fluxes determination**
CO$_2$ fluxes were determined using direct methods:     i) from $p$CO$_2$ in the water column measured using
an equilibrator connected to an IRGA     ii) from $p$CO$_2$ determined by the calculation using T$_{alk}$ and pH
measured in-situ.

**2.4.1 From measured $p$CO$_2$**
Equilibrator was used to determine the $p$CO$_2$ in water balanced with the air. The equilibrator was
designed, as described in Frankignoulle et al. (2001), as follow: a vertical plastic tube (height: 73 cm,
diameter: 9 cm) which is filled up with about 250 glass marbles (diameter = 1.5 cm) in order to
increase the surface exchange between water and air. The river water (water inlet) through a submerged
pump at 20 cm below the river surface water comes into the equilibrator from the top of the tube. The
water inlet can be regulated by a flow controller installed under the tygon tubing, which joins the water
inlet with the pump. A closed air circuit ensures circulation through the equilibrator (from the bottom
to the top), a water trap, a particle filter, a regulator and an IRGA (Licor 820, Licor®, USA), which was
calibrated before each sampling campaign using a series of standards concentrations of 0, 551 and 2756
ppm CO$_2$ (Air Liquide®). The IRGA was connected to a computer interface, which allows recording
the $p$CO$_2$ every second. Values were recorded during 24 h continuously.



The water-air $CO_2$ fluxes from the equilibrator measurement at each site were calculated by

the formula proposed by Raymond and Cole (2011) as followings:

$F_{Equi} = k_{600} * \alpha * (pCO_2 \text{ water} – pCO_2 \text{ air})$      Eq. (1)

Where F was the $CO_2$ flux from water ($\mu mol\ m^{-2}\ s^{-1}$) and converted in $mmol\ m^{-2}\ d^{-1}$;

$k_{600,}$ was gas transfer velocity of $CO_2$ or piston velocity ($cm\ h^{-1}$), and was calculated according to a

wind speed function (k vs. $U_{10}$). $K_{600}$ was calculated using the equation from Raymond and Cole

(2001), which developed from a data set of different estuaries and rivers as follow:

$k_{600} = 1.91 \times exp(0.35 \times U_{10}) \times (Sc\ /\ 600)^{-0.5}$    Eq. (2)

Where $U_{10}$ was the wind speed ($m\ s^{-1}$) at 10 m height above the surface water (was calculate

from the wind speed ($m\ s^{-1}$) at 2 m height, ("Appendix 8.6-Wind Speed Calculations" 2010); and Sc

was the Schmidt number, normalized to Schmidt number of 600 (Cole and Caraco, 1998).

$\alpha$ was the solubility coefficient of $CO_2$ for given temperature and salinity (Weiss, 1974) (mol

$L^{-1}\ atm^{-1}$). In this case, $\alpha = 0.034\ mol\ L^{-1}\ atm^{-1}$

$pCO_2$ water was $CO_2$ concentration in surface river water from Equilibrator measurement or

from CO2_SYS calculation.

$pCO_2$ air was considered as a constant of 400 ppm

Since we measured wind speed on field at a height of 2 meters (Uz = 2) with a handle

anemometer, the formula of Amorocho and DeVries (1980) was used to estimate the wind speed at 10

173        m height (U10).

The determination of the emissions depends on the gas exchange velocities ($k_{600}$), and the

latter may represent a considerable source of uncertainty when calculating water-air $CO_2$ outgassing

flux (Raymond et al., 2012; Raymond et al., 2013). There were some studies focused on the estimation

of $k_{600}$, based on empirical equations with different techniques (Jähne et al., 1984; Liss and Merlivat,

1986), for different type of ecosystems and specific weather conditions (Devol et al., 1987; Marino and

Howarth, 1993; Clark et al., 1995; Carini et al., 1996; Guérin et al., 2007; Vachon et al., 2010). As a

result, different empirical equations have been proposed, which lead to different quantitative

estimations of $k_{600}$ coefficient. In our study, we decided to use the empirical equation of $k_{600}$-wind from

Raymond and Cole (2001) developed from a data set of different estuaries and rivers.

Some studies indicate that $k_{600}$ values are closely related to flow velocity and channel gradient

for rivers (Alin et al., 2011), less related to wind velocity as in lakes and oceans (Abril et al., 2000;

Abril and Borges, 2004; Abril et al., 2009) and that water current can strongly affect the magnitude of

$k_{600}$, especially in location where the wind speed is low but water current is high (Borges et al., 2003).

In our study, $k_{600}$ is determined by the Eq. (2) on which a wind speed function (k vs. $U_{10}$) strongly

affected that may lead to underestimate $k_{600.}$ The underestimation of $k_{600}$ may lead to lower $CO_2$ flux

out-gassing. Thus, we consider that the $CO_2$ outgassing fluxes in our study reflected likely a low

estimate.



### 2.4 From calculated $p\mathrm{CO_2}$

DIC content may be calculated from the sum of total dissolved inorganic carbon in water including $HCO_3^-$, $CO_3^{2-}$, $H_2CO_3$ and $CO_2$, or can be calculated from a combination of any two of the following measured parameters total alkalinity, pH, or partial pressure of $CO_2$ ($p\mathrm{CO_2}$) (Cai et al., 2008; Sun et al., 2010). In this study, DIC contents were calculated from the sum of including $HCO_3^-$, $CO_3^{2-}$, $H_2CO_3$ and $CO_2$ contents, which were given by the calculation from the $CO_2$-SYS Software (version 2.0) basing on the total alkalinity contents and pH values measured in-situ as described above (Sect. 2.3).

In order to compare with $p\mathrm{CO_2}$ in water, which was measured by the equilibrator, $p\mathrm{CO_2}$ in water at the 5 sites was also simultaneously calculated on the CO2-SYS EXCEL Macro Software basing on total alkalinity contents and pH values measured in-situ at the 5 sites as described above (Sect. 2.3).

We feel confident about our measurements of $p\mathrm{CO_2}$, because both methods yielded similar results.

### 2.5. Statistical analysis

To detect the correlation between environmental variables and $p\mathrm{CO_2}$ or $CO_2$ outgassing flux, statistical software R version 3.3.2 (R Core Team, 2016) was applied to calculate the Pearson correlation coefficients. Some environmental variables were evaluated by "cor" to compare the correlation and selected representative variables. PCA analysis was then used for identifying representative variables that could relate to the dynamic of $p\mathrm{CO_2}$ or $CO_2$ flux.

### 3. Results

#### 3.1. Physical and chemical variables of the lower Red River

Water temperature varied from 23.3 to 29.4 °C, and the mean value in rainy period (27.4 °C) was higher than the one in dry period (24.5 °C) at almost all stations, except at the Hoa Binh site where the water temperatures did not show seasonal variations, remaining around 26.3 - 26.5 °C. Among the five hydrological stations, the higher water temperatures were recorded at the Hanoi and Ba Lat stations, ranging from 28 to 29 °C in the wet period, whereas they were close to 23 °C during the dry period. Temperatures at the Yen Bai and Vu Quang stations were approximately 26 °C in the wet period and 24 °C in the dry period (Table 2, Fig. 2).

pH values were not hugely different among the two periods. pH was slightly basic, ranging from 7.7 to 8.2 with an average 8.1 for all the sites. The lowest pH values were observed at the Hoa Binh in both periods (< 8), whereas they ranged between 8.0 and 8.4 at the other sites. pH was slightly higher in dry season than in the wet season at all the sites (Table 2, Fig. 2).The percentage of dissolved oxygen (% DO) varied from 50.5 to 70.7 % with an average value of 64.3 %. There were not significant differences in % DO with season, 63.6 % during the rainy seasons and 64.9 during the dry




season (Table 2). The mean values were the highest for the Yen Bai station (70.1 %) in the wet period,
and 69.5 % for the Ba Lat station in the dry period. The lowest values were observed at the Hoa Binh
station in both periods (55.0 % in the wet period, 51.4% in the dry period) (Table 2, Fig. 2).

Salinity at the four upstream sites was under the detection limit both in the rainy and dry

seasons, but in the estuary downstream river at the Ba Lat station, values up to 8.75 were measured
during the dry season (Table 2).

Alkalinity concentrations ranged from 84.3±1.9 to 152.9±6.6 mg L$^{-1}$, with higher values in the

dry season than in the rainy season, except at Vu Quang station. Conductivity ranged from 0.2±0.0 to
6.6±3.4 mS cm$^{-1}$, but did not show a clear variation between the dry and rainy season (Table 2, Fig. 2).

Chlorophyll *a* was quite low during the two sampling campaigns, ranging from 0.23 to 2.77

µg L$^{-1}$, with an average of 1.61 µg L$^{-1}$. Higher values in the rainy season than in the dry season were
observed at almost sites. From Yen Bai to Ba Lat, Chl-a concentrations in the main axe (at Yen Bai and
Hanoi stations) were higher than in the two tributaries Da and Lo (Table 2, Fig. 2), even under the
higher values of turbidity.

The wind speed ranged from 0.2±0.3 to 3.1±1.5 m s$^{-1}$, and the highest value occurred in the

dry season at Ba Lat. The results showed a clear variation in wind speed between day and night, with
the higher values during the day time for all stations (Table 3).

**3.2. Carbon concentrations of the lower Red River**

During the two sampling campaigns, DOC concentration ranged from 0.5 to 4.6 mgC L$^{-1}$, averaging 1.5
mgC L$^{-1}$. Higher values were observed during the rainy season (2.0 mgC L$^{-1}$ vs.1.5 mgC L$^{-1}$ during the
dry season), and the highest value was recorded at Hanoi site (Table 2). POC concentrations varied
from 0.4 to 4.6 mgC L$^{-1}$. The mean value in the rainy season was 1.6 mgC L$^{-1}$, slightly higher than in
dry season (1.4 mgC L$^{-1}$). Among the 5 sites, POC concentrations in the main reach of the Red River
(Yen Bai, Hanoi and Ba Lat sites) were higher than in the two tributaries Da and Lo, where dams were
constructed (Table 2, Fig. 2).

DIC concentrations at the five sites fluctuated from 16.7 to 32.9 mgC L$^{-1}$, averaging 23.8 mgC

L$^{-1}$. Lower values were observed in the rainy season (22.3 mgC L$^{-1}$), lower than the values in the dry
season (25.3 mgC L$^{-1}$) (Table 2, Fig. 2).

**3.3. Comparisons of the *p*CO$_2$ results for the two methods**

*p*CO$_2$ along the lower Red River (Vietnam) in the dry and monsoon seasons were determined by two
methods: i) direct measurements using an equilibrator connected to an IRGA, ii) calculated from pH
and alkalinity using the CO$_2$-SYS® software. The direct measurements gave higher values than the
calculated (Table 2), but the values of two methods were well correlated (R$^2$ = 0.76, Fig. 3). This result





is opposite to what was observed in organic-rich freshwater (Abril et al., 2015) or in estuarine
mangrove waters (Leopold et al., 2017), where large overestimation of the calculated $pCO_2$ was
observed. The higher calculated values compared to direct measurements were related to a more
significant contribution of organic acids anions to total alkalinity in waters with low carbonate
alkalinity and high DOC concentrations, and to a lower buffering capacity of the carbonate system at
low pH (Abril et al., 2015). However along the Red River, pH values remained around 8 and DOC
concentrations were lower than 2 mg L$^{-1}$ (Table 2), which may explain for the lower calculated $pCO_2$
values (Table 2). Thus, in this study, we used the results of the measured $pCO_2$ for the discussion and
to determine the $CO_2$ emissions from the lower Red River to the atmosphere (Fig. 4).

**3.4. Daily variations of $pCO_2$ and $CO_2$ emissions**
The riverine water $pCO_2$ was supersaturated with $CO_2$ in contrast to the atmospheric equilibrium (400
ppm), averaging 1,588.6 ± 884.6 ppm. In general, the results did not show a clear variation in $pCO_2$
between the day and night at all stations, except higher values at the night time in the dry season at the
Ba Lat site (Table 2, Fig. 2).

$CO_2$ flux values in the day time were higher than in the night time for all the stations during
all sampling campaigns (Table 3, Fig. 4). Previous studies revealed that photosynthesis of
phytoplankton may have a strong influence on circadian variation of $CO_2$ outgassing, since this process
consumes $CO_2$ during the day (Linn and Doran, 1984; Cole et al., 2000). However, in the present study,
very low Chl-a concentrations were measured, from 0.5 to 3.1 µg L$^{-1}$, probably as a result of the high
turbidity limiting light penetration in the water column. Thus, phytoplankton activity had a low
influence on C dynamic in the lower Red River system.

In the present study, water temperature was slightly higher in the day time than in the night
time (around 0.1 – 0.5 ºC) (Table 2), and positive correlations between temperature and $CO_2$ flux was
observed at Hoa Binh and Yen Bai station ($R^2$ = 0.72 and 0.84, respectively). Water temperature could
alter the riverine $pCO_2$ value. Previous studies indicated that $CO_2$ water solubility decreases with the
temperature increase during the day (Parkin and Kaspar, 2003; Roulet et al., 1997), leading to higher
$CO_2$ outgassing from river water interface. This effect was observed for some rivers in the world
(Roulet et al., 1997; Guasch et al., 1998; Dornblaser and Striegl, 2013). However for the Ba Lat site,
$CO_2$ fluxes were higher during the day but it was not always the case for $pCO_2$, suggesting that another
factors may be responsible for these results. The other parameter that differed during the night and day
was the wind speed which was higher during the day (Table 3). Previous studies suggested that the
large changes of $CO_2$ fluxes observed between the day and night can be caused not only by biological
cycle but also mainly by daily wind patterns (Hamilton et al., 1994). Small changes in wind pattern
may modify the direction and magnitude of the $CO_2$ flux during the day-night cycle (Sellers et al.,
1995). Our results were in agreement with some reports for rivers in tropical region (e.g. the Shark
River in USA (Ho et al., 2016)), which emphasised the influence of strong diurnal signal in wind speed
leading to higher k values in the day time than in night time (Rey et al., 2012; Ho et al., 2016).





Different explanations were given for the day-night variation of $pCO_2$ and $CO_2$ flux for
aquatic ecosystems in the world. Ho et al. (2016) found higher $CO_2$ flux in the Shark River (USA)
during the day than at night was due to the strong diurnal signal in u10. Similar results have also been
reported for the Yangtze River Delta (China) (Xu et al., 2017).  Roulet et al. (1997) reported that $CO_2$
flux in some ponds in Canada were always higher during the day because of higher water
temperature.  Higher $CO_2$ concentrations in day time were also observed for the two second-order
Mediterranean streams: Riera Major, a siliceous shaded stream and La Solana, a calcareous open
stream where changes in $CO_2$ concentration was greatly influenced by stream metabolism (Guasch et
al., 1998). However, average $CO_2$ outgassing during the night higher than during day by up to 1.8 times
was also observed for the Alpine streams, where temperature and hydrology were major drivers of
$pCO_2$ dynamics (Peter et al., 2014). $CO_2$ increased in the night time with the values of 6-46% higher
than the day time values during each 24-h period have been described by Sellers et al. (1995) for the
lake Northwest Ontario, Canada. Thus, it is possible that the contrast between day and night time
concentrations being a consequence of specific features of the measuring sites (Wang et al., 2010).
**3.5. Seasonal variations of $pCO_2$ and $CO_2$ emission**
$pCO_2$ values fluctuated from 694 ppm (at Yen Bai) in the dry season to 3,887 ppm (at Hoa Binh) in the
wet season. The mean values were highest for the Hoa Binh site in both seasons whereas the lowest one
was observed at the Yen Bai site.  Higher values of $pCO_2$ in the wet season than in the dry season were
observed at almost sites, except at Yen Bai (Table 2).
$CO_2$ flux values varied with seasons, and ranged from 4.4±2.0 mmol m$^{-2}$ d$^{-1}$ (at Vu Quang) in the dry
season to 48.4±23.4 mmol m$^{-2}$ d$^{-1}$ (at Hoa Binh) in the wet season. $CO_2$ flux values were higher in the
wet season for all stations (Table 3, Fig. 4). During the monsoon season, the river discharges were
about 2 to 3 times higher at all the sites (Table 1). As known in tropical regions, the wet season usually
experienced higher $pCO_2$ than the dry season because of the intense rainfall that induces higher OM
inputs into the river. This process was observed in some subtropical rivers: the Longchuan River (Li et
al., 2012) and the Xijiang River (Yao et al., 2007), with $pCO_2$ values increasing significantly when
baseflow and interflow increased, and flushed significant amount of carbon into the streams. In
addition, it was suggested a link between the seasonal variation in soil $CO_2$ content and riverine $CO_2$
outgassing due to porewater export (Hope et al., 2004; Melack et al., 2009; Rudorff et al., 2011). Thus
the higher $pCO_2$ values observed along the lower Red River during the rainy season may reflect the
influence of soil organic matter inputs to the riverine water column, evidenced by the higher values of
DOC and POC in the rainy seasons at all the sites.

In addition, seasonal variation of temperature may also explain the seasonal variability of
$pCO_2$ values of the Red River, being higher during the hot rainy season. Some authors suggested that
higher water temperatures in wet season in tropical regions were responsible for increased $pCO_2$ and
higher $CO_2$ emissions towards the atmosphere (Chanda et al., 2013; Takahashi et al., 2002). Dessert et
al., (2003) suggested that higher temperature should induce higher weathering rates, leading to higher





DIC export. This effect was not only observed in tropical rivers but also in mixed mangrove forest in
India (Chanda et al., 2013), or in the Southern Ocean (Nomura et al., 2014). Increase in temperature
decreases $CO_2$ solubility but increase OM decomposition processes, which produce $CO_2$. This may
partly explain the higher $pCO_2$ of the Red River during the hot rainy season. However, direct
relationship between temperature and $pCO_2$ during the rainy season was not evident, probably because
riverine inputs were the dominant factor driving $pCO_2$, while during the dry season $pCO_2$ clearly
increased with temperature, suggesting that metabolic rate controlled $pCO_2$ when adjacent soils inputs
are limited (Fig. 5).

## 4. Discussion

PCA and Pearson correlation coefficient were performed to analyze the relationships between nine
environmental variables and $pCO_2$ or $CO_2$ evasion flux at five sampling stations of the lower Red River
in the wet season (September 2014) and dry season (November 2014). The PCA of the seasonal data
for five sampling stations presented a clear separation between two periods: rainy and dry season (Fig.
6). The rainy period was characterized by high levels of temperature, $pCO_2$, $CO_2$ flux, POC, DOC and
Chl $a$. The Pearson correlation coefficient showed a low positive correlation between $pCO_2$ and DIC
and oxygen saturation (%) (r = 0.19) and between $CO_2$ flux and DOC (0.17). Both $pCO_2$ and $CO_2$ flux
were found to be negatively correlated with pH (r= -0.3) (Table 4).
Consequently, we suggest that the $pCO_2$ and $CO_2$ outgassing are the results of a combination
of multiple parameters, rather than a single one: season (including precipitation and temperature), dam
construction, population density, geomorphological characteristics of the catchment, etc.

### 4.1. Influence of dams on $pCO_2$ and $CO_2$ emission

Along the lower Red River, $pCO_2$ values were characterized by significant spatial variations (Table 2,
Fig. 2). In the upstream part, $pCO_2$ ranged from 964 to 3,830 ppm. Within the three tributaries, in
addition to the two large dams (Hoa Binh and Son La) in the Vietnamese part, a series of dams in the
Chinese territory were constructed in the 2000s for the Da River, whereas only some small
dams/reservoirs were built up in the Thao River in the Chinese territory. In the present study, the
highest $pCO_2$ values and the lower pH values were measured at the Hoa Binh site, which is situated
downstream a series of reservoirs. Previously, reservoirs were suggested to decrease river $pCO_2$ due to
increased residence times and autotrophic production (Wang et al., 2007). However, Lauerward et al.,
(2015) found a low negative correlation between them. Abril et al., (2005) noted that intense
mineralization of organic matter (OM) originating from the reservoir was possibly a significant source
for $pCO_2$ value in downstream river. Thus, the high $pCO_2$ measured at this site may reflect the
increased decomposition of OM and/or the water perturbation due to dam construction in this study.

### 4.2 Influence of water discharge and geomorphological characteristics on $pCO_2$ and $CO_2$ emission

$pCO_2$ differences between the three upstream tributaries and the main downstream axe of the Red River
are suggested to be partially related to different hydrological characteristics and management of the





three sub-basins and delta area, as observed in other systems (Yao et al., 2007; Li et al., 2012). Our
results showed that within the 3 upstream sites studied, the highest $pCO_2$ values were always measured
in the Da River at Hoa Binh site, where river discharges were the highest ($2,189 \pm 39$ m$^3$ s$^{-1}$ in wet
season and $868 \pm 319$ m$^3$ s$^{-1}$ in dry season), whereas the lowest $pCO_2$ were measured at the Yen Bai
station of the Thao River, where river discharges were the lowest ($840 \pm 68$ m$^3$ s$^{-1}$ in wet season and
$260 \pm 18$ m$^3$ s$^{-1}$ in dry season) (Table 1 and Table 2). Figure 7 showed the increased trend of both $pCO_2$
and $fCO_2$ when river discharge increased in both rainy and dry seasons for the lower Red River.
Differences in geomorphological characteristics between the upstream and downstream parts
of the lower Red River may be another reason responsible for the variability of $pCO_2$ observed among
the five stations. As presented above, the upstream part of the Red River is located in mountainous
areas, where chemical and mechanical erosion are among the world highest (500 mm per 1,000 years)
(Meybeck et al., 1989), whereas the downstream part is located in a very flat and low land, with an
elevation ranging from 0.4 to 12 m above sea level (Nguyen Ngoc Sinh et al., 1995). Regarding the Ba
Lat site, which is situated in the Red River estuary and thus in a very low and flat land, $pCO_2$ values
were lower than in Hanoi. It is interesting to observe that the river water discharge at Hanoi site ($3,296$
$\pm 86$ and $1,915 \pm 149$ m$^3$ s$^{-1}$) was about 3 times higher than the one at Ba Lat ($1,269 \pm 93$ and $453 \pm 31$
m$^3$ s$^{-1}$) in both wet and dry seasons respectively (Table 1), whereas higher $pCO_2$ values were measured
during the dry season in Hanoi than in Ba Lat (1,150 and 800 ppm, respectively), but during the rainy
season the values were close, i.e. around 1,450 ppm. We think that dilution by seawater may lead to a
reduction of riverine surface water $pCO_2$, especially in the dry season when the river flow was lower.
The higher salinity values measured at Ba Lat site in the dry season (3.6) than in the wet season (0.2)
may confirm our suggestion that tidal action influenced at Ba Lat site in the Red River estuary. This
result is consistent with previous observations in the Changjiang River estuary (Chen et al., 2008; Bai
et al., 2015) or in the Pearl River Estuary (Semiletov et al., 2004; Delille, 2006; Zemmelink et al.,

2006).


**4.3 Influence of population density on $pCO_2$ and $CO_2$ emission**

From the upstream to the downstream part of the main axe of the lower Red River, $pCO_2$ increased
from Yen Bai (mean value of 995 ppm) to Hanoi (mean value of 1,256 ppm) and then decreased to the
estuary at Ba Lat (mean value of 1,154 ppm). Higher values at Hanoi than at Yen Bai may be explained
by carbon inputs from the Hanoi wastewater system (Trinh et al., 2007). Indeed, high $CO_2$ efflux in the
peri-urban rivers of the Red River Delta that run through megacity Hanoi, e.g. the Day River (Trinh et
al., 2009) and the Nhue River (Trinh et al., 2012) was reported, whereas the upstream zone (Yen Bai) is
less subject to anthropogenic pressure. The higher values of both DOC and POC concentrations in
Hanoi site than in Yen Bai site may further indicate the influence of organic matter inputs at Hanoi site.

**4.4. Comparison with World Rivers**

$CO_2$ emissions from the Red River, as determined in the present study, were higher than the ones of
some Asian rivers such as the Lupar River ($13 \pm 3.0$ mmol m$^{-2}$ d$^{-1}$) and the Saribas River ($14.6 \pm 3.3$
mmol m$^{-2}$ d$^{-1}$) in Malaysia (Wit et al., 2015), and much higher than the results from some rivers in



Indonesia (Musi, Batanghari, Indragiri, Siak Rivers: 5±1.1, 1.8±0.4, 9.7±2.2, 8.3±1.9 mmol m$^{-2}$ d$^{-1}$,
respectively) (Wit et al., 2015) or in upper Yukon River (6 mmol m$^{-2}$ d$^{-1}$) (Striegl et al., 2007).
However, $CO_2$ outgassing from the lower Red River system was found much lower than the ones of
many rivers in America and Africa such as the Capibaribe river with 225 mmol m$^{-2}$ d$^{-1}$ (Moacyr et al.,
2013), the Amazon river 190 ± 55 mmol m$^{-2}$ d$^{-1}$ (Richey et al., 2002), the Mississipi River (270 mmol
m$^{-2}$ d$^{-1}$) (Dubois et al., 2010), some rivers in South America reported by Rasera et al. (2013) such as the
Negro, the Solimoes, the Caxiuana rivers (855±294, 518±17, 778±17 mmol m$^{-2}$ d$^{-1}$, respectively), or
even much lower than the ones of some large Asian rivers as the Yellow river (856±409 mmol m$^{-2}$ d$^{-1}$)
(Ran et al., 2015) and the Xijiang river (357 mmol m$^{-2}$ d$^{-1}$) (Yao et al., 2007) (Table 5). Our lower
values for the Red River were surprising when considering that large Asian rivers (in Himalayas and
Tibet Plateau regions) played important role in terms of chemical weathering (via carbon consumption
due to tectonic forcing) (Raymo and Ruddiman, 1992) and carbon burial (France-Lanord and Derry,
1997). The present low value of the Red River $CO_2$ outgassing may be related to the large decrease of
suspended solids, associated with strong decrease of particulate organic carbon due to a series of dam
construction in the upstream part of the Red River (Le et al., 2017) or due to low water flow in the
measured year.


**5. Conclusions**

This work presented the estimates of $CO_2$ emissions at the water-air interface at the 5 sites along the
lower Red River system in the dry and rainy seasons. The riverine water $pCO_2$ was supersaturated with
$CO_2$ in contrast to the atmospheric equilibrium (400 ppm), averaging about 1,588.6 ± 884.6 ppm, thus
resulting in a mean water–air $CO_2$ flux of 26.9 ± 18.4 mmol m$^{-2}$ day$^{-1}$ from the lower Red River system.
The $CO_2$ outgassing rate from the water surface Red River network was characterized by significant
spatial variations, being the highest at the Hoa Binh dam downstream and in the main axe at Hanoi
station. The highest value obtained at Hoa Binh site may reflect the important impact of a series of
large dams (Son La, Hoa Binh) in the Da river, but also the high water discharge, whereas the high
$pCO_2$ value in Hanoi may reflect the influence of population density, notably through the release of
elevated amount of organic carbon into the river. The monsoon season resulted in an increased amount
of OM inputs from adjacent soil, and combined to an increase of temperature, led to higher $pCO_2$
values. During the dry season, temperature appeared to be the main factor controlling $pCO_2$. Regarding
$CO_2$ evasion, differences appearred between the day and night in both dry and rainy seasons at almost
all sites with higher values found in the day time (30.4 ± 21.2 mmol m$^{-2}$ day$^{-1}$) than in the night time
(23.3 ± 15.4 mmol m$^{-2}$ day$^{-1}$). This result was related to the combined higher wind speed and higher
temperatures during the day. Consequently, this study evidenced that $CO_2$ dynamic along the lower
Red River was controlled by both anthropogenic activities (dam, urban effluents), and natural
meteorological-hydrological characteristics (rainfall- river discharge and temperature).
**Author contribution**

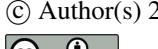


Le TPQ, Marchand C and Ho TC designed the experiments. Le TPQ, Ho TC and Vu DA carried the in-
situ experiments. Phuong KD and Le ND contribute to data treatment and calculations. Le TPQ and
Marchand C prepared the manuscript with the contributions from all co-authors.
**Acknowledgements**
This work was performed in the framework of the *ARCP2014-03CMY-Quynh/ARCP2013-06CMY-*
*Quynh/ARCP2012-11MY-Quynh* and the *Vietnam-NAFOSTED 105.09-2012.10* projects. The authors
would like to thank and the Asia-Pacific Network for Global Change Research (APN) and the
Vietnam's National Foundation for Science and Technology Development (NAFOSTED-Vietnam) for
their financial supports. We thank Ms Nguyen Bich Ngoc and Mr Nguyen Trung Kien for helping field
work.

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



Appendix 8.6 wind speed calculations. : Partnerships For Renewables, HMP Standford Hill Wind
Energy Development, p 1- 4, 2010.

Battin, T. J., Luyssaert, S., Kaplan, L. A., Aufdenkampe, A. K., Richter, A., and Tranvik, L. J.: The
boundless carbon cycle, Nat. Geosci., 2, 598-600, 2009.

Bauer, J. E., Cai, W.-J., Raymond, P. A., Bianchi, T. S, Hopkinson, C.S. and Regnier, P.A.G.: The
changing carbon cycle of the coastal ocean, Nature, 504(7478), 61–70, doi:10.1038/nature12857,
2013.

Borges, A.V., Djenidi, S., Lacroix, G., Theate, J., Delille B., and Frankignoulle, M.: Atmospheric $CO_2$
flux from mangrove surrounding waters, Geophys. Res. Lett., 30(11), 1558, 2003.

Bai, Y., Cai, W.-J., He, X., Zhai, W., Pan, D., Dai, M., and Yu, P.: A mechanistic semi-analytical
method for remotely sensing sea surface $pCO_2$ in river-dominated coastal oceans: A case study
from the East China Sea, J. Geophys. Res. Oceans, 120, 2331– 2349, doi:10.1002/2014JC010632,
2015.

Cai, W.J., Guo, X., Chen, C-T. A., Dai, M., Zhang, L., Zhai, W., Lohrenz, S., Yin, K., Harrison P. and
Wang, Y.: A comparative overview of weathering intensity and $HCO_3^-$ flux in the world's major
rivers with emphasis on the Changjiang, Huanghe, Zhujiang (Pearl) and Mississippi Rivers, Cont.
Shelf. Res., 28, 1538-1549, 2008.

Carini, S., Weston, N., Hopkinson, C., Tucker, J., Gilbin, A., and Vallino, J.: Gas exchange in the
Parker estuary, MA. Biol. Bull. 191: 333-334, 1996.

Clark, J. F., Shlosser, P., Simpson, H. J., Stute, R., Wanninkhof, R. and Ho, D.T.: Relationship between
gas transfer velocities and wind speeds in the tidal Hudson River determined by the dual tracer
technique, In Air-Water Gas Transfer, edited by B.Jahne and E.Monahan, pp. 175-800, Aeon
Verlag, Hanau, 1995.

Cole, J. J. and Caraco, N. F.: Atmospheric exchange of carbon dioxide in a low-wind oligotrophic lake
measured by the addition of SF6, Limnol. Oceanogr. 43, 647-656, 1998.

Cole, J. J., and Caraco, NF.: Carbon in catchments: connecting terrestrial carbon losses with aquatic
metabolism, J. Mar. Freshwater Res., 52(1), 101 – 110, 2001.

Cole, J. J., Pace, M. L., Carpenter, S. R., and Kitchell, J. F.: Persistence of net heterotrophy in lakes
during nutrient addition and food web manipulations, Limnol. Oceanogr., 45(8), 1718–1730.
DOI: 10.4319/lo.2000.45.8.1718, 2000.

Chanda, A., Akhand, A., Manna, S., Dutta, S., Hazra, S, Das, I., and Dadhwal, V. K.: Characterizing
spatial and seasonal variability of carbon dioxide and water vapour fluxes above a tropical mixed
mangrove forest canopy, India. J. Earth Syst. Sci., 122(2), 503–513, 2013.

Chen, C.T.A., Zhai, W.D., and Dai, M..: Riverine input and air–sea $CO_2$ exchanges near the
Changjiang (Yangtze River) Estuary: Status quo and implication on possible future changes in
metabolic status, Cont. Shelf Res., 28, 1476–1482, 2008.



Dang, T.H., Coynel, A., Orange, D., Blanc, G., Etcheber, H., and Le, L.A.: Long-term monitoring
(1960–2008) of the river-sediment transport in the Red River Watershed (Vietnam): Temporal
variability and dam-reservoir impact, Sci Total Environ., 408, 4654–4664, 2010.

Dessert, C., Dupré, B., Gaillardet, J., Francois, L. M., and Allegre, C.J.: Basalt weathering laws and the
impact of basalt weathering on the global carbon cycle. Chem. Geol., 202, 257–273, 2003.

Devol, A.H., Quay, P.E., Richey, J.E., and Martinelli, L.A.: The role of gas exchange in the inorganic
carbon, oxygen, and [222]Rn budgets of the Amazon River, Limnol Oceanogr, 32, 235-248, 1987.

Dornblaser, M. and Striegl, R.: Seasonal variation in diel carbon dynamics, Beaver Creek, Alaska,
AGU Fall Meeting Abstracts, p 15–27, 2013.

Dubois, K. D., Lee, D., and Veizer, J.: Isotopic constraints on alkalin- ity, dissolved organic carbon,
and atmospheric carbon dioxide fluxes in the Mississippi River, J. Geophys. Res., 115, G02018,
doi:10.1029/ 2009JG001102, 2010.

Delille, B.: Inorganic carbon dynamics and air-ice-sea $CO_2$ fluxes in the open and coastal waters of the
Southern Ocean, Ph. D. thesis, Univ. of Liége, Liége, Belgium, 1–297, 2006.

Frankignoulle, M., Borges, A., and Biondo, R.: A new design of equilibrator to monitor carbon dioxide
in highly dynamic and turbid environments, Wat. Res. 35(5), 1344–1347, 2001.

France-Lanord, C. and Derry, L.A.: Organic carbon burial forcing of the carbon cycle from Himalayan
erosion, Nature, 390(6), 65-67. DOI: 10.1038/36324, 1997.

Fullen, M. A., Mitchell, D. J., Barton, A. P., Hocking, T. J., Liguang, L., Zhi, W. B., Yi, Z., and Yuan,
X.Z.: Soil erosion and Conservation in the Headwaters of the Yangtze River, Yunnan Province,
China, In Headwaters: Water resources and Soil conservation, edited by Haigh MJ, Krecek J,
Rajwar S, Kilmartin MP, pp. 299–306, Balkema, Rotterdam/Oxford and IBH, New Delhi, 460pp,
1998.

Guasch, H., Armengol, J. and Sabater, S.: Diurnal variation in dissolved oxygen and carbon dioxide in
two low-order streams, Water Res., 32,1067−1074, 1998.

Guérin, F., Abril, G., Serça, D., Delon, C., Richard, S., Delmas, R., Tremblay, A., and Varfalvy, L.:
Gas transfer velocities of $CO_2$ and $CH_4$ in a tropical reservoir and its river downstream, J. Mar.
Syst., 66, 161–172, 2007.

Hope, D., Palmer, S.M., Billett, M.F., and Dawson, J.J.: Variations in dissolved CO and $CH_4$ in a first-
order stream and catchment: an investigation of soil-stream linkages, Hydrol. Process., 18, 3255–
75, 2004.

Ho, D. T., Coffineau, N., Hickman, B., Chow, N., Koffman, T., and Schlosser, P.: Influence of current
velocity and wind speed on air-water gas exchange in a mangrove estuary, Geophys. Res. Lett., 43,
3813–3821, doi:10.1002/ 2016GL068727, 2016.

Jähne, B., Huber, W., Dutzi, A., Wais, T., and Ilmberger, J.: Wind/wave-tunnel experiments on the
Schmidt number and wave field dependence of air-water gas exchange, In Gas Transfer at Water
Surfaces, Ed. W. Brutsaert,G.H. Jirka, pp. 303–309, Riedel, Hingham, MA, 1984.



Lauerwald, R., Laruelle, G. G., Hartmann, J., Ciais, P., and Regnier, P. A. G.: Spatial patterns in
$CO_2$ evasion from the global river network, Global Biogeochem. Cy., 29(5), 534–554,
DOI: 10.1002/2014GB004941, 2015.
Leopold, A., Marchand, C., Deborde, J., and Allenbach, M.: Biogeochemistry of an estuary surrounded
by mangroves under semi-arid climate (New Caledonia), Estuaries and Costs, 40, 3, 773-791,
2017.

Le, T. P. Q., Garnier, J., Billen, G., Thery, S., and Chau, V.M.: The changing flow regime and
sediment load of the Red River, Viet Nam, J. Hydrol., 334, 199– 214,
doi:10.1016/j.jhydrol.2006.10.020, 2007.
Le, T. P. Q., Billen, G., Garnier, J., Chau, V. M.: Long-term biogeochemical functioning of the Red
River (Vietnam): past and present situations, Reg. Environ. Change., DOI: 10.1007/s10113-014-
0646-4, 2015.

Le, T. P. Q., Dao, V. N., Rochelle-Newall, E., Garnier, J., Billen, G., Lu, X. X., Echetbet, H., Duong,
T. T., Ho, C. T., Nguyen, T. B. N., Nguyen, B. T., Nguyen, T. M. H., Le, N. D., and Pham, Q. L.:
Total organic flux of the Red River system (Vietnam), Earth. Surf. Proc. Land.,
DOI: 10.1002/esp.4107, 2017.
Liss, P.S. and Merlivat, L.: Air-sea gas exchange rates: introduction and synthesis. In The Role of Air-
Sea Exchange in Geochemical Cycling, ed. P Buat-Menard, 113–129, Reidel publishing company,
Boston, 1986.
Li, S., Lu, X. X., He, M., Yue, Z., Li L., and Ziegler, A. D.: Daily $CO_2$ partial pressure and $CO_2$
outgassing in the upper Yangtze River basin: A case study of the Longchuan River, China, J.
Hydrol., 466–467, 141–150, http://dx.doi.org/10.1016/j.jhydrol.2012.08.011, 2012.
Li, S., Lu, X. X., and Bush, R. T.: $CO_2$ partial pressure and $CO_2$ emission in the Lower Mekong River,
J. Hydrol., 504, 40–56, http://dx.doi.org/10.1016/j.jhydrol.2013.09.024, 2013.
Linn, D. M. and Doran, J. W.: Effect of water-filled pore space on carbon dioxide and nitrous oxide
production in tilled and non- tilled soils, Soil Sci. Soc. 48(6), 1267-272, 1984.
Lu, X. X.: Vulnerability of water discharge of large Chinese rivers to environmental changes: an
overview, Reg. Environ. Change, 4, 182–191, 2004.
Lu, X. X., Oeurng, C., Le T. P. Q., and Duong T. T.: Sediment budget as affected by construction of a
sequence of dams in the lower Red River, Viet Nam. Geomorphology 248, 125-133, 2015.
Marino, R., and Howarth, R.W.: Atmospheric oxygen exchange in the Hudson River: Dome
measurements and comparison with other natural waters, Estuaries, 16, 433-445, 1993.
Meybeck, M., Chapman, D. V. and Helmer, R.: Global freshwater quality: a first assessment.
Cambridge, MA, World Health Organization/United Nations Environment Programme, Basil
Blackwell, Inc. 306 p, 1989.



Melack, J. M. and Engle, D. L.: An organic carbon budget for an Amazon floodplain lake, Verh. Int.
Ver. Limnol., 30, 1179–1182, 2009.
Moacyr, A., Carlos, N., Doris, V., and Nathalie, L.: Nutrient input and $CO_2$ flux of a tropical coastal
fluvial system with high population density in the northeast region of Brazil, J. Water Resource
Prot., 5, 362-375, 2013.
MONRE.: Vietnamese Ministry of Environment and Natural Resources, Report Annual on
Hydrological Observation in Vietnam, Hanoi, 2016.
Nguyen, N. S., Hua, C. T., Nguyen, C. H., Nguyen, V. T., Lang, V. K, Pham, V. N., and Nguyen, V.
T.: Case study report on Red River Delta in Vietnam - Project on integrated management and
conservation of near shore coastal and marine areas in East Asia region (EAS-35) United Nations
Environment program, Regional coordinating for the East Seas (ESA/RCU), U.N. Environ.
Programme, Nairobi, 78pp, 1995.
Nomura, D., Yoshikawa-Inoue, H., Kobayashi, S., Nakaoka, S., Nakata, K., and Hashida, G.: Winter-
to-summer evolution of $pCO_2$ in surface water and air–sea $CO_2$ flux in the seasonal ice zone of the
Southern Ocean, Biogeosciences, 11, 5749–5761, doi:10.5194/bg-11-5749-2014, 2014.
Parkin, T. B. and Kaspar, T. C.: Temperature controls on diurnal carbon dioxide flux, Soil Sci. Soc.
Am. J., 67, 1763−1772, 2003.
Peter, H., Singer, G. A., Preiler, C., Chifflard, P., Steniczka, G., and Battin, T. J.: Scales and drivers of
temporal $pCO_2$ dynamics in an Alpine stream. J. Geophys. Res. Biogeosciences, 119 (6), 1078–
1091, doi 10.1002/2013JG002552, 2014.
Patricia, S., Raymond, H. H., and Kelly, C.A.: Continuous measurement of $CO_2$ for estimation of air-
water fluxes in lakes: An in situ technique, Limnol. Oceanogr., 40(3), 575-581, 1995.
Roulet, N. T., Crill, P., Comer, N., Dove, A., and Boubonniere, R.: $CO_2$ and $CH_4$ flux between a boreal
beaver pond and the atmosphere, J. Geophys. Res. D, 102, 29313−29319, 1997.
Ran, L., Lu, X. X., Richey, J. E., Sun, H., Han, J., Yu, R., Liao, S., and Yi, Q.: Long term spatial and
temporal variation of $CO_2$ partial pressure in the Yellow River, China, Biogeosciences, 12, 921-
932, DOI: 10.5194/bg-12-921-2015, 2015.
R Core Team. R.,: A language and environment for statistical computing. Vienna, Austria: R
Foundation for Statistical Computing, Retrieved from https://www.rproject.org/, 2016.
Rasera M de Fatima, F.L., Krusche, A.V., Richey, J. E., Ballester, M. V. R., and Victória, R. L.: Spatial
and temporal variability of $pCO_2$ and $CO_2$ efflux in seven Amazonian Rivers, Biogeochem.,
116:241–259, doi: 10.1007/s10533-013-9854-0, 2013.
Raymond, M. E., and Ruddiman, W. F.: Tectonic forcing of late Cenozoic climate, Nature, 359, 117-
122, 1992.

Raymond, P. A., and Cole, J. J.: Gas Exchange in Rivers and Estuaries: Choosing a Gas Transfer
Velocity, Estuaries, 24(2), 312-317, doi:10.2307/1352954, 2001.




Raymond, P. A., Caraco, N. F. and Cole, J. J.: Carbon dioxide concentration and atmospheric flux in
the Hudson River, Estuaries, 20, 381–390, 1997.
Raymond, P. A., Zappa, C. J., Butman, D., Bott, T. L., Potter, J., Mulholland, P., Laursen, A. E.,
McDowell, W.H., and Newbold, D.: Scaling the gas transfer velocity and hydraulic geom- etry in
streams and small rivers, Limnol. Oceanogr., 2, 41–53, doi:10.1215/21573689-1597669, 2012.
Raymond, P. A., et al. : Global carbon dioxide emissions from inland waters, Nature, 503(7476), 355–
359. Doi:10.1038/nature12760, 2013.
Regnier, P., Friedlingstein, P., Ciais, P., Mackenzie, F. T., Gruber, N., Janssens, I. A., Laruelle, G. G.,
Lauerwald, R., Luyssaert, S., Andersson, A. J., Arndt, S., Arnosti, C.,  Borges, A. V., and Dale,
A.W.: Anthropogenic perturbation of the carbon fluxes from land to ocean, Nat. Geosci., 6(8),
597–607, doi:10.1038/ngeo1830, 2013.
Rey, A., Belelli-Marchesini, L., Were, A., Serrano-Ortiz, P., Etiope, G., Papale, D., Domingo, F., and
Pegoraro, E.: Wind as a main driver of the net ecosystem carbon balance of a semiarid
Mediterranean steppe in the South East of Spain, Glob. Change Biol., 18, 539–554, doi:
10.1111/j.1365-2486.2011.02534.x, 2012.
Richey, J. E., Melack, J. M., Aufdenkampe, A. K., Ballester, V. M., and Hess, L. L.: Outgassing from
Amazonian rivers and wetlands as a large tropical source of atmospheric $CO_2$, Nature, 416, 617–
620, 2002.

Roulet, N. T., Crill, P., Comer, N., Dove, A., and Boubonniere, R.: $CO_2$ and $CH_4$ flux between a boreal
beaver pond and the atmosphere, J. Geophys. Res. D, 102, 29313−29319, 1997.
Rudorff, C. M., Melack, J. M., Sally, M., Cláudio, C. F. B., and Evlyn, M. L. M. N.: Seasonal and
spatial variability of $CO_2$ emission from a large floodplain lake in the lower Amazon, J. Geophys.
Res., 116, G04007, doi:10.1029/2011JG001699, 2011,
Semiletov, I., Makshtas, S., Akasofu, S., and Andreas, E. L.: Atmospheric $CO_2$ balance: The role of
Arctic sea ice, Geophys. Res. Lett., 31(5), doi:10.1029/2003GL017996, 2004.
Servais, P., Barillier, A., and Garnier, J.: Determination of the biodegradable fraction of dissolved and
particulate organic carbon in waters. Int. J. Limnol. 31(1):75-80. Doi:10.1051/limn/1995005, 1995.
Striegl, R. G., Dornblaser, M. M., Aiken, G. R., Wickland, K. P., and Raymond, P. A.: Carbon export
and cycling by the Yukon, Tanana, and Porcupine rivers, Alaska, 2001–2005, Water Resour. Res.,
43, W02411, doi:10.1029/2006WR005201, 2007.
Striegl, R. G., Dornblaser, M. M., McDonald, C. P., Rover, J. R., and Stets, E. G.: Carbon dioxide and
methane emissions from the Yukon River system, Global Biogeochem. Cy., 26, GB0E05,
doi:10.1029/2012GB004306, 2012.
Sun, H. G., Han, J., Lu, X. X., Zhang, S. R., and Li, D.: An assessment of the riverine carbon flux of
the Xijiang River during the past 50 years, Quatern. Int., 226, 38-43, 2010.



Tranvik, L .J., Downing, J. A., Cotner, J. B., Loiselle, S. A., Striegl, R. G., Ballatore, T. J., Dillon, P.,
Finlay, K., Fortino, K., Knoll, L.B., Kortelainen, P.L.: Lakes and reservoirs as regulators of carbon
cycling and climate. Limnol. Oceanogr., 54 (6, part 2), 2298–2314, 2009.
Trinh, A. D., Vachaud, G., Bonnet, M. P., Prieur, N., Vu, D. L., and Le, L. A.: Experimental
investigation and modelling approach of the impact of urban wastewater on a tropical river; a case
study of the Nhue River, Hanoi, Vietnam, J. Hydrol., 334, 347–358, doi:10.1016/j.
jhydrol.2006.10.022, 2007.
Trinh, A. D., Giang, N. H., Vachaud, G., and Choi. S.U.: Application of excess carbon dioxide partial
pressure (EpCO$_2$) to the assessment of trophic state of surface water in the Red River Delta of
Vietnam, Int. J. Environ. Stud., 66(1), 27–47, doi:10.1080/00207230902760473, 2009.
Trinh, A.D., Meysman, F., Rochelle-Newall E., and Bonnet, M. P.: Quantification of sediment-water
interactions in a polluted tropical river through biogeochemical modeling, Global Biogeochem
Cy., 26, GB3010, doi:10.1029/2010GB003963, 2012.
Takahashi, T., Sutherland, S.C., Sweeney, C., Poisson, A., Metzl, N., Tilbrook, B., Bates, N.,
Wanninkhof, R., Feely, R. A., Sabine, C., Olafsson, J., and Nojiri, Y.:  Global air-sea CO$_2$ flux
based on climatological surface ocean pCO$_2$ and seasonal biological and temperature effects,
Deep-Sea Res., 49, 1601–1622, 2002.
Vachon, D., Prairie, Y.T., and Cole, J.J.: The relationship between near-surface turbulence and gas
transfer velocity in freshwater systems and its implications for floating chamber measurements of
gas exchange, Limnol. Ocean., 55, 1723–1732, 2010.
Xu, J., Lee, X., Xiao, W., Cao, C., Liu, S., Wen, X., Xu, J., Zhang, Z., and Zhao, J.: Interpreting the
$^{13}$C= $^{12}$C ratio of carbon dioxide in an urban airshed in the Yangtze River Delta, China, Atmos.
Chem. Phys., 17, 3385–3399, doi:10.5194/acp-17-3385-2017, 2017.
Yao, G., Quanzhou, G., Zhengang, W., Xiakun, H., Tong, H., Yongling, Z., Shulin, J., and Jian, D.:
Dynamics of CO$_2$ partial pressure and CO$_2$ outgassing in the lower reaches of the Xijiang River, a
subtropical monsoon river in China, Sci. Total Environ., 376, 255–266. DOI:
10.1016/j.scitotenv.2007.01.080, 2007.
Walling, D. E., and Fang, D.: Recent trends in the suspended sediment loads of the world's rivers,
Glob. Planet. Change, 39(1-2), 111-126, 2003.
Walling, D. E.: Human impact on land-ocean sediment transfer by the world's rivers, Geomorphology,
79(3-4), 192-216, 2006.

Wang, F., Wang, Y., Zhang, J., Xu,  H., and Wei, X.: Human impact on the historical change of CO$_2$
degassing flux in the River Changjiang, Chem. Trans., Doi:10.1186/1467-4866-8-7, 2007.
Wang, Y., Munger, J. W., Xu, S., McElroy, M. B., Hao, J., Nielsen, C. P. and Ma, H.: CO$_2$ and its
correlation with CO at a rural site near Beijing: implications for combustion efficiency in China,
Atmos. Chem. Phys., 10, 8881–8897, doi:10.5194/acp-10-8881-2010, 2010.





Wang, F., Wang, B., Liu, C., Wang, Y., Guan, J., Liu, X., and Yu, Y.: Carbon dioxide emission from
surface water in cascade reservoirs-river system on the Maotiao River, southwest of China, Atmos.
Environ., 45(23), 3827–3834, 2011.

Weiss, R. F.: Carbon dioxide in water and seawater: the solubility of a non ideal gas, Mar. Chem., 2(3),
203-215, 1974.

Wit, F., Muller, D., Baum, A., Warneke, T., Pranowo, W. S., Muller, M. and Rixen T.: The impact of
disturbed peatlands on river outgassing in Southeast Asia, Nat. Commun. 6:10155, DOI:
10.1038/ncomms10155, 2015.

Zemmelink, H. J ., Delille, B., Tison, J. L., Hintsa, E. J., Houghton, L. and Dacey, J. W. H.: $CO_2$
deposition over the multi-year ice of the western Weddell Sea, Geophys. Res. Lett., 33, L13606,
doi:10.1029/2006GL026320, 2006.





**List of tables**
**Table 1.** Average values of river water discharge at five hydrological stations of the Red River in 2014
**Table 2.** Mean values of the different physico-chemical variables at 5 sites in wet and dry season in
2014.

**Table 3.** Wind speed, $k_{600}$ parameterization, and calculated water-air $CO_2$ fluxes for day and night at
five hydrological stations of the Red River in dry and wet seasons in 2014.
**Table 4.** Summary of the statistical analysis at Hanoi station with the environmental variables.
**Table 5.** $CO_2$ out-gassing flux from some World Rivers.







**Table 1.** Average values of river water discharge at five hydrological stations of the Red River in 2014.

| Hydrological stations | Altitude (m a.s.l.) | Latitude | Average daily water discharge in 2014, $m^3 \ s^{-1}$ | Water discharge, $m^3 \ s^{-1}$ | | | |
|---|---|---|---|---|---|---|---|
| | | | | *Wet season* | | *Dry season* | |
| | | | | Mean value in wet season in 2014 (May – Oct) | On the date of measurement Sept 2014 | Mean value in wet season in 2014 (May – Oct) | On the date of measurement in Nov 2014 |
| Yen Bai | 56 | $104^051'$- $21^042'$ | $527 \pm 515$ | $788 \pm 459$ | $840 \pm 68$ | $262 \pm 530$ | $260 \pm 18$ |
| Hoa Binh | 23 | $105^019'$- $20^049'$ | $1,369 \pm 833$ | $1,907 \pm 451$ | $2,189 \pm 39$ | $825 \pm 515$ | $868 \pm 319$ |
| Vu Quang | 25 | $105^015'$- $21^034'$ | $1,302 \pm 517$ | $1,618 \pm 378$ | $2,240 \pm 88$ | $982 \pm 284$ | $725 \pm 11$ |
| Hanoi | 5 | $105^051'$- $21^001$ | $1,867 \pm 1089$ | $2,598 \pm 780$ | $3,296 \pm 86$ | $1,127 \pm 490$ | $1,915 \pm 149$ |
| Ba Lat | 0 | $106^000'$ - $19^030'$ | $615 \pm 293$ | $824 \pm 200$ | $1,269 \pm 93$ | $403 \pm 96$ | $453 \pm 31$ |





**Table 2.** Average values in day and night times of the different physico-chemical variables at 5 sites in wet and dry seasons in 2014.

| Stations | Temperature <br> $^0$C | pH | TAlk <br> mg L$^{-1}$ | Salinity | Chl-a <br> µg L$^{-1}$ | Turbidity <br> NTU | Conductivity <br> mS cm$^{-1}$ | DOC <br> mg L$^{-1}$ | POC <br> mg L$^{-1}$ | DO <br> % | Measured $p$CO$_2$ <br> ppm | Calculated $p$CO$_2$ <br> ppm |
|---|---|---|---|---|---|---|---|---|---|---|---|---|
| Wet season | | | | | | | | | | | | |
| *1-Yen Bai* | | | | | | | | | | | | |
| Day | 26.4± 0.1 | 8.2±0.1 | 105.1±5.2 | 0.1±0.0 | 3.1±0.1 | 141.6±8.6 | 0.2±0.0 | 1.5±0.2 | 2.1±0.4 | 69.9±0.2 | 964.3±9.9 | 269.6±36.4 |
| Night | 26.6±0.0 | 8.3±0.0 | 103.8±3.3 | 0.1±0.0 | 3.1±0.1 | 135.4±4.0 | 0.2±0.0 | 1.4±0.2 | 1.9±0.2 | 70.4±0.1 | 979.5±9.6 | 225.7±10.4 |
| *2-Vu Quang* | | | | | | | | | | | | |
| Day | 26.8±0.1 | 8.1±0.0 | 148.9±6.7 | 0.1±0.0 | 1.2±0.2 | 51.4±7.5 | 0.2±0.0 | 1.1±0.3 | 1.4±0.2 | 63.6±1.2 | 1598.7±53.3 | 453.8±55.2 |
| Night | 27.0±0.1 | 8.2±0.0 | 144.9±3.3 | 0.1±0.0 | 1.2±0.2 | 49.9±5.3 | 0.2±0.0 | 1.0±0.2 | 1.4±0.2 | 63.2±0.7 | 1583.0±36.6 | 404.0±12.5 |
| *3-Hoa Binh* | | | | | | | | | | | | |
| Day | 26.5±0.1 | 7.8±0.0 | 110.4±3.3 | 0.1±0.0 | 0.8±0.3 | 42.5±4.7 | 0.2±0.0 | 1.5±0.4 | 1.1±0.2 | 54.9±0.2 | 3827.1±60.6 | 881.1±123.0 |
| Night | 26.4±0.0 | 7.8±0.0 | 107.8±5.4 | 0.1±0.0 | 1.2±0.0 | 41.0±0.1 | 0.2±0.0 | 1.3±0.2 | 1.1±0.1 | 55.1±0.6 | 3830.2±19.1 | 722.0±23.4 |
| *4-Ha Noi* | | | | | | | | | | | | |
| Day | 28.6±0.2 | 8.0±0.0 | 84.3±1.9 | 0.1±0.0 | 2.0±0.6 | 88.9±1.3 | 0.2±0.0 | 4.7±0.6 | 2.0±0.3 | 64.0±0.4 | 1412.3±4.2 | 336.6±38.6 |
| Night | 28.6±0.2 | 8.1±0.0 | 84.5±1.5 | 0.1±0.0 | 2.7±0.1 | 88.5±2.7 | 0.2±0.0 | 4.2±0.9 | 2.2±0.4 | 63.5±0.3 | 1411.4±6.7 | 279.4±18.7 |
| *5-Ba Lat* | | | | | | | | | | | | |
| Day | 28.9±0.4 | 8.0±0.1 | 116.4±4.6 | 0.3±0.3 | 1.8±0.3 | 47.7±8.8 | 0.6±0.6 | 1.5±0.4 | 1.1±0.4 | 66.1±1.4 | 1489.1±104.2 | 495.4±108.6 |
| Night | 28.8±0.3 | 8.1±0.0 | 114.9±3.5 | 0.1±0.1 | 2.5±0.1 | 81.3±10.0 | 0.3±0.2 | 1.7±0.5 | 1.5±0.2 | 65.1±1.6 | 1483.1±117.5 | 430.2±31.8 |
| Dry season | | | | | | | | | | | | |





| 1-Yen Bai | | | | | | | | | | | | |
|---|---|---|---|---|---|---|---|---|---|---|---|---|
| Day | 24.1±0.5 | 8.1±0.1 | 113.9±7.9 | 0.1±0.0 | 1.2±0.3 | 49.3±7.9 | 0.2±0.0 | 1.3±0.3 | 1.4±0.1 | 69.3±0.7 | 995.8±17.5 | 376.6±140.1 |
| Night | 24.2±0.3 | 8.2±0.0 | 109.3±2.8 | 0.1±0.0 | 1.6±0.2 | 42.5±4.7 | 0.2±0.0 | 1.2±0.2 | 1.2±0.3 | 69.1±0.5 | 1030.6±21.5 | 266.6±7.8 |
| 2-Vu Quang | | | | | | | | | | | | |
| Day | 24.7±0.2 | 8.3±0.0 | 134.9±5.1 | 0.1±0.0 | 1.0±0.2 | 28.1±1.7 | 0.2±0.0 | 1.1±0.2 | 1.1±0.2 | 67.0±0.9 | 1235.3±76.2 | 301.8±36.4 |
| Night | 24.8±0.4 | 8.4±0.0 | 129.3±2.7 | 0.1±0.0 | 1.4±0.2 | 32.4±3.7 | 0.2±0.0 | 1.1±0.2 | 1.2±0.1 | 69.0±0.9 | 1163.3±86.3 | 235.2±13.0 |
| 3-Hoa Binh | | | | | | | | | | | | |
| Day | 26.3±0.0 | 7.8±0.0 | 122.5±6.1 | 0.1±0.0 | 0.5±0.1 | 16.9±0.3 | 0.2±0.0 | 0.9±0.2 | 1.0±0.2 | 51.5±0.4 | 2399.3±33.6 | 869.7±96.9 |
| Night | 26.3±0.0 | 7.8±0.0 | 120.6±6.1 | 0.1±0.0 | 0.5±0.1 | 17.1±0.5 | 0.2±0.0 | 0.9±0.2 | 1.1±0.2 | 51.3±0.2 | 2458.9±14.0 | 833.1±85.3 |
| 4-Ha Noi | | | | | | | | | | | | |
| Day | 23.8±0.1 | 8.2±0.0 | 123.5±2.4 | 0.1±0.0 | 1.7±0.2 | 65.2±1.8 | 0.2±0.0 | 2.7±0.7 | 1.5±0.3 | 66.8±0.4 | 1141.3±33.5 | 333.5±43.2 |
| Night | 23.8±0.1 | 8.3±0.0 | 123.8±1.5 | 0.1±0.0 | 1.6±0.1 | 62.6±0.7 | 0.2±0.0 | 2.0±0.7 | 1.3±0.1 | 67.1±0.3 | 1136.0±24.2 | 300.6 ± 12.6 |
| 5-Ba Lat | | | | | | | | | | | | |
| Day | 23.7±0.1 | 8.3±0.0 | 152.9±6.6 | 3.9±2.4 | 1.8±0.2 | 34.1±8.3 | 6.6±3.4 | 1.4±0.2 | 2.0±0.4 | 70.0±0.5 | 751.4±49.3 | 311.8± 20.5 |
| Night | 23.4±0.1 | 8.3±0.0 | 150.3±5.6 | 3.3±1.6 | 1.3±0.2 | 28.8±4.2 | 5.7±2.6 | 1.2±0.2 | 1.9±0.4 | 68.8±0.6 | 881.0±88.4 | 331.6± 20.4 |





**Table 3.** Wind speed, $k_{600}$ parameterization, and calculated water-air $CO_2$ fluxes for daytime and nighttime at five hydrological stations of the Red River in dry and wet seasons in 2014.

| | Wind speed m s$^{-1}$ | $k_{600}$ cm h$^{-1}$ | Water-air $CO_2$ flux mmol m$^{-2}$ d$^{-1}$ | |
| --- | --- | --- | --- | --- |
| | | | With $pCO_2$ measured from equilibrator | With $pCO_2$ calculated from $CO_2$-SYS |
| **Wet season** | | | | |
| Yen Bai | | | | |
| Day | 1.1±0.6 | 2.9±0.6 | 13.0±2.7 | 7.2±2.3 |
| Night | 0.5±0.6 | 2.3±0.5 | 10.9±2.5 | 4.0±1.0 |
| Vu Quang | | | | |
| Day | 0.9±0.7 | 2.7±0.7 | 27.7±7.3 | 16.8±5.6 |
| Night | 0.4±0.6 | 2.3±0.5 | 20.4±3.9 | 12.2±2.9 |
| Hoa Binh | | | | |
| Day | 1.3±1.1 | 3.3±1.2 | 82.7±36.3 | 48.4±23.4 |
| Night | 0.2±0.5 | 2.1±0.5 | 61.3±15.3 | 23.1±4.3 |
| Hanoi | | | | |
| Day | 1.8±0.7 | 3.6±0.8 | 29.4±7.1 | 15.0±4.6 |
| Night | 1.2±0.6 | 2.9±0.6 | 24.8±4.6 | 8.7±2.8 |
| Ba Lat | | | | |
| Day | 0.5±0.5 | 2.3±0.5 | 19.0±2.3 | 17.9±7.8 |
| Night | 0.2±0.3 | 2.0±0.3 | 18.1±4.0 | 12.7±2.5 |
| **Dry season** | | | | |
| 1-Yen Bai | | | | |
| Day | 1.4±0.9 | 3.2±1.0 | 14.4±3.7 | 12.0±7.1 |
| Night | 0.5±0.8 | 2.4±0.8 | 12.5±4.7 | 4.9±1.7 |
| 2-Vu Quang | | | | |
| Day | 1.3±0.6 | 3.1±0.6 | 20.3±5.1 | 8.9±3.1 |
| Night | 0.7±1.3 | 2.8±1.5 | 15.8±8.0 | 4.4±2.0 |
| 3- Hoa Binh | | | | |
| Day | 1.2±0.8 | 3.0±0.8 | 50.2±13.6 | 40.3±8.9 |
| Night | 0.5±0.5 | 2.3±0.5 | 37.2±6.9 | 29.8±4.8 |
| 4-Hanoi | | | | |
| Day | 2.4±0.5 | 4.6±0.9 | 28.5±5.8 | 15.2±6.6 |
| Night | 1.4±0.5 | 3.2±0.5 | 19.3±3.0 | 8.5±1.3 |
| 5-Ba Lat | | | | |
| Day | 3.1±1.5 | 6.5±3.8 | 18.9±14.6 | 18.4±10.7 |
| Night | 1.3±0.8 | 3.1±0.8 | 13.1±2.8 | 9.9±1.9 |




**Table 4**. Relationship between $CO_2$ outgassing flux with other water quality variables at Hanoi station

| Variables | Temp. | DOC | POC | DIC | $pCO_2$ | pH | Chl-a | %DO | $CO_2$ Flux |
|---|---|---|---|---|---|---|---|---|---|
| Temp. | 1 | | | | | | | | |
| DOC | **0.72** | 1 | | | | | | | |
| POC | **0.74** | 0.53 | 1 | | | | | | |
| DIC | **-0.99** | -0.73 | -0.73 | 1 | | | | | |
| $pCO_2$ | -0.11 | -0.05 | 0.00 | 0.19 | 1 | | | | |
| pH | **-0.90** | -0.69 | -0.70 | **0.87** | -0.31 | 1 | | | |
| Chl_a | **0.65** | 0.29 | 0.47 | -0.65 | -0.51 | -0.37 | 1 | | |
| %DO | **-0.96** | -0.70 | -0.72 | **0.97** | 0.19 | **0.84** | -0.65 | 1 | |
| $CO_2$ Flux | 0.06 | 0.17 | 0.04 | -0.02 | **0.60** | -0.31 | -0.11 | -0.05 | 1 |



**Table 5.** $CO_2$ flux out-gassing in some World Rivers.

| River or Tributary | Location | Country | Mean $pCO_2$ µatm | $F_{CO2}$ mmol m$^{-2}$ day$^{-1}$ | $k_{600} \pm SD$ cm h$^{-1}$ | $U_{10}$ m s$^{-1}$ | References |
|---|---|---|---|---|---|---|---|
| Red | | Vietnam | 1,589 | 28.6 ± 19.3 | 3.25 ± 0.92 | 1.2 ± 0.7 | This study |
| Mekong | Downstream | Laos and Cambodia | 703 – 1597 | 88.1 -378.4 | 12.4 - 44.5 | 1.3 – 4.9 | *Alin et al., 2011* |
| Tonle Sap | Stung Siem Reap | Cambodia | 3,066 | 139.1 | 5.6 ± 0.9 | 0.8 | *Alin et al., 2011* |
| Tonle Sap | Pousat River | Cambodia | 1,404 | 98.5 | 10.8 ± 2.8 | nd | *Alin et al., 2011* |
| Musi | | Indonesia | 4,316±928 | 5±1.1 | 21.8±4.7 | nd | *Wit et al., 2015* |
| Batanghari | | Indonesia | 2,400±18 | 1.8±0.4 | 21.8±4.7 | nd | *Wit et al., 2015* |
| Indragiri | | Indonesia | 5,777±527 | 9.7±2.2 | 21.8±4.7 | nd | *Wit et al., 2015* |
| Siak | | Indonesia | 8,555±528 | 8.3±1.9 | 22.0±4.7 | nd | *Wit et al., 2015* |
| Lupar | | Malaysia | 1,274±148 | 13±3.0 | 26.5±9.3 | nd | *Wit et al., 2015* |
| Saribas | | Malaysia | 1,159±29 | 14.6±3.3 | 17.0±13.6 | nd | *Wit et al., 2015* |
| Changjiang | | China | 1,297±901 | 143 | 8 - 15 | 5 | *Wang et al., 2007* |
| Maotiao | | China | 3740 | 108 | 10 | nd | *Wang et al. 2011* |
| Longchuan | | China | 2,100 | 156 | 8 | nd | *Li et al., 2012* |
| Yellow | | China | 2,810 ± 1,985 | 856±409 | 42.1 ± 16.9 | nd | *Ran et al., 2015* |
| Xijiang | | China | 2,600 | 357 | 15 | 2.7 | *Yao et al., 2007* |
| Negro | South America | | 4,260±1387 | 855±294 | 38.3±19.1 | nd | *Rasera et al., 2013* |
| Solimoes | | | 6,691±55 | 518±17 | 8.0±2.2 | | |
| Arguaia | | | 2,674±802 | 207±104 | 9.0±1.8 | | |
| Javaes | | | 3,065±1208 | 156±69 | 5.0±0.9 | | |
| Caxiuana | | | 4,849±208 | 778±17 | 20.5±2.5 | | |
| Teles Pires | | | 1,624±425 | 78±43 | 9.2±2.3 | | |
| Cristalino | | | 3,507±482 | nd | 8.1±1.9 | | |
| Krishna | | India | **17,205±3500** | nd | nd | | *Sarma et al., 2012* |
| Godavari | | | **49,819±1042** | | | | |
| Mahanadi | | | **95,859±2234** | | | | |
| Ganges | | | 5,029±100 | | | | |





| Name | Location | Country/Region | pCO₂ | DIC | DOC | — | Reference |
|---|---|---|---|---|---|---|---|
| Gaderu Creek | | India | 2,215 ± 864a | 56.0 ± 100.9 | 4 ± 5 | 1.4 ± 1.9 | *Borges et al., 2003* |
| Rhone | | France | 2,015±944 | nd | 15 | nd | *Cole et al., 2001* |
| Negro | | South America | 4,260±1387 | 855±294 | 38.3±19.1 | nd | *Rasera et al., 2013* |
| Solimoes | | | 6,691±55 | 518±17 | 8.0±2.2 | | |
| Arguaia | | | 2,674±802 | 207±104 | 9.0±1.8 | | |
| Javaes | | | 3,065±1208 | 156±69 | 5.0±0.9 | | |
| Caxiuana | | | 4,849±208 | 778±17 | 20.5±2.5 | | |
| Teles Pires | | | 1,624±425 | 78±43 | 9.2±2.3 | | |
| Cristalino | | | 3,507±482 | nd | 8.1±1.9 | | |
| Capibaribe | Coastal region of the State of Pernambuco | Brazil | 8,340 | 225 | nd | 2.2 | *Moacyr et al., 2013* |
| Hudson | | USA | 1,125±403 | nd | 4.1 | 3.5± 2.0 | *Raymond et al., 1997* |
| Yukon | Upper | North America | 1,220 ± 9.1 | 6 | 1.25 | nd | *Striegl et al., 2007; 2012* |
| | Middle | | 1,890 ± 9.9 | 62 | 7.92 | nd | |
| | Lower | | 3,090 ± 16.5 | 193 | 15 | nd | |
| Ottawa | | Canada | 1,200 | 80.8 | 4 | nd | *Telmer and Veizer, 1999* |
| Amazon | | | 4,350 ± 1900 | 190 ± 55 | 9.6 ± 3.8 | 1 - 3 | *Richey et al., 2002* |
| Mississippi | | | 100 - 600 | 270 | 3.9 | 5.3 | *Dubois et al., 2010; Lohrenz and Cai, 2006* |
| Nagada Creek | The northern Papua New Guinea coast | | 799 ± 357a | 43.6 ± 33.2 | 8 ± 6 | 3.0 ± 2.1 | *Borges et al., 2003* |
| Norman's Pond | Bahamas archipelago | | 165 ± 86a | 13.8 ± 8.3 | 13 ± 3 | 5.5 ± 1.3 | *Borges et al., 2003* |

[a] calculated the values for the $p$CO$_2$ Water-Air Gradient (ΔpCO$_2$ in μatm)

nd. No data





**List of figures**

**Figure 1**: The Red River system and sampling sites.

**Figure 2**: Spatial and seasonal variation of different environmental variables in the Red River system in 2014

**Figure 3:** Comparison the result of riverine $p\mathrm{CO_2}$ of the lower Red River by measured (equilibrator) and calculated ($\mathrm{CO_2\_SYS}$) methods

**Figure 4:** Spatial and seasonal variation of $\mathrm{CO_2}$ flux out-gassing in the Red River system in 2014.

**Figure 5:** Relationship between $p\mathrm{CO_2}$, $f\mathrm{CO_2}$ and water temperature at 5 sites observed of the lower Red River in dry season in 2014

**Figure 6:** Relationship between environmental variables and $p\mathrm{CO_2}$ or $\mathrm{CO_2}$ flux at five sites of the Red River.

**Figure 7:** : Relationship between $p\mathrm{CO_2}$, $f\mathrm{CO_2}$ and river discharge at 5 sites observed of the lower Red River in wet and dry season in 2014.





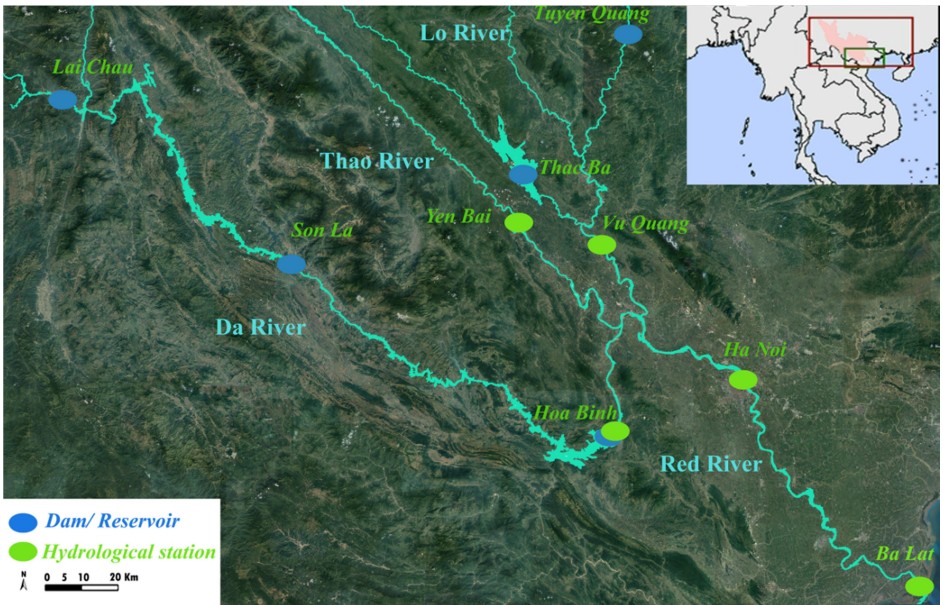

**Figure 1**. **The Red River system and sampling sites**



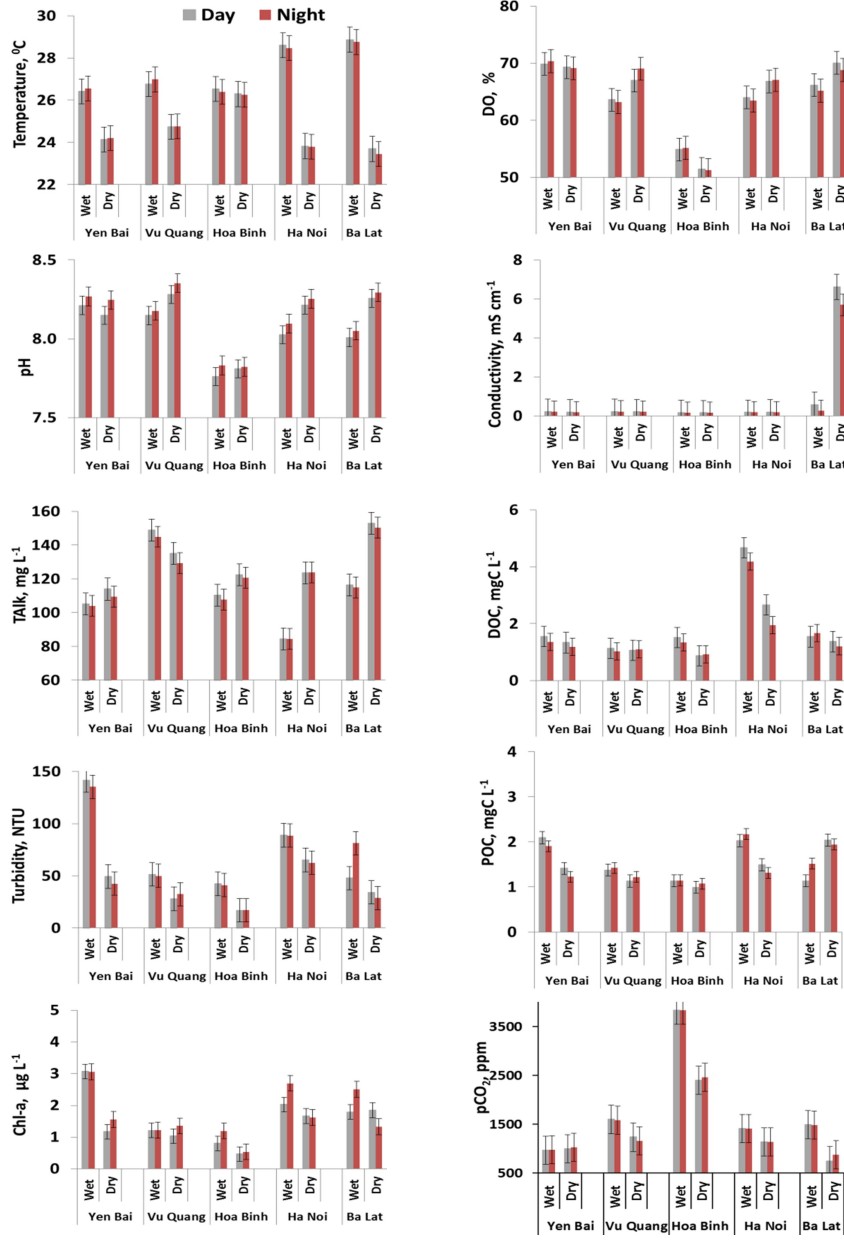

**Figure 2. Spatial and seasonal variation of different environmental variables in the Red River system in 2014**



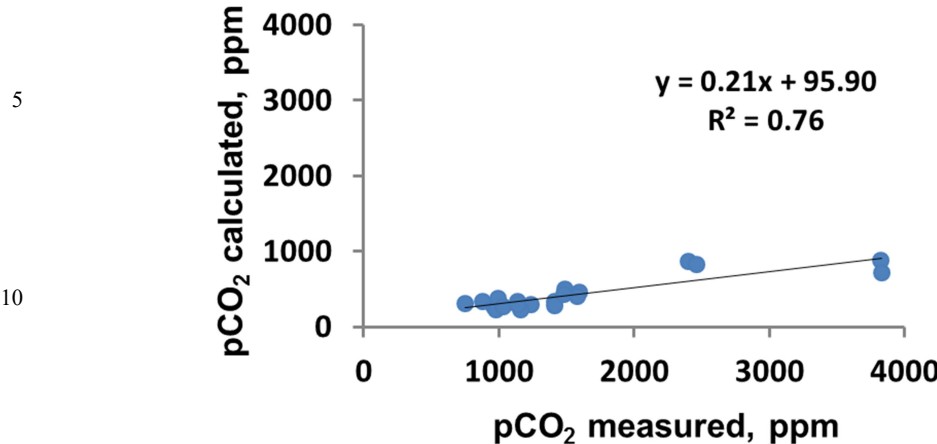

15 **Figure 3**. **Comparison the result of riverine $p$CO$_2$ of the lower Red River by measured (equilibrator) and calculated (CO2_SYS) methods**





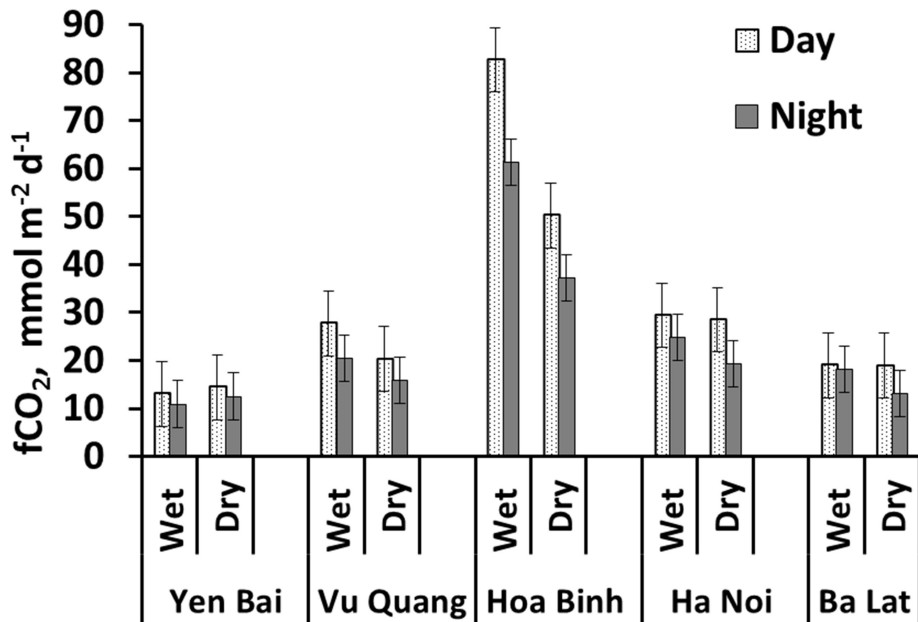

5          **Figure 4. Spatial and seasonal variation of CO$_2$ flux out-gassing in the Red River system in 2014.**

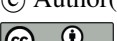



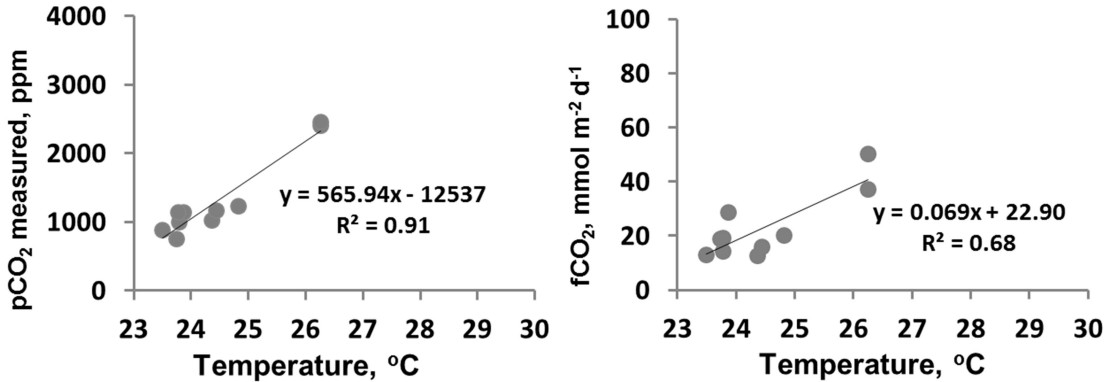

**Figure 5**. Relationship between $p$CO$_2$, fCO$_2$ and water temperature at 5 sites observed of the lower Red River in dry season in 2014.





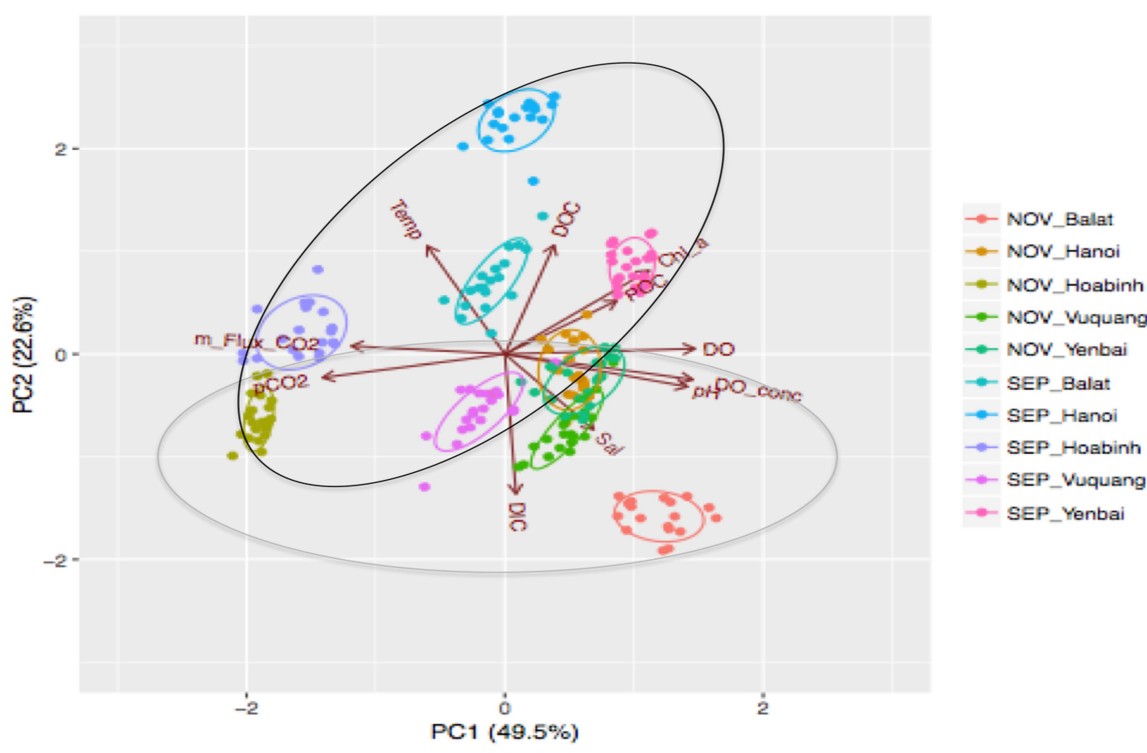

**Figure 6. Relationship between environmental variables and $p$CO$_2$ or CO$_2$ flux at five sites of the Red River**



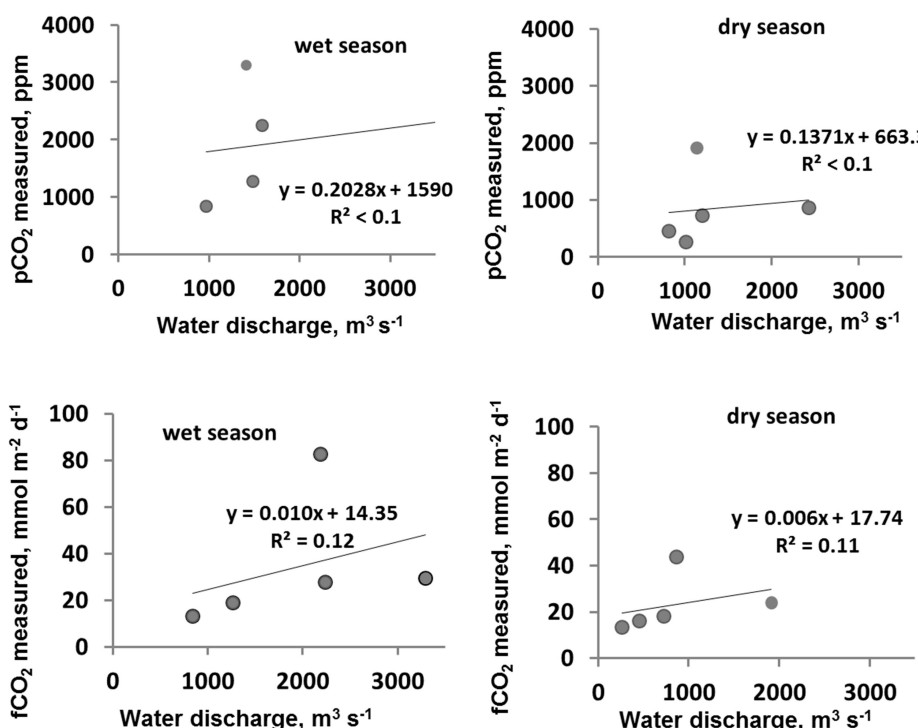

10 **Figure 7. Relationship between $p$CO$_2$, $f$CO$_2$ and river discharge at 5 sites observed of the lower Red River in wet and dry season in 2014.**