# Peer review of "CO2 partial pressure and CO2 emission along the lower Red River (Vietnam)"

_Biogeosciences, 2017_

## Short Comment (SC1) · 8 Jan 2018

This paper documents the chemical conditions and concentrations of dissolved carbon dioxide in the Red River system of Vietnam. The data contribute to the "database" of concentration values for the globe, with one goal of further constraining the CO2 source strength of inland waters. Therefore, the data are valuable on their own, especially given that they fill in geographic gaps for SE Asia. The main criticism of this paper is the use of a wind-driven gas exchange model. While criticism of gas exchange models are prevalent within the community of researchers, this example is especially problematic as it relies on unlikely drivers of turbulence (and thus gas exchange) in riverine systems. There is some evidence that gas exchange is enhanced by some wind patterns in very large rivers, however, gas exchange in rivers is not considered

to be a major driver. Rather, it is turbulence generated by water flow that drives gas exchange rates in these systems. Therefore, the CO2 emission estimates are not only biased, as recognized by the authors, but are likely to be highly inaccurate due to the model selected. It is hard to believe the results without some other line of confirmation. In addition to the criticism of the estimates of gas exchange, I did not find the discussion points to be well supported by the data especially given the limited time and geographic scope of the measurements. There is simply not enough evidence to support any of the inferred drivers of CO2 variability in this river system.

Specific Comments:

48: what references support plate tectonics as major drivers of carbon fluxes in this system?

53: are changes in sediments the hypothesized drivers of changing carbon fluxes in this study?

184: if exchange is less related to wind, then what is the justification for using this model in the present study?

214: such low temperature variability leads to skepticism of this environmental para-menter being a significant driver of CO2 variability. In addition, the broad conclusion here is that water chemistry seems to be quite stable over time.

273: a lack of CO2 diel variability, but a finding of diel exchange variability, is a direct function of the model. This diel variability in fluxes then, is simply due to changes in wind which I do not believe are likely drivers of gas exchange in most river systems.

276: this section reads more like discussion than results

346: in contrast, this opening paragraph of the discussion most likely belongs in the results section of the manuscript

359: what part of the study design allows for a significant investigation on the role of

dams and gas exchange?

401: paragraph is too speculative

449: but the temperature variation was very small. How much could this have possibly contributed to the variation in CO2 exchange?

---

## Referee Comment (RC1) · Anonymous Referee #1 · 19 Jan 2018

General Comments:

This paper provides some important quantification of CO2 concentration, evasion rates, and temporal and spatial heterogeneity in an understudied Southeast Asian river system. Given the lack of data available on these systems, the concentration data presented in the paper is valuable on its own. I am concerned, however, at the large discrepancy between calculated and measured CO2 given the lack of a reasonable explanation aside from calculation error. Furthermore, along with what the other reviewer wrote, the reliance on wind-speed as the only determinant factor for k600 is subject to large errors in flux estimation. Knowing this, it is nearly impossible to assess how well correlated CO2 flux is with any of the environmental parameters used in the multi-variate analysis at the end of the paper. I think if the authors figured out why

their calculated values are off, used a more broadly accepted model to estimate k600 (or better yet, measured it directly), and simply presented the concentration and flux measurements from the Red River, it would be a valuable contribution to the literature.

Specific Comments:

-How exactly does the data presented in this manuscript relate to anthropogenic impacts? The authors suggest that the Red River is "strongly" affected by human activities but do not provide results that suggest humans have altered the amount or way by which CO2 is evaded from the river. The "influence of dams" based simply on observing higher concentrations at that site is unconvincing – there is no direct evidence. Likewise, the ascribed influence of human population is equally weak

-There are more accurate ways to calculate k600 than from wind speed. As the other commenter suggested, instantaneous discharge, flow hydraulics, and even channel slope may provide better or more robust ways to model k600 compared to wind speed

-Although the authors suggest that their direct and calculated pCO2 values were well correlated, they do not seem to correspond very well at all. The y-axis in Figure 3 makes it impossible to appreciate the noise in this relationship. More importantly, there is no plausible explanation as to why the slope is not close to 1, but rather that calculated pCO2 was nearly 1/5 of that measured directly by their equilibrator

-The explanation that wind speed is driving the diel signature in CO2 flux is circular. Wind speed is how the flux was calculated in the first place, so higher winds during the day will always yield a "higher flux" of CO2 during the day. CO2 might have a completely different diel pattern if modelled with something other than wind speed

-No specific reasoning given as to why temperature was positively correlated with pCO2 concentration except enhanced weathering rates. Did the authors consider in-stream respiration?

-It's hard to evaluate the seasonal variation of CO2 fluxes generated from only wind

speed and concentration

-What are the +/- values in any of the tables? Standard deviation?

-There seems to be overall very little diel signature in any of the water chemistry data presented in the paper. All of the differences seem to fit within the error bars of each "average" measurement.

Technical Corrections:

17: I would avoid the use of the word "good" when describing a river system

19: Not sure what is meant by "carbon dynamic"

21-22: Do you mean "relative" rather than "in contrast"?

42-43: Unclear sentence summarizing Raymond 2013

113: How was alkalinity measured? There are no methods detailing this

277: Discussion mixed in with results, difficult to follow

286: More discussion in the results section

346: Results being presented in the "Discussion"

359: Results being presented in the Discussion

---

## Referee Comment (RC2) · Anonymous Referee #2 · 9 Feb 2018

General comments:

The authors reported new data on CO2 partial pressure and CO2 evasion from the lower Red River in Vietnam. This paper also provides useful water chemistry data of the river system. Considering that river systems in the Southeast Asia are under-represented in the global budget of riverine carbon fluxes despite their large river discharge and carbon loads, this study could provide valuable datasets. However, the paper can be improved further by (1) strengthening the estimates of CO2 evasions, (2) providing detailed discussion on the observed patterns, and (3) reorganizing the paragraphs (e.g. some paragraphs in results fit to discussion, and vice versa). I would also suggest that the paper receive a thorough editing for grammar and clarity by the authors. Specific comments are below, which the authors can consider when revising

[Figure]

the manuscript.

Specific comments:

Lines 43–47: These are confusing because the first one (2.7 Pg C yr-1) includes riverine carbon transport, mineralization, and deposition, while the second one ("a lower value" is for CO2 evasion only. I don't think the estimate of CO2 evasion (2.1 Pg C yr-1) from inland waters by Raymond et al. (2013) is a lower value than the previous estimate. Lauerwald et al. (2015) provided a lower estimate, though.

Lines 125–126: Alkalinity measurement is critical for the calculation of pCO2 and CO2 evasion. Please provide more detailed information on how the alkalinity was measured and double checked. Was there a difference in alkalinity between filtered and unfiltered samples? It seems the turbidity can go up sometimes and I wonder how this could influence the alkalinity of filtered water.

Lines 138–: Have you measured CO2 flux directly from the surface of the water and compared that with the calculated values? (e.g. Duc et al., 2013, Environmental Science & Technology, 47, 968-975)

Lines 143–: I agree with the other reviewers that using wind speed as a component in the equation may not appropriate. Detailed explanation is needed on why the equation was chosen.

Lines 256–269: Detailed explanation is needed in "discussion" on why there is such a large difference in the measured and calculated pCO2.

Lines 299–312: This paragraph would belong to discussion rather than results.

Discussion: The results and discussion are mixed.

Lines 400: Are the differences statistically significant?

Lines 411–422: This sentence is too long. Extracting only essential information would be better than this long sentence.

Tables: Is the "+/-" standard error or standard deviation? Please clarify it.

Figures: Is the error bar standard error or standard deviation? Please clarify it.

Figure captions needs to more detailed description of the figures including explanations on legends.

---

## Referee Comment (RC3) · Anonymous Referee #3 · 13 Feb 2018

General Comments:

The manuscript entitled "CO2 partial pressure and CO2 emissions from the lower Red River (Vietnam)" by Le et al provides new and important data on CO2 in the Lower Red River. The results are a good contribution for the understanding of the role of Asian rivers for the global carbon evasion from freshwater ecosystems. However, there are many technical problems that need to be address to increase the quality of the paper. I agree with the other reviewers that the estimate of the CO2 evasion is the main problem in the paper and must be reworked. The figures and tables are clear and well designed. However, I suggested some changes in the number of figures and tables. In summary, the authors should review the flux calculation (see specific comments below); consider k600 calculations based on turbulence generated by water flow and;

reorganize Results and Discussion sections. Specific comments and suggestions are below.

Specific comments:

66 – This sentence suggests that there is a "limited knowledge" about carbon flux. Does it mean that there are published information about carbon flux in the Red River, or not?

79 – In the figure 1, the Ba Lat station does not seem to be located about 50 km from the sea. Please clarify.

Some references in the introduction are missing (e.g. lines 48, 56)

86 – I suggest removing the sentence "see below for the detail river discharge in 2014" and insert the correct reference - figure, table etc.

109 – The sentence "All solutions used for. . ." is unnecessary.

151 – What is the "regulator"? You mean pump or a flux regulator? Please, clarify.

162 – By convention k600 is the gas transfer velocity normalized to a Schmidt number of 600. And k is the gas transfer velocity. "Sc" should not be the "Schmidt number, normalized to Schmidt number of 600". "Sc" should be the Schmidt number of a given gas ($CO_2$ in this case) at a given temperature. Please, clarify the calculation and check if the calculation is correct.

166 – I wonder why the authors did not considered the temperature variation measured by HOBO sensor to calculate "a"? Were temperature records not made at the same time? In addition, conductivity is much higher in Ba Lat station suggesting differences in salinity among stations. If "a" is a constant (line 167), the salinity variation seems not to be considered in the calculation.

170 – Please, check the units of each element of the equation. The flux was calculated in $\mu$mol m-2 s-1 but the k, a and pCO2 air is cm h-1, mol L-1 atm-1 and ppm,
respectively. Please reorganize the section 2.4.1 for clarification.

183 – If k600 values are closely related to flow velocity and channel gradient, why author used k600 values calculated from wind speed? The authors should present a convincing explanation or review the calculation and values.

220 – The same results are presented in Table 2 and Figure 2. Please, remove the table or the figure.

223 – I suggest add Alkalinity values in the table or figure 2.

257 – Change "monsoon" to "wet".

263-269 – Abril et al., 2015 discuss overestimation of pCO2 using total alkalinity in acidic, organic-rich freshwaters. However, if this water condition is not the case of Red River, the find of Abril is not an explanation for the opposite (lower calculate values when compared with the measured values). These values should be similar or, please, provide a clear and convincing explanation.

276 – Discussion inside the Results section. Results and Discussion are mixed in several parts of these sections. I suggest re-write and reorganize these sections.

Table 1 – Please double check the station location. I plot the station locations and there are some inconsistences (e.g. Ba Lat is located in the ocean and not in the Red River).

Table 2 and Figure 2 – Why the values after +/- in table 2 and the error bar in Figure 2 are different? What does they mean?

Figure 6 – What the gray and dark circles mean. Dry and wet season? Insert these information in the figure caption.

[Figure]

---

## Referee Comment (RC4) · Anonymous Referee #4 · 14 Feb 2018

GENERAL COMMENTS

Le et al. report a valuable and potentially interesting data-set of CO2 data measured with an equilibrator at several stations of the Red River. This is a valuable data-set as CO2 data directly measured are lacking worldwide, and in particular in sub-tropical and tropical environments.

However, the paper suffers from a poor writing (English phrasing and syntax) that absolutely needs to be improved.

Also, the presentation and discussion of the results are extremely convoluted. The authors make a list of numerous possible hypothesis but do not really provide a convincing interpretation of the data (a clear and solid "story-line").

[Figure]

Some of the comparisons in the discussion are out of scope and irrelevant such as comparisons with mangroves, beaver ponds, sea-ice and the Southern Ocean.

MAJOR COMMENTS

The discussion is extremely convoluted and goes in all sorts of directions but nothing really conclusive comes out of it. By looking at the pCO2 plot in Figure 2, we can conclude:

- Differences between night and day are not statistically different whatever the site. The authors should try to explain this by comparing with other *river* sites where night-day pCO2 differences have been reported. The low phytoplankton biomass as indicated by chlorophyll-a content and the low differences in night-day temperature that induce low variations in CO2 solubility probably explain the low daily variations in pCO2. Anyway the low daily variations of pCO2 are an interesting aspect of the paper that deserves a longer discussion. Possibly make a table with studies that have shown daily variations of pCO2 with other variables such as average Chlorophyll-a content or POC, daily changes in temperature, . . .

- Overall the spatial gradients (differences among sites) are very low, except for Hoa Binh. Probably that all sites are located very close and correspond to similar sized rivers with similar catchment characteristics (land cover + lithology) so that the spatial differences are low. An explanation for the higher values of Hoa Binh is needed. Is it because it's downstream of a dam ? This would make sense based on existing literature (Sinnamary river, cf paper by Guérin et al.). Maybe it's mentioned somewhere in the text, but I missed the information being distracted by all of the other marginal bits and pieces of discussion.

- The seasonal variations are comparatively very small (except for Hoa Binh). This is an intriguing result that deserves being explored. The authors should compile in a table studies that report seasonal cycles of pCO2 (report the min-max of pCO2) and other relevant variables such as ratio of max/min of discharge, seasonal changes (min-

max) of POC and DOC. It might be useful to show a plot with the full seasonal cycle of freshwater discharge and indicate the two sampling periods. This would allow readers to situate the samplings on the hydrograph.

- The fact that the city of Hanoi does not seem to influence markedly the O2 and pCO2 levels is also intriguing and deserves some discussion. Was the station located within the city itself or slightly downstream ? Could this be due to the fact that freshwater discharge is relatively important (2000-3000 m3/s ?). A tabular comparison of studies that have shown the influence of cities on river CO2 might be useful.

In rivers, the main driver of the gas transfer velocity is turbulence generated by flow. I suggest that the authors use flow velocity data (that are available at gauging stations) and use the equations of Raymond et al. (2012) to compute the gas transfer velocity and fluxes (equation N°5 of Table 2 of Raymond et al. (2012) is recommended).

For all of the spatial and seasonal comparisons, some firm statistical testing is required (t-test, ANOVA, ...).

SPECIFIC COMMENTS

L29-30: could also be due direct inputs of CO2 from soils or wetlands.

L48-51: this is a very complicated way to state that C fluxes from rivers depend on lithology and land cover.

L59: improve instead of "precise"

L75: Please justify the choice of the 5 stations.

Also, it needs to be clearly explained that one of the stations is influenced by seawater intrusion. Are the other stations affected by tidal wave propagation ?

L94: provide information on land cover, lithology, and other relevant catchment characteristics for the studied rivers.

L107: replace "sensor" by CTD

L110: "all data must be entered on the documents" ? What does this mean ?

L113: total "alkalinity"

L117-126: provide precision and accuracy of all the measured variables.

Everywhere in the ms (text and tables) pCO2 values should not be given at the tenth of ppm, given that with an equilibrator precision of pCO2 measurements is typically of +/-1 ppm and accuracy of pCO2 measurements is typically of +/- 3-5 ppm (at best).

L125: what was the volume of sample water ? What acid and concentration was used ? How was the end-point determined ? What titrator was used ? The authors discuss the differences of pCO2 measured directly and calculated from pH and total alkalinity (TA). However, for this discussion to be meaningful it is necessary to have an idea of the quality of the pH and TA measurements, and this is only possible if analytical techniques are described in detail. This is lacking for TA here, and elsewhere the authors need to provide information on the pH electrode calibration. Type of buffers, frequency of calibration, etc. . .

TA data need to be expressed in $\mu$mol/L (as by convention). This is the unit of the TA as input variable for the CO2sys program. So why express the data in mg/L ? Further the data expressed per mass (mg/L) instead of per moles are extremely confusing. Is it mg of C ? Or mg of H+ ? (based on the conventional definition of TA as the quantity (number of moles) of protons to titrate bases in one kilogram (or L) of water). Should it be mg of C, then the TA for the first line of Table 2 (105 mg/L) would be 8750 $\mu$mol/L. In this case the computation of pCO2 gives 3670 ppm for a pH of 8.2 and 26.4°C (and not 270 ppm as stated). Should it be mg of CaCO3, then the TA for the first line of Table 2 (105 mg/L) would be 1050 $\mu$mol/L. In this case the computation of pCO2 gives 440 ppm for a pH of 8.2 and 26.4°C (and not 270 ppm as stated). Should it be mg of CO3, then the TA for the first line of Table 2 (105 mg/L) would be 1750 $\mu$mol/L. In this case

the computation of pCO2 gives 733 ppm for a pH of 8.2 and 26.4°C (and not 270 ppm as stated).

L140: Abbreviation IRGA not defined.

L144: replace "balanced" by equilibrated.

L156: This equation was established in the 1970's well before the paper of Raymond and Cole.

L160: replace "function" by parameterization.

L167: solubility changes with temperature, why did you state a constant value ?

L174-190: I suggest that this section is removed, as this topic has been discussed at length in several dedicated papers.

L195: The speciation of DIC was established decades before the papers of Cai et al. 2008 and Sun et al. 2010). The paper of Park (1969) formalised the equations to compute the speciation from all of the possible combinations.

L203: The calculated pCO2 is about 5 times lower than the pCO2 measured directly, this can hardly be considered as 'similar results' as stated.

L230-235: Present and discuss either salinity or conductivity but not both as they provide the same information.

L233: Statistical test needed.

L245-251: Statistical test needed.

L260: The "good" correlation is of marginal interest what's relevant is that the pCO2 values from the two methods differ by a factor of 5

L261-269: This result is extremely intriguing because the sampled rivers are within the "acceptable" range of applicability of the computation of pCO2 from pH and TA, with low DOC and high pH values. The under-estimation of the computed pCO2 could be due

to a bias in the measurements of pH and TA, which is not possible to evaluate given the lack of information in the material and methods. Anyway, the under-estimation of TA and/or over-estimation of pH could explain why the computed pCO2 is very low compared to directly measured pCO2.

L271-298: This discussion is of marginal interest because it depends on the way the gas transfer velocity was computed from a parameterisation as function of wind speed. pCO2 did not change markedly from night to day, but wind speed was higher during the day than the night. This is somewhat trivial, and if the authors re-compute the fluxes with a parameterization as function of flow, this day-night difference will be erased.

L298: Regarding the Ho et al. (2016) study, a note of caution is needed. This study used wind speeds from a sonic anemometer above the mangrove forest (obviously higher wind speeds that the level of the river below the canopy) and located 4 km from the coastline while the tracer injection point was located 12 km away from the coastline. For both these reasons, wind speed data used to build the Ho et al. (2016) relationship is over-estimated, meaning that the relationship itself is not reliable, and the role of currents in generating turbulence (and driving the gas transfer velocity) probably under-estimated.

L299-312: In this discussion there's a mix of studies in lakes and mangroves, which makes little sense when discussing CO2 dynamics in rivers.

L302: Roulet et al (1997) report on beaver ponds. This is a very specific environment that is not very relevant for comparison with Red river.

L321: references are needed to back this statement.

L323: or (in addition) inputs of CO2 from wetlands.

L335: Chanda study deals with mangroves and Takahashi with oceans. This is irrelevant (and out of scope) to discuss CO2 dynamics in rivers.

L338: Comparison with Southern Ocean is irrelevant (and out of scope).

L346: Why did you not include freshwater discharge in the PCA and the correlation matrix analysis ?

L359: Statistical test needed.

L397: The cited references deal with Artic Ocean and sea-ice, not with Pearl River.

L401: Statistical test needed.

L407-408: The city of Ha Noi does seem to influence DOC and POC but there seems to be little effect on O2 and pCO2. This is intriguing.

L410: some of the cited "rivers" are in fact mangrove brackish systems (irrelevant and out of scope).

L426-429: the authors provide no evidence to back up this hypothesis that is mainly speculation.

L427: Elsewhere in the discussion the authors say that the Hoa Binh site has high pCO2 because downstream of a dam, and here the authors say the low pCO2 values in the Red River is also due dams. This cannot work both ways.

Should the authors want to explore why pCO2 values are lower in the Red River than other rivers worldwide comparing levels of DOC and POC with other rivers might be useful. Also, some studies have provided hypothesis to explain large-scale variations of pCO2 across river catchments such as productivity, population density, temperature, . . . e.g. the Lauerwald study that is cited by the authors.

L434-451: This section provides a summary of the paper (and duplicates the abstract) and does not provide a real conclusion.

L565: typo in "Costs"

L598: Nathalie is the first name not the family name.

L616: Patricia is the first name not the family name.

L618 and L650: This reference appears twice.

Table 3: There's no need to provide the air-water CO2 fluxes derived from the pCO2 calculated from pH and TA. Since calculated pCO2 is under-estimated compared to directly measured pCO2, this is also the case for the fluxes, in a very predictable and trivial way.

In tables and figures is given in the turbidity of the probe in NTU, when in fact authors measured TSM. Why not show also (or instead) the TSM data ?

Also, it might be useful to show the %POC data instead (or in addition to) POC, since POC follows closely TSM, but %POC gives some info on the origin of POC.

---

## Author Comment (AC1) · 30 Apr 2018

Dear Dr J. Crawford

Thank you very much for your comments. We revised the ms in taking into account all you comments and suggestions Below we try to reply and explain our revision following to your questions and comments Best regards, On behalf of co-authors Thi Phuong Quynh LE

J. Crawford jtcrawford@usgs.gov

This paper documents the chemical conditions and concentrations of dissolved carbon dioxide in the Red River system of Vietnam. The data contribute to the "database" of concentration values for the globe, with one goal of further constraining the CO2 source

strength of inland waters. Therefore, the data are valuable on their own, especially given that they fill in geographic gaps for SE Asia. The main criticism of this paper is the use of a wind-driven gas exchange model. While criticism of gas exchange models are prevalent within the community of researchers, this example is especially problematic as it relies on unlikely drivers of turbulence (and thus gas exchange) in riverine systems. There is some evidence that gas exchange is enhanced by some wind patterns in very large rivers, however, gas exchange in rivers is not considered to be a major driver. Rather, it is turbulence generated by water flow that drives gas exchange rates in these systems. Therefore, the $CO_2$ emission estimates are not only biased, as recognized by the authors, but are likely to be highly inaccurate due to the model selected. It is hard to believe the results without some other line of confirmation. In addition to the criticism of the estimates of gas exchange, I did not find the discussion points to be well supported by the data especially given the limited time and geographic scope of the measurements. There is simply not enough evidence to support any of the inferred drivers of $CO_2$ variability in this river system. Thank you for the comments. We revised the manuscript in taking into account all comments from reviewers.

Specific Comments: 48: what references support plate tectonics as major drivers of carbon fluxes in this system? We added the information in page 3 "The Delta is located in a very flat and low land, with an elevation ranging from 0.4 to 12 m above sea level (Nguyen Ngoc Sinh et al., 1995). Previous studies showed the difference of lithology in the three upstream tributaries: Paleozoic sedimentary rocks (55.5

53: are changes in sediments the hypothesized drivers of changing carbon fluxes in this study? Yes. The information concerning changing of suspended solids of the Red River was added in page 12, in the section "Influence of dams on $pCO_2$ and $CO_2$ emission " Noted that the Hoa Binh site is situated downstream a series of reservoirs, which have been constructed in both Chinese and Vietnamese parts including two large dams Hoa Binh (in 1989) and Son La (in 2010). The Vu Quang site is located in the downstreamof a series of reservoirs, including two important Thac Ba (in 1970) and

Tuyen Quang (in 2010). Previous studies emphasized that these dams have impacted water and sediment discharges downstream (Ha and Vu 2012; Ngo et al. 2014; Lu et al. 2015) with significant sediment deposition being observed in the reservoirs (Dang et al. 2010; Vinh et al. 2014; Lu et al. 2015).

184: if exchange is less related to wind, then what is the justification for using this model in the present study? Thank you for the suggestion. But we now revise the ms by recalculating k600 proposed by Raymond et al 2012 K600 was calculated as presented in the section "2.5 CO2 fluxes determination" , page 5-6: "In this study, k600 was calculated using the equation from Raymond et al. (2012) based on stream velocity (V, in m s -1 ), slope (S, unitless), depth (D, in meters) and discharge (Q, in m3 s-1), as follow: k600 = 4725 ± 445 x (V x S) 0.86 ± 0.016 x Q-0.14 ± 0.012 x D 0.66 ± 0.029 Eq. (2)

214: such low temperature variability leads to skepticism of this environmental paramenter being a significant driver of CO2 variability. In addition, the broad conclusion here is that water chemistry seems to be quite stable over time. Thank you for the suggestion. The test by ANOVA and t-test results showed that no clear day-night variation but clear seasonal (dry-wet) variation of temperature was found at 5 sites. Other variables including pCO2, organic matters . . . showed seasonal variation. So, we revised the section of temporal variation of pCO2 and CO2 flux in page 9 -10

273: a lack of CO2 diel variability, but a finding of diel exchange variability, is a direct function of the model. This diel variability in fluxes then, is simply due to changes in wind which I do not believe are likely drivers of gas exchange in most river systems. Again, we recalculated the k600 by Raymond et al (2012), that lead to the change in CO2 flux. We re-write the discussion.

276: this section reads more like discussion than results We removed this section in the revised ms.

346: in contrast, this opening paragraph of the discussion most likely belongs in the results section of the manuscript Thank you for the suggestion. In the revised ms, this paragraph is in the section Results/ 3.4. Relations between pCO2 and water chemistry variables in page 8 -9.

359: what part of the study design allows for a significant investigation on the role of dams and gas exchange? We added the paragraph for describing the 5 sites observed in page 3 "Five stations were studied along the lower Red River (Vietnam): Yen Bai station (at the outlet of the Thao River); Hoa Binh station (after Son La and Hoa Binh reservoirs, at the outlet of the Da River); Vu Quang (at the outlet of the Lo River); Hanoi and Ba Lat stations (in the main course of the Red River downstream). The three stations Yen Bai, Vu Quang and Hoa Binh are representative for water quality of the three main tributaries (Thao, Da and Lo) of the upstream Red River, whereas the Hanoi station is representative for the main course Red River after confluence of three main tributaries. Only the Ba Lat station, which is located at the Red River mouth (about 13 km from the sea) is influenced by seawater intrusion (Fig 1). A more detailed description of the river characteristics of the Thao, Da, Lo and the main branch of the Red River can be found in (Le et al., 2007)"

401: paragraph is too speculative The paragraph was revised in page 13 (line 451)

449: but the temperature variation was very small. How much could this have possibly contributed to the variation in CO2 exchange? Thank you for the suggestion. As we mention above, we recalculate the k600 and fCO2. So the results now were represented and synthetized. The discussion concerning day-night variations was rewritten. pCO2 differences between night and day were really low, most probably because of low temperature difference and low photosynthetic activity due to the turbidity of the Red River. The conclusion was also revised.

Please also note the supplement to this comment:
https://www.biogeosciences-discuss.net/bg-2017-505/bg-2017-505-AC1-supplement.pdf

[Figure]

**Supplementary Material**

**$CO_2$ partial pressure and $CO_2$ emission along the lower Red River (Vietnam)**

Thi Phuong Quynh Le[1*], Cyril Marchand[2,3], Cuong Tu Ho[4], Nhu Da Le[1], Thi Thuy Duong[4], Trung Kien Nguyen[4], XiXi Lu[5], Phuong Kieu Doan[1], Thi Mai Huong Nguyen[1] and Duy An Vu[1]

[1]: Institute of Natural Product Chemistry, Vietnam Academy of Science and Technology, 18 Hoang Quoc Viet Road, Cau Giay, Hanoi, Vietnam.
[2]: IMPMC, Institut de Recherche pour le Développement (IRD), UPMC, CNRS, MNHN, Noumea, New Caledonia, France.
[3]: Faculty of Chemistry, University of Science – VNUHCM, 225 Nguyen Van Cu, Ho Chi Minh City, Vietnam
[4]: Institute of Environmental Technology, Vietnam Academy of Science and Technology, 18 Hoang Quoc Viet Road, Cau Giay, Hanoi, Vietnam.
[5]: Department of Geography, National University of Singapore, Arts Link 1, Singapore 117570, Singapore.

*Correspondence to*: Thi Phuong Quynh Le  (quynhltp@yahoo.com   or    quynhltp@gmail.com)

[Figure]

**Figure SM1**: Daily variation of river discharge at the outlet of the Thao (Yen Bai), Da (Hoa Binh), Lo

**Supplement:**

[revised manuscript text omitted]

---

## Author Comment (AC2) · 30 Apr 2018

Dear anonymous reviewer,

Thank you very much for your kind suggestions and comments for ameliorating our manuscript.

We revised the ms in taking into account all comments and suggestions

Best regards,

On behalf of all co-authors

Thi Phuong Quynh LE

[Figure]

REVIEWER COMMENTS FOR THE AUTHOR Interactive comment on "CO2 partial pressure and CO2 emissions from the lower Red River (Vietnam)" by Thi Phuong Quynh Le et al. Anonymous Referee 1

General Comments: This paper provides some important quantification of CO2 concentration, evasion rates, and temporal and spatial heterogeneity in an understudied Southeast Asian river system. Given the lack of data available on these systems, the concentration data presented in the paper is valuable on its own. I am concerned, however, at the large discrepancy between calculated and measured CO2 given the lack of a reasonable explanation aside from calculation error. Furthermore, along with what the other reviewer wrote, the reliance on wind-speed as the only determinant factor for k600 is subject to large errors in flux estimation. Knowing this, it is nearly impossible to assess how well correlated CO2 flux is with any of the environmental parameters used in the multi-variate analysis at the end of the paper. I think if the authors figured out why their calculated values are off, used a more broadly accepted model to estimate k600 (or better yet, measured it directly), and simply presented the concentration and flux measurements from the Red River, it would be a valuable contribution to the literature. Thank you very much for the helpful comments. We revised the paper taking into account your comments concerning k600. k600 now is calculated from the formula proposed by Raymond et al (2012), based on different variables such as river discharge, water velocity, slope. The values of k600 are now considered more realistic and then CO2 flux evasion was recalculated.

Specific Comments: -How exactly does the data presented in this manuscript relate to anthropogenic impacts? The authors suggest that the Red River is "strongly" affected by human activities but do not provide results that suggest humans have altered the amount or way by which CO2 is evaded from the river. The "influence of dams" based simply on observing higher concentrations at that site is unconvincing – there is no direct evidence. Likewise, the ascribed influence of human population is equally weak. Thank you for the comments. We revised the discussion about the human impact on

pCO2 and fCO2 of the Red river in the section "4.2 Spatial variations of pCO2 and fCO2 outgassing in pages 11- 12. In this section, separated factors such as dam impoundment, population density and land-use were discussed.

- There are more accurate ways to calculate k600 than from wind speed. As the other commenter suggested, instantaneous discharge, flow hydraulics, and even channel slope may provide better or more robust ways to model k600 compared to wind speed. The explanation that wind speed is driving the diel signature in CO2 flux is circular. Wind speed is how the flux was calculated in the first place, so higher winds during the day will always yield a "higher flux" of CO2 during the day. CO2 might have a completely different diel pattern if modelled with something other than wind speed. It's hard to evaluate the seasonal variation of CO2 fluxes generated from only wind speed and concentration Thank you very much for the helpful comments. As mentioned above, k600 value was determined by the method proposed by Raymond et al (2012), then the CO2 outgassing fluxes were recalculated and discussed for their seasonal and spatial variations. K600 was calculated as presented in the section "2.5 CO2 fluxes determination" , page 5-6: "In this study, k600 was calculated using the equation from Raymond et al. (2012) based on stream velocity (V, in m s $^{-1}$ ), slope (S, unitless), depth (D, in meters) and discharge (Q, in m3 s-1), as follow: k600 = 4725 $\pm$ 445 x (V x S) 0.86 $\pm$ 0.016 x Q-0.14 $\pm$ 0.012 x D 0.66 $\pm$ 0.029 Eq. (2)

- Although the authors suggest that their direct and calculated pCO2 values were well correlated, they do not seem to correspond very well at all. The y-axis in Figure 3 makes it impossible to appreciate the noise in this relationship. More importantly, there is no plausible explanation as to why the slope is not close to 1, but rather that calculated pCO2 was nearly 1/5 of that measured directly by their equilibrator. We apologize for the errors of pCO2 calculation values in the previous version of the manuscript. Values were corrected in the revised manuscript (see table 2). Figure 2 (fig 3 old) was also corrected. A difference between pCO2 calculated and measured was found and discussed in the revised manuscript page 7.

- No specific reasoning given as to why temperature was positively correlated with pCO2 concentration except enhanced weathering rates. Did the authors consider in-stream respiration? This is possibly related to enhanced decomposition rates of organic matter.

-What are the +/- values in any of the tables? Standard deviation? They are standard deviation. We added the information in the table captions.

- There seems to be overall very little diel signature in any of the water chemistry data presented in the paper. All of the differences seem to fit within the error bars of each "average" measurement. Yes, we fully agree with the reviewer, and the discussion concerning day-night variations (first paragraph of the Discussion) was rewritten. pCO2 differences between night and day were really low, most probably because of low temperature difference and low photosynthetic activity due to the turbidity of the Red River.

Technical Corrections: 17: I would avoid the use of the word "good" when describing a river system The word "good" was replaced by "representative"

19: Not sure what is meant by "carbon dynamic" The sentence was revised "This study aims to quantify the spatial and seasonal variability of CO2 partial pressure and CO2 emission of the lower Red River system"

21-22: Do you mean "relative" rather than "in contrast"? "Relative" was replaced as suggested

42-43: Unclear sentence summarizing Raymond 2013 The sentence was revised "Raymond et al. (2013) estimated a global evasion rate of 2.1 Pg C yr-1 from inland waters, and that global hot spots in stream and rivers which occupy only 20

113: How was alkalinity measured? There are no methods detailing this The method for alkalinity determination was added in the revised manuscript (see page 4)

277: Discussion mixed in with results, difficult to follow Results and discussion are now better separated, as suggested.

286: More discussion in the results section This was separated between results and discussion as suggested

346: Results being presented in the "Discussion" This was separated between results and discussion

359: Results being presented in the Discussion Results were added in the "Discussion" in page 12.

Please also note the supplement to this comment:
https://www.biogeosciences-discuss.net/bg-2017-505/bg-2017-505-AC2-supplement.pdf

—————————————————————

Supplementary Material

**$CO_2$ partial pressure and $CO_2$ emission along the lower Red River (Vietnam)**

Thi Phuong Quynh Le[1*], Cyril Marchand[2,3], Cuong Tu Ho[4], Nhu Da Le[1], Thi Thuy Duong[4], Trung Kien Nguyen[4], XiXi Lu[5], Phuong Kieu Doan[1], Thi Mai Huong Nguyen[1] and Duy An Vu[1]

[1]: Institute of Natural Product Chemistry, Vietnam Academy of Science and Technology, 18 Hoang Quoc Viet Road, Cau Giay, Hanoi, Vietnam.
[2]: IMPMC, Institut de Recherche pour le Développement (IRD), UPMC, CNRS, MNHN, Noumea, New Caledonia, France.
[3]: Faculty of Chemistry, University of Science – VNUHCM, 225 Nguyen Van Cu, Ho Chi Minh City, Vietnam
[4]: Institute of Environmental Technology, Vietnam Academy of Science and Technology, 18 Hoang Quoc Viet Road, Cau Giay, Hanoi, Vietnam.
[5]: Department of Geography, National University of Singapore, Arts Link 1, Singapore 117570, Singapore.

*Correspondence to*: Thi Phuong Quynh Le (quynhltp@yahoo.com or quynhltp@gmail.com)

**Figure SM1**: Daily variation of river discharge at the outlet of the Thao (Yen Bai), Da (Hoa Binh), Lo (Vu Quang) rivers and in the main branch of the Red River at Hanoi and Ba Lat stations in 2014.

**Fig. 1.**

**Supplement:**

[revised manuscript text omitted]

---

## Author Comment (AC3) · 30 Apr 2018

Dear anonymous reviewer,

Thank you very much for your kind suggestions and comments for ameliorating our manuscript.

We revised the ms in taking into account all comments and suggestions

Best regards,

On behalf of all co-authors

Thi Phuong Quynh LE

[Figure]

Anonymous Referee 2 General comments: The authors reported new data on CO2 partial pressure and CO2 evasion from the lower Red River in Vietnam. This paper also provides useful water chemistry data of the river system. Considering that river systems in the Southeast Asia are underrepresented in the global budget of riverine carbon fluxes despite their large river discharge and carbon loads, this study could provide valuable datasets. However, the paper can be improved further by (1) strengthening the estimates of CO2 evasions, (2) providing detailed discussion on the observed patterns, and (3) reorganizing the paragraphs (e.g. some paragraphs in results fit to discussion, and vice versa). I would also suggest that the paper receive a thorough editing for grammar and clarity by the authors. Specific comments are below, which the authors can consider when revising the manuscript. Thank you very much for the comments. We revised the manuscript and checked the grammar as suggested.

Specific comments: Lines 43–47: These are confusing because the first one (2.7 Pg C yr-1) includes riverine carbon transport, mineralization, and deposition, while the second one ("a lower value" is for CO2 evasion only. I don't think the estimate of CO2 evasion (2.1 Pg C yr-1) from inland waters by Raymond et al. (2013) is a lower value than the previous estimate. Lauerwald et al. (2015) provided a lower estimate, though. The paragraph was revised "Natural hydrological processes and biogeochemistry of many rivers in the world have suffered from the influences of climate change and human activities in their drainage basins. Riverine carbon fluxes and outgassing are important parts of the carbon exchange among terrestrial, oceanic and atmospheric environment. Rivers and streams not only transfer various forms of carbon (dissolved and particulate) to oceans, but also evade a significant amount of carbon to the atmosphere (Battin et al., 2009; Richey et al., 2002). Due to CO2 evasion, the flux of carbon that leaves the terrestrial biosphere through global fluvial network was suggested to be twice larger than the amount that ultimately reaches the coastal ocean (Bauer et al., 2013; Regnier et al., 2013). Raymond et al. (2013) estimated a global evasion rate of 2.1 Pg C yr-1 from inland waters, and that global hot spots in stream and rivers which

occupy only 20

Lines 125–126: Alkalinity measurement is critical for the calculation of $pCO_2$ and $CO_2$ evasion. Please provide more detailed information on how the alkalinity was measured and double checked. Was there a difference in alkalinity between filtered and unfiltered samples? It seems the turbidity can go up sometimes and I wonder how this could influence the alkalinity of filtered water. The method for alkalinity determination was added in the revised manuscript in page 4. " Total alkalinity of the hourly samples was immediately determined on non-filtered water samples (30 ml water sample) in situ by titration method with 0.01M HCl (APHA, 1995). For each sample, triplicates were titrated and the analytical error was below 3

We also tried to test the difference in alkalinity between filtered and unfiltered samples at the Hanoi site in March 2018 and found that the alkalinity values were not so different in dry season when suspended solids was not so high .

Lines 138–: Have you measured $CO_2$ flux directly from the surface of the water and compared that with the calculated values? (e.g. Duc et al., 2013, Environmental Science Technology, 47, 968-975) Yes, we measured $CO_2$ flux directly from the surface of the water using a floating chamber method. We obtained an average $CO_2$ flux of 189 mmol.m-2.d-1 (with values ranging from 37.8 – 492.1 mmol.m-2.day-1). These values were lower than the values calculated from $pCO_2$ measured using equilibrator method and the equation provided by Frankignoulle et al. (2001) and Raymond and Cole (2011). This result was surprising since it is considered that when the water flow is higher than 0.2 m s-1, the presence of the chamber induces artificial turbulence that results in elevated $CO_2$ fluxes.

Lines 143–: I agree with the other reviewers that using wind speed as a component in the equation may not appropriate. Detailed explanation is needed on why the equation was chosen. As suggested by reviewers, we calculated k600 by other method proposed by Raymond et al (2012), basing on different variables such as river discharge,
water velocity, depth and slope. The values of k600 now are ameliorated and then CO2 flux evasion was recalculated. K600 was calculated as presented in the section "2.5 CO2 fluxes determination" , page 5-6: "In this study, k600 was calculated using the equation from Raymond et al. (2012) based on stream velocity (V, in m s -1 ), slope (S, unitless), depth (D, in meters) and discharge (Q, in m3 s-1), as follow: k600 = 4725 ± 445 x (V x S) 0.86 ± 0.016 x Q-0.14 ± 0.012 x D 0.66 ± 0.029 Eq. (2)

Lines 256–269: Detailed explanation is needed in "discussion" on why there is such a large difference in the measured and calculated pCO2. We apologize for the errors of pCO2 calculation. Values were corrected in the revised manuscript (see table 2). However, a difference between pCO2 calculated and measured was found. An explanation concerning this difference was added in the manuscript revised in page 7 -8. "3.3. Comparisons of the pCO2 results obtained by the two methods pCO2 along the lower Red River (Vietnam) in the dry and the wet seasons were determined by two methods: i) direct measurements using an equilibrator connected to an IRGA, ii) calculated from pH and alkalinity using the CO2-SYS$^{®}$ software. The direct pCO2 measurements gave slightly higher values than the calculated ones (Table 2), but the values of two methods were similar and presented the same trend of spatial and seasonal variations (R2 = 0.77, Fig. 2; Table 2). Lower values of the calculated pCO2 in this study may be caused by the analytical errors in pH or under-estimation of total alkalinity. Similarly, the CO2 outgassing rates which were calculated from measured pCO2 from equilibrator were higher than the ones derived from the calculated pCO2 from CO2-SYS, however they are in the same orders and have similar variation trends (Table 3, Fig 2)"

Lines 299–312: This paragraph would belong to discussion rather than results. Discussion: The results and discussion are mixed. We revised the paragraph

Lines 400: Are the differences statistically significant? Yes. The results from t-test showed that: - Seasonal (dry –wet) variation : difference appeared for all variables (p< 0.05) - Day-night variation: difference for pH (p< 0.05) but no clear difference for other variables

The results from t-test and ANOVA test showed that Spatial variation (5 sites) : temperature was not different within 5 sites. Other variables were significant different (p< 0.05). We added the difference statistics significant through the revised ms.

Lines 411–422: This sentence is too long. Extracting only essential information would be better than this long sentence. This sentence is now revised and moved to the section " 4.2 Spatial variations of pCO2 and fCO2 outgassing Âż in page 10 - 11

Tables: Is the "+/-" standard error or standard deviation? Please clarify it. It's standard deviation. We added the information in the table captions

Figures: Is the error bar standard error or standard deviation? Please clarify it. It is the standard deviation. We added the information in the figure captions

Figure captions needs to more detailed description of the figures including explanations on legends. Figure captions were revised as suggested, and thus more detailed.

Please also note the supplement to this comment:
https://www.biogeosciences-discuss.net/bg-2017-505/bg-2017-505-AC3-supplement.pdf

––––––––––––––––––––––––––

**Supplementary Material**

**CO$_2$ partial pressure and CO$_2$ emission along the lower Red River (Vietnam)**

Thi Phuong Quynh Le[1*], Cyril Marchand[2,3], Cuong Tu Ho[4], Nhu Da Le[1], Thi Thuy Duong[4], Trung Kien Nguyen[4], XiXi Lu[5], Phuong Kieu Doan[1], Thi Mai Huong Nguyen[1] and Duy An Vu[1]

[1]: Institute of Natural Product Chemistry, Vietnam Academy of Science and Technology, 18 Hoang Quoc Viet Road, Cau Giay, Hanoi, Vietnam.
[2]: IMPMC, Institut de Recherche pour le Développement (IRD), UPMC, CNRS, MNHN, Noumea, New Caledonia, France.
[3]: Faculty of Chemistry, University of Science – VNUHCM, 225 Nguyen Van Cu, Ho Chi Minh City, Vietnam
[4]: Institute of Environmental Technology, Vietnam Academy of Science and Technology, 18 Hoang Quoc Viet Road, Cau Giay, Hanoi, Vietnam.
[5]: Department of Geography, National University of Singapore, Arts Link 1, Singapore 117570, Singapore.

*Correspondence to*: Thi Phuong Quynh Le  (quynhltp@yahoo.com  or   quynhltp@gmail.com)

**Figure SM1**: Daily variation of river discharge at the outlet of the Thao (Yen Bai), Da (Hoa Binh), Lo

---

## Author Comment (AC4) · 30 Apr 2018

Anonymous Referee 3 General Comments: The manuscript entitled "CO2 partial pressure and CO2 emissions from the lower Red River (Vietnam)" by Le et al provides new and important data on CO2 in the Lower Red River. The results are a good contribution for the understanding of the role of Asian rivers for the global carbon evasion from freshwater ecosystems. However, there are many technical problems that need to be address to increase the quality of the paper. I agree with the other reviewers that the estimate of the CO2 evasion is the main problem in the paper and must be reworked. The figures and tables are clear and well designed. However, I suggested some changes in the number of figures and tables. In summary, the authors should review the flux calculation (see specific comments

below); consider k600 calculations based on turbulence generated by water flow and; reorganize Results and Discussion sections. Specific comments and suggestions are below. We thank you very much for the helpful comments and suggestions.

Specific comments: 66 – This sentence suggests that there is a "limited knowledge" about carbon flux. Does it mean that there are published information about carbon flux in the Red River, or not? Yes, there are some published information about carbon fluxes for a peri-urban area in the Red River Delta (Trinh et al., 2012) and (Nguyen et al., 2018). We revised the sentence and add the references: "However, there is a lack of data concerning $CO_2$ outgassing and carbon budget of the lower Red River (Trinh et al., 2012, Nguyen et al., 2018)."

79 – In the figure 1, the Ba Lat station does not seem to be located about 50 km from the sea. Please clarify. We apologize for the error. The Ba Lat site is 13 km from the sea. The information was corrected in the text page 3.

Some references in the introduction are missing (e.g. lines 48, 56) References were added in the revised manuscript

86 – I suggest removing the sentence "see below for the detail river discharge in 2014" and insert the correct reference - figure, table etc. The sentence was removed, the word "table 1" was replaced

109 – The sentence "All solutions used for. . ." is unnecessary. This sentence was deleted

151 – What is the "regulator"? You mean pump or a flux regulator? Please, clarify. It was changed into "a flow regulator".

162 – By convention k600 is the gas transfer velocity normalized to a Schmidt number of 600. And k is the gas transfer velocity. "Sc" should not be the "Schmidt number, normalized to Schmidt number of 600". "Sc" should be the Schmidt number of a given gas ($CO_2$ in this case) at a given temperature. Please, clarify the calculation and check

if the calculation is correct. We revised the paper by recalculating the k600 proposed
by (Raymond et al., 2012). Thus, Sc was removed from the new calculation. K600 was
calculated as presented in the section "2.5 CO2 fluxes determination" , page 5-6: "In
this study, k600 was calculated using the equation from Raymond et al. (2012) based
on stream velocity (V, in m s -1 ), slope (S, unitless), depth (D, in meters) and discharge
(Q, in m3 s-1), as follow: k600 = 4725 $\pm$ 445 x (V x S) 0.86 $\pm$ 0.016 x Q-0.14 $\pm$ 0.012
x D 0.66 $\pm$ 0.029 Eq. (2)

166 – I wonder why the authors did not considered the temperature variation measured
by HOBO sensor to calculate "a"? Were temperature records not made at the same
time? In addition, conductivity is much higher in Ba Lat station suggesting differences
in salinity among stations. If "a" is a constant (line 167), the salinity variation seems
not to be considered in the calculation. We calculated the fCO2 at 5 sited of the Red
River with different $\alpha$ values : $\alpha$ = 3.941 10-2 mol.L-1.atm-1 at 24 oC for dry season
and $\alpha$ = 3.138 10-2 mol.L-1.atm-1 at 27 oC for rainy season. The average flux of CO2
outgassing was 525.7 mmol.m-2.d-1. Then, this value was very closed with the one
calculated from the constant $\alpha$ = 3.4 10-2 mol.L-1.atm-1 we used for both rainy and
dry seasons of the Red River in this study, which resulted in the fCO2 average of the
whole Red River 530 mmol.m-2.d-1. An explanation was added in page 6 "ïĄą is the
solubility coefficient of CO2 for given temperature and salinity (Weiss, 1974) (mol L-1
atm-1). In this case, ïĄą = 0.034 mol L-1 atm-1. In this study, salinity variations were
low, except for the Ba Lat station. Temperature did not change a lot. We checked the
influence of different ïĄą values in the dry ($\alpha$ = 3.941 10-2 mol L-1 atm-1 at 24 oC) and
the wet season ($\alpha$ = 3.138 10-2 mol L-1 atm-1 at 27 oC) at the 5 sites and compared
with the constant ïĄą value of 0.034 mol L-1 atm-1. "

170 – Please, check the units of each element of the equation. The flux was calcu-
lated in $\mu$mol m-2 s-1 but the k, a and pCO2 air is cm h-1, mol L-1 atm-1 and ppm,
respectively. Please reorganize the section 2.4.1 for clarification. Thank you for the
comments. The flux is now calculated from different variables taking into account different units. We added the explanation for converting the final unit of flux calculation in mmol m-2 d-1 in the section 2.4 CO2 fluxes determination in page 5

183 – If k600 values are closely related to flow velocity and channel gradient, why author used k600 values calculated from wind speed? The authors should present a convincing explanation or review the calculation and values. Thank you for the comment. We now calculate the k600 by the formula proposed by Raymond et al (2012) which is based on river flow velocity, river discharge, depth, slope in pages 5 and 6: "k600, was gas transfer velocity of CO2 or piston velocity (cm h-1). Some studies indicate that k600 values are closely related to flow velocity and channel gradient for rivers (Alin et al., 2011). In this study, k600 was calculated using the equation from Raymond et al. (2012) based on stream velocity (V, in m s -1 ), slope (S, unitless), depth (D, in meters) and discharge (Q, in m3 s-1), as follow: k600 = 4725 ± 445 x (V x S) 0.86 ± 0.016 x Q-0.14 ± 0.012 x D 0.66 ± 0.029 Eq. (2)"

220 – The same results are presented in Table 2 and Figure 2. Please, remove the table or the figure. Figure 2 was removed as suggested.

223 – I suggest add Alkalinity values in the table or figure 2. Alkalinity values are presented in the table 2, in the column named "TAlk".

257 – Change "monsoon" to "wet". It was changed

263-269 – Abril et al., 2015 discuss overestimation of pCO2 using total alkalinity in acidic, organic-rich freshwaters. However, if this water condition is not the case of Red River, the find of Abril is not an explanation for the opposite (lower calculate values when compared with the measured values). These values should be similar or, please, provide a clear and convincing explanation. Thank you for the suggestion. We removed the sentence from the revised manuscript.

276 – Discussion inside the Results section. Results and Discussion are mixed in several parts of these sections. I suggest re-write and reorganize these sections. Thank

you for the suggestion. We re-writed and reorganized the Results and Discussion sections

Table 1 – Please double check the station location. I plot the station locations and there are some inconsistences (e.g. Ba Lat is located in the ocean and not in the Red River). There was a mistake for the location of the Ba Lat site. This was corrected in table 1 and in the text. Other sites were also checked.

Table 2 and Figure 2 – Why the values after +/- in table 2 and the error bar in Figure 2 are different? What does they mean? We checked the table 2. The figure 2 was removed in the revised manuscript.

Figure 6 – What the gray and dark circles mean. Dry and wet season? Insert these information in the figure caption.

The figure 4 demonstrates the Relationship between environmental variables and pCO2 at the five sites of the lower Red River in the wet (Sept) and the dry season (Nov) 2014. The figure and caption were revised.

Please also note the supplement to this comment:
https://www.biogeosciences-discuss.net/bg-2017-505/bg-2017-505-AC4-supplement.pdf

—————————————————

[Figure]

BGD
Supplementary Material

**CO$_2$ partial pressure and CO$_2$ emission along the lower Red River (Vietnam)**

Thi Phuong Quynh Le[1*], Cyril Marchand[2,3], Cuong Tu Ho[4], Nhu Da Le[1], Thi Thuy Duong[4], Trung Kien Nguyen[4], XiXi Lu[5], Phuong Kieu Doan[1], Thi Mai Huong Nguyen[1] and Duy An Vu[1]

[1]: Institute of Natural Product Chemistry, Vietnam Academy of Science and Technology, 18 Hoang Quoc Viet Road, Cau Giay, Hanoi, Vietnam.
[2]: IMPMC, Institut de Recherche pour le Développement (IRD), UPMC, CNRS, MNHN, Noumea, New Caledonia, France.
[3]: Faculty of Chemistry, University of Science – VNUHCM, 225 Nguyen Van Cu, Ho Chi Minh City, Vietnam
[4]: Institute of Environmental Technology, Vietnam Academy of Science and Technology, 18 Hoang Quoc Viet Road, Cau Giay, Hanoi, Vietnam.
[5]: Department of Geography, National University of Singapore, Arts Link 1, Singapore 117570, Singapore.

*Correspondence to*: Thi Phuong Quynh Le (quynhltp@yahoo.com or quynhltp@gmail.com)

**Figure SM1**: Daily variation of river discharge at the outlet of the Thao (Yen Bai), Da (Hoa Binh), Lo (Vu Quang) rivers and in the main branch of the Red River at Hanoi and Ba Lat stations in 2014.
**Fig. 1.**

---

## Author Comment (AC5) · 30 Apr 2018

Anonymous Referee 4 GENERAL COMMENTS Le et al. report a valuable and potentially interesting data-set of CO2 data measured with an equilibrator at several stations of the Red River. This is a valuable data-set as CO2 data directly measured are lacking worldwide, and in particular in sub-tropical and tropical environments. However, the paper suffers from a poor writing (English phrasing and syntax) that absolutely needs to be improved. Also, the presentation and discussion of the results are extremely convoluted. The authors make a list of numerous possible hypothesis but do not really provide a convincing interpretation of the data (a clear and solid "story-line"). Some of the comparisons in the discussion are out of scope and irrelevant such as comparisons with mangroves, beaver ponds,

sea-ice and the Southern Ocean.

Thank you for the comments and suggestions. We revised the manuscript in taking into account all comments/suggestions from the reviewers.

MAJOR COMMENTS The discussion is extremely convoluted and goes in all sorts of directions but nothing really conclusive comes out of it. By looking at the pCO2 plot in Figure 2, we can conclude: - Differences between night and day are not statistically different whatever the site. The authors should try to explain this by comparing with other *river* sites where night-day pCO2 differences have been reported. The low phytoplankton biomass as indicated by chlorophyll-a content and the low differences in night-day temperature that induce low variations in CO2 solubility probably explain the low daily variations in pCO2. Anyway the low daily variations of pCO2 are an interesting aspect of the paper that deserves a longer discussion. Possibly make a table with studies that have shown daily variations of pCO2 with other variables such as average Chlorophyll-a content or POC, daily changes in temperature, . . . Thank you very much for the suggestion. We revised the text for discussing the temperature influence on pCO2 and fCO2 of the Red River in page 9, section "4.1 Temporal variations of pCO2 and CO2 fluxes of the lower Red River" as suggested. For example, we revise the text in page 9: "Concerning the lower Red River, water temperature did not show clear variation between the day and the night. In addition, low Chl-a concentrations were measured, from 0.5 to 3.1 $\mu$g L-1, probably as a result of the high turbidity limiting light penetration in the water column. Thus, phytoplankton activity had a low influence on C dynamic in the lower Red River system. Consequently, there are no clear variations of pCO2 and CO2 fluxes between the day and the night time at the different stations along the lower Red River" - Overall the spatial gradients (differences among sites) are very low, except for Hoa Binh. Probably that all sites are located very close and correspond to similar sized rivers with similar catchment characteristics (land cover + lithology) so that the spatial differences are low. An explanation for the higher values of Hoa Binh is needed. Is it because it's downstream of a dam? This would make sense based on

existing literature (Sinnamary river, cf paper by Guérin et al.). Maybe it's mentioned somewhere in the text, but I missed the information being distracted by all of the other marginal bits and pieces of discussion.

Thank you for the suggestion. We revised the discussion about the influence of the dam in page 10 and 11, section "Influence of dams on pCO2 and CO2 emission"

"4.2.4 Influence of dams on pCO2 and CO2 emission Previously, reservoirs were suggested to decrease riverine pCO2 due to increased residence times and autotrophic production (Wang et al., 2007). However, Lauerward et al., (2015) found a low negative correlation between them. Abril et al., (2005) noted that intense mineralization of organic matter (OM) originating from the reservoir was possibly a significant source for pCO2 value in downstream river. In addition, the influence of the dam on the gas transfer velocity and then CO2 outgassing flux in the river downstream of the dam was also demonstrated in the study of the Sinnamary River (Guérin et al., 2007). In the present study, in the upstream part, pCO2 ranged from 964 ppm (at Yen Bai) to 3,830 ppm (at Hoa Binh), being highest at the Hoa Binh site where the lowest pH values were measured. Higher k600 values (from 63 to 68 cm h-1) were also observed at the Hoa Binh and Vu Quang sites. Noted that the Hoa Binh site is situated downstream a series of reservoirs, which have been constructed in both Chinese and Vietnamese parts including two large dams Hoa Binh (in 1989) and Son La (in 2010). The Vu Quang site is located in the downstream of a series of reservoirs, including two important Thac Ba (in 1970) and Tuyen Quang (in 2010). Previous studies emphasized that these dams have impacted water and sediment discharges downstream (Ha and Vu 2012; Ngo et al. 2014; Lu et al. 2015) with significant sediment deposition being observed in the reservoirs (Dang et al. 2010; Vinh et al. 2014; Lu et al. 2015). Thus, the higher pCO2 measured at these sites (average value of 3129 ± 32 ppm) may reflect the increased decomposition of OM and/or the water perturbation due to dam construction, especially for the Da River. The impact of dams on downstream pCO2 may be less for the Lo and the Thao Rivers (average values of 1395 ± 63 ppm and 993 ± 14 ppm, respectively),

where less numbers and less size (only small and medium) of dams/reservoirs were built up in their upstream parts. Thus, the high pCO2 measured at these stations may reflect the increased decomposition of OM and/or the water perturbation due to the large dam construction."

- The seasonal variations are comparatively very small (except for Hoa Binh). This is an intriguing result that deserves being explored. The authors should compile in a table studies that report seasonal cycles of pCO2 (report the min-max of pCO2) and other relevant variables such as ratio of max/min of discharge, seasonal changes (minC2 max) of POC and DOC. It might be useful to show a plot with the full seasonal cycle of freshwater discharge and indicate the two sampling periods. This would allow readers to situate the samplings on the hydrograph. A table with the values of pCO2, POC, DOC, Chl a, SS and water discharge in dry and wet season was added in the in the Supplementary Material. A figure of daily river discharges at the outlets of the Thao, Da and Lo tributaries and the main axe of the Red River at Hanoi and SonTay was added in the Supplementary Material.

- The fact that the city of Hanoi does not seem to influence markedly the O2 and pCO2 levels is also intriguing and deserves some discussion. Was the station located within the city itself or slightly downstream ? Could this be due to the fact that freshwater discharge is relatively important (2000-3000 m3/s ?). A tabular comparison of studies that have shown the influence of cities on river CO2 might be useful. In rivers, the main driver of the gas transfer velocity is turbulence generated by flow. I suggest that the authors use flow velocity data (that are available at gauging stations) and use the equations of Raymond et al. (2012) to compute the gas transfer velocity and fluxes (equation NâŮę5 of Table 2 of Raymond et al. (2012) is recommended).

Thank you very much for the suggestion. We now calculate the k600 by using the equation of (Raymond et al 2012) as suggested. Hanoi station locates within the city itself and at this site, the river has not received yet the wastewater discharge of the whole city. This may explain that the city of Hanoi does not seem to influence markedly

the O2 and pCO2 levels. We revised the section "Influence of population density on pCO2 and CO2 emission" in page 12 as followings:

"4.2.5 Influence of population density on pCO2 and CO2 emission Previous studies demonstrated very high value of pCO2 in river estuaries as a result of different human activities. For instance, pCO2 up to âĹij25,000 ppm was measured in the Rhine estuary (Kempe, 1982) or up to âĹij15,200 ppm in the Scheldt estuaries due to high discharge of pollutants (Borges and Frankignoulle, 2002). Concerning the Red River, from the upstream to the downstream part of the main axe, pCO2 together with CO2 outgassing flux slightly increased from Yen Bai (993 ± 14 ppm and 364.9 ± 10.3 mmol m-2 d-1 respectively) to Hanoi (1,275 ± 17 ppm and 304 ± 7.3 mmol m-2 d-1), whatever the season. However, it is worth to note that the Hanoi station was located within the city itself and at this station, the river has not yet received the wastewater discharge of the whole city. Consequently, the Hanoi station in this study may not reflect the influence of whole city, with probably lower O2 and higher pCO2 levels as observed for other urban rivers in the Red River Delta (Trinh et al., 2007; 2009; 2012)"

For all of the spatial and seasonal comparisons, some firm statistical testing is required (t-test, ANOVA, . . .). We now checked all of the spatial and seasonal comparisons by statistical testing from t-test and ANOVA, in page 6 for the method and throughout the ms. "Student t-test was used to test the difference of variables values between the two different times (the wet and the dry) and (the day and the night), whereas ANOVA was used to test the difference of variables within stations on the measured mean variables. Probabilities (p) were determined and a p value of < 0.05 was considered to be significant"

SPECIFIC COMMENTS L29-30: could also be due direct inputs of CO2 from soils or wetlands. Yes, it is a possibility, and it was added

L48-51: this is a very complicated way to state that C fluxes from rivers depend on lithology and land cover. This sentence was revised "Riverine carbon concentrations

and CO2 outgassing from rivers are impacted by both natural and human factors (Liu et al., 2016; Liu et al., 2017)."

L59: improve instead of "precise" It was changed

L75: Please justify the choice of the 5 stations. Also, it needs to be clearly explained that one of the stations is influenced by seawater intrusion. Are the other stations affected by tidal wave propagation ? The paragraph was revised as followings: "Five stations were studied in the lower Red River (Vietnam): Yen Bai station (at the outlet of the Thao river); Hoa Binh station (after Son La and Hoa Binh reservoirs, at the outlet of the Da River); Vu Quang (at the outlet of the Lo River); Hanoi and Ba Lat stations (in the main course of the Red River downstream). The three stations Yen Bai, Vu Quang and Hoa Binh are representative for three main tributaries (Thao, Da and Lo) of the upstream Red River whereas the Ha Noi station is representative for the main course Red River after confluence of three main tributaries. Within 5 stations observed, only the Ba Lat station which is located in the Red River mouth, about 13 km from the sea is influenced by seawater intrusion (Fig 1)."

L94: provide information on land cover, lithology, and other relevant catchment characteristics for the studied rivers. Some information concerning geomorphology, lithology or other river characteristics of the studied rivers were added in the text (page 3) as suggested. "The Delta is located in a very flat and low land, with an elevation ranging from 0.4 to 12 m above sea level (Nguyen Ngoc Sinh et al., 1995). Previous studies showed the difference of lithology in the three upstream tributaries: Paleozoic sedimentary rocks (55.5

L107: replace "sensor" by CTD It is not a CTD. It is a YSI probe with different sensors. So we revised the sentence "Physico-chemical parameters were automatically recorded every minute during 24h for each sampling campaign: pH, turbidity, salinity, chlorophyll a by a YSI6920 multiparameters probe (YSI, USA)"

L110: "all data must be entered on the documents" ? What does this mean ? This

sentence was deleted

L113: total "alkalinity" It was corrected

L117-126: provide precision and accuracy of all the measured variables. Everywhere in the ms (text and tables) pCO2 values should not be given at the tenth of ppm, given that with an equilibrator precision of pCO2 measurements is typically of +/-1 ppm and accuracy of pCO2 measurements is typically of +/- 3-5 ppm (at best).

Precision and accuracy of the measured variables were added in the section 2.2 Sampling procedures and analysis

L125: what was the volume of sample water? What acid and concentration was used ? How was the end-point determined? What titrator was used? The authors discuss the differences of pCO2 measured directly and calculated from pH and total alkalinity (TA). However, for this discussion to be meaningful it is necessary to have an idea of the quality of the pH and TA measurements, and this is only possible if analytical techniques are described in detail. This is lacking for TA here, and elsewhere the authors need to provide information on the pH electrode calibration. Type of buffers, frequency of calibration, etc. The method for alkalinity determination was added in page 4 "Total alkalinity of the hourly samples was immediately determined on non-filtered water samples (30ml water sample) in situ by titration method with 0.01M HCl (APHA, 1995). For each sample, triplicates were titrated and the analytical error was below 3 pH electrode was calibrated using standard solutions (pH = 4.01 and pH = 6.88, Merck) and the pH precision and accuracy was $\pm$ 0.01.

TA data need to be expressed in $\mu$mol/L (as by convention). This is the unit of the TA as input variable for the CO2sys program. So why express the data in mg/L ? Further the data expressed per mass (mg/L) instead of per moles are extremely confusing. Is it mg of C ? Or mg of H+ ? (based on the conventional definition of TA as the quantity (number of moles) of protons to titrate bases in one kilogram (or L) of water). Should it be mg of C, then the TA for the first line of Table 2 (105 mg/L) would be 8750 $\mu$mol/L.

In this case the computation of pCO2 gives 3670 ppm for a pH of 8.2 and 26.4âŲęC (and not 270 ppm as stated). Should it be mg of CaCO3, then the TA for the first line of Table 2 (105 mg/L) would be 1050 $\mu$mol/L. In this case the computation of pCO2 gives 440 ppm for a pH of 8.2 and 26.4âŲęC (and not 270 ppm as stated). Should it be mg of CO3, then the TA for the first line of Table 2 (105 mg/L) would be 1750 $\mu$mol/L. In this case the computation of pCO2 gives 733 ppm for a pH of 8.2 and 26.4âŲęC (and not 270 ppm as stated). The result from TA analysis is in mg/l CaCO3 as units. During calculating pCO2 by CO2-SYS, when needed, it was converted into $\mu$muol/l or mol/l. In the revised version we re-calculated pCO2 values

L140: Abbreviation IRGA not defined. It was added in the text "a portable InfraRed Gas Analyser (IRGA) (Licor 820, Licor[®], USA)"

L144: replace "balanced" by equilibrated. It was replaced

L156: This equation was established in the 1970's well before the paper of Raymond and Cole. This sentence was deleted for the revised version

L160: replace "function" by parameterization. This sentence was deleted for the revised version.

L167: solubility changes with temperature, why did you state a constant value ? We calculated the fCO2 at 5 sited of the Red River with different $\alpha$ values : $\alpha$ = 3.941 10-2 mol.L-1.atm-1 at 24 oC for dry season and $\alpha$ = 3.138 10-2 mol.L-1.atm-1 at 27 oC for rainy season. The average flux of CO2 outgassing was 525.7 mmol.m-2.d-1. Then, this value was very closed with the one calculated from the constant $\alpha$ = 3.4 10-2 mol.L-1.atm-1 we used for both rainy and dry seasons of the Red River in this study, which resulted in the fCO2 average of the whole Red River 530 mmol.m-2.d-1. An explanation was added in page 6 "ïĄą is the solubility coefficient of CO2 for given temperature and salinity (Weiss, 1974) (mol L-1 atm-1). In this case, ïĄą = 0.034 mol L-1 atm-1. In this study, salinity variations were low, except for the Ba Lat station. Temperature did not change a lot. We checked the influence of different ïĄą values in

the dry ($\alpha$ = 3.941 10-2 mol L-1 atm-1 at 24 oC) and the wet season ($\alpha$ = 3.138 10-2 mol L-1 atm-1 at 27 oC) at the 5 sites and compared with the constant ïĄą value of 0.034 mol L-1 atm-1"

L174-190: I suggest that this section is removed, as this topic has been discussed at length in several dedicated papers. This section was deleted as suggested

L195: The speciation of DIC was established decades before the papers of Cai et al. 2008 and Sun et al. 2010). The paper of Park (1969) formalised the equations to compute the speciation from all of the possible combinations. The reference Park (1969) was added

L203: The calculated pCO2 is about 5 times lower than the pCO2 measured directly, this can hardly be considered as 'similar results' as stated. We apologize for the error of values pCO2 calculated in the table 2. The text was also revised for discussion about the difference between pCO2 measured and calculated in page 7-8, as followings:

"3.3. Comparisons of the pCO2 results obtained by the two methods pCO2 along the lower Red River (Vietnam) in the dry and the wet seasons were determined by two methods: i) direct measurements using an equilibrator connected to an IRGA, ii) calculated from pH and alkalinity using the CO2-SYS$^{®}$ software. The direct pCO2 measurements gave slightly higher values than the calculated ones (Table 2), but the values of two methods were similar and presented the same trend of spatial and seasonal variations (R2 = 0.77, Fig. 2; Table 2). Lower values of the calculated pCO2 in this study may be caused by the analytical errors in pH or under-estimation of total alkalinity. Similarly, the CO2 outgassing rates which were calculated from measured pCO2 from equilibrator were higher than the ones derived from the calculated pCO2 from CO2-SYS, however they are in the same orders and have similar variation trends (Table 3, Fig 2). "

L230-235: Present and discuss either salinity or conductivity but not both as they provide the same information. We revised the sentence as followings: "Conductivity followed the same trend of the salinity variation which were around 0.2±0.0 mS cm-1 for 4 upstream sites and higher values were found for the Ba Lat site, especially in dry season (up to 6.6±3.4 mS cm-1)(p< 0.05)" in page 7.

L233: Statistical test needed. Statistical test was added throughout the revised ms as suggested

L245-251: Statistical test needed. Statistical test was added throughout the revised ms as suggested

L260: The "good" correlation is of marginal interest what's relevant is that the pCO2 values from the two methods differ by a factor of 5 Again, the text was also revised for discussion about the difference between pCO2 measured and calculated, as followings in page 7 -8 "3.3. Comparisons of the pCO2 results obtained by the two methods pCO2 along the lower Red River (Vietnam) in the dry and the wet seasons were determined by two methods: i) direct measurements using an equilibrator connected to an IRGA, ii) calculated from pH and alkalinity using the CO2-SYS$^®$ software. The direct pCO2 measurements gave slightly higher values than the calculated ones (Table 2), but the values of two methods were similar and presented the same trend of spatial and seasonal variations (R2 = 0.77, Fig. 2; Table 2). Lower values of the calculated pCO2 in this study may be caused by the analytical errors in pH or under-estimation of total alkalinity. Similarly, the CO2 outgassing rates which were calculated from measured pCO2 from equilibrator were higher than the ones derived from the calculated pCO2 from CO2-SYS, however they are in the same orders and have similar variation trends (Table 3, Fig 2).

L261-269: This result is extremely intriguing because the sampled rivers are within the "acceptable" range of applicability of the computation of pCO2 from pH and TA, with low DOC and high pH values. The under-estimation of the computed pCO2 could be due to a bias in the measurements of pH and TA, which is not possible to evaluate given the lack of information in the material and methods. Anyway, the under-estimation of

TA and/or over-estimation of pH could explain why the computed pCO2 is very low compared to directly measured pCO2. The method for TA and pH determination was added in the revised manuscript in page 4. We may under-estimate TA resulting the lower computed pCO2 compared to directly measured pCO2. A paragraph was added to explain the difference of pCO2 computed and pCO2 measured in page 7 -8

"3.3. Comparisons of the pCO2 results obtained by the two methods pCO2 along the lower Red River (Vietnam) in the dry and the wet seasons were determined by two methods: i) direct measurements using an equilibrator connected to an IRGA, ii) calculated from pH and alkalinity using the CO2-SYS$^{®}$ software. The direct pCO2 measurements gave slightly higher values than the calculated ones (Table 2), but the values of two methods were similar and presented the same trend of spatial and seasonal variations (R2 = 0.77, Fig. 2; Table 2). Lower values of the calculated pCO2 in this study may be caused by the analytical errors in pH or under-estimation of total alkalinity. Similarly, the CO2 outgassing rates which were calculated from measured pCO2 from equilibrator were higher than the ones derived from the calculated pCO2 from CO2-SYS, however they are in the same orders and have similar variation trends (Table 3, Fig 2).

L271-298: This discussion is of marginal interest because it depends on the way the gas transfer velocity was computed from a parameterisation as function of wind speed. pCO2 did not change markedly from night to day, but wind speed was higher during the day than the night. This is somewhat trivial, and if the authors re-compute the fluxes with a parameterization as function of flow, this day-night difference will be erased. Thank you for the comments. We remove this previous discussion since we recalculated the k600 and flux of CO2 outgassing. K600 was calculated as presented in the section "2.5 CO2 fluxes determination" , page 5-6: "In this study, k600 was calculated using the equation from Raymond et al. (2012) based on stream velocity (V, in m s -1 ), slope (S, unitless), depth (D, in meters) and discharge (Q, in m3 s-1), as follow: k600 = 4725 $\pm$ 445 x (V x S) 0.86 $\pm$ 0.016 x Q-0.14 $\pm$ 0.012 x D 0.66 $\pm$ 0.029 Eq. (2) L298:

Regarding the Ho et al. (2016) study, a note of caution is needed. This study used wind speeds from a sonic anemometer above the mangrove forest (obviously higher wind speeds that the level of the river below the canopy) and located 4 km from the coastline while the tracer injection point was located 12 km away from the coastline. For both these reasons, wind speed data used to build the Ho et al. (2016) relationship is over-estimated, meaning that the relationship itself is not reliable, and the role of currents in generating turbulence (and driving the gas transfer velocity) probably underestimated. Thank you for the comments. We remove this part of the discussion since we now recalculate the k600 and flux of CO2 outgassing of the Red River using a different equation. K600 was calculated as presented in the section "2.5 CO2 fluxes determination", page 5-6: "In this study, k600 was calculated using the equation from Raymond et al. (2012) based on stream velocity (V, in m s -1 ), slope (S, unitless), depth (D, in meters) and discharge (Q, in m3 s-1), as follow: k600 = 4725 $\pm$ 445 x (V x S) 0.86 $\pm$ 0.016 x Q-0.14 $\pm$ 0.012 x D 0.66 $\pm$ 0.029 Eq. (2)

L299-312: In this discussion there's a mix of studies in lakes and mangroves, which makes little sense when discussing CO2 dynamics in rivers. Thank you for the suggestion. We revised the paragraph and removed all reference studies un-necessary in lakes and mangroves throughout the revised ms.

L302: Roulet et al (1997) report on beaver ponds. This is a very specific environment that is not very relevant for comparison with Red river. The reference Roulet et al (1997) was removed as suggested.

L321: references are needed to back this statement. The reference (Richey et al 2002) was added in the revised ms.

L323: or (in addition) inputs of CO2 from wetlands. It was added as suggested

L335: Chanda study deals with mangroves and Takahashi with oceans. This is irrelevant (and out of scope) to discuss CO2 dynamics in rivers. The references were removed.

L338: Comparison with Southern Ocean is irrelevant (and out of scope). The sentence was removed

L346: Why did you not include freshwater discharge in the PCA and the correlation matrix analysis ? The PCA and the correlation matrix took into account river discharge.

L359: Statistical test needed. Statistical test was added throughout the revised ms.

L397: The cited references deal with Artic Ocean and sea-ice, not with Pearl River. This was removed from the revised ms.

L401: Statistical test needed. Statistical test was added throughout the revised ms.

L407-408: The city of Ha Noi does seem to influence DOC and POC but there seems to be little effect on O2 and pCO2. This is intriguing. We revised the discussion in page 12, in the section "4.2.5 Influence of population density on pCO2 and CO2 emission Previous studies demonstrated very high value of pCO2 in river estuaries as a result of different human activities. For instance, pCO2 up to âĹij25,000 ppm was measured in the Rhine estuary (Kempe, 1982) or up to âĹij15,200 ppm in the Scheldt estuaries due to high discharge of pollutants (Borges and Frankignoulle, 2002). Concerning the Red River, from the upstream to the downstream part of the main axe, pCO2 together with CO2 outgassing flux slightly increased from Yen Bai (993 ± 14 ppm and 364.9 ± 10.3 mmol m-2 d-1 respectively) to Hanoi (1,275 ± 17 ppm and 304 ± 7.3 mmol m-2 d-1), whatever the season. However, it is worth to note that the Hanoi station was located within the city itself and at this station, the river has not yet received the wastewater discharge of the whole city. Consequently, the Hanoi station in this study may not reflect the influence of whole city, with probably lower O2 and higher pCO2 levels as observed for other urban rivers in the Red River Delta (Trinh et al., 2007; 2009; 2012). "

L410: some of the cited "rivers" are in fact mangrove brackish systems (irrelevant and out of scope). We checked the cited "rivers" in this paragraph

L426-429: the authors provide no evidence to back up this hypothesis that is mainly speculation. The sentence was removed from the revised manuscript.

L427: Elsewhere in the discussion the authors say that the Hoa Binh site has high pCO2 because downstream of a dam, and here the authors say the low pCO2 values in the Red River is also due dams. This cannot work both ways. Should the authors want to explore why pCO2 values are lower in the Red River than other rivers world-wide comparing levels of DOC and POC with other rivers might be useful. Also, some studies have provided hypothesis to explain large-scale variations of pCO2 across river catchments such as productivity, population density, temperature, . . . e.g. the Lauer-wald study that is cited by the authors. The sentence was removed from the revised manuscript. The discussion about different factors that impacted on pCO2 and fCO2 of the Red River was re-written and re-organized through the revised manuscript.

L434-451: This section provides a summary of the paper (and duplicates the abstract) and does not provide a real conclusion. The conclusion was revised

L565: typo in "Costs" The reference was removed from the revised manuscript

L598: Nathalie is the first name not the family name. The first name and family name of the authors in this reference were corrected

L616: Patricia is the first name not the family name. The reference was removed from the revised manuscript.

L618 and L650: This reference appears twice. The reference was removed from the revised manuscript.

Table 3: There's no need to provide the air-water CO2 fluxes derived from the pCO2 calculated from pH and TA. Since calculated pCO2 is under-estimated compared to directly measured pCO2, this is also the case for the fluxes, in a very predictable and trivial way. The flux of CO2 outgassing derived from the pCO2 calculated from pH and TA was removed from the table 3.

In tables and figures is given in the turbidity of the probe in NTU, when in fact authors measured TSM. Why not show also (or instead) the TSM data ? Also, it might be useful to show the POC follows closely TSM, but The data on TSM and

Please also note the supplement to this comment:
https://www.biogeosciences-discuss.net/bg-2017-505/bg-2017-505-AC5-supplement.pdf

[Figure]

Supplementary Material

**CO$_2$ partial pressure and CO$_2$ emission along the lower Red River (Vietnam)**

Thi Phuong Quynh Le[1*], Cyril Marchand[2,3], Cuong Tu Ho[4], Nhu Da Le[1], Thi Thuy Duong[4], Trung Kien Nguyen[4], XiXi Lu[5], Phuong Kieu Doan[1], Thi Mai Huong Nguyen[1] and Duy An Vu[1]

[1]: Institute of Natural Product Chemistry, Vietnam Academy of Science and Technology, 18 Hoang Quoc Viet Road, Cau Giay, Hanoi, Vietnam.
[2]: IMPMC, Institut de Recherche pour le Développement (IRD), UPMC, CNRS, MNHN, Noumea, New Caledonia, France.
[3]: Faculty of Chemistry, University of Science – VNUHCM, 225 Nguyen Van Cu, Ho Chi Minh City, Vietnam
[4]: Institute of Environmental Technology, Vietnam Academy of Science and Technology, 18 Hoang Quoc Viet Road, Cau Giay, Hanoi, Vietnam.
[5]: Department of Geography, National University of Singapore, Arts Link 1, Singapore 117570, Singapore.

*Correspondence to*: Thi Phuong Quynh Le  (quynhltp@yahoo.com   or    quynhltp@gmail.com)

**Figure SM1**: Daily variation of river discharge at the outlet of the Thao (Yen Bai), Da (Hoa Binh), Lo

---

## Author Response (AR1)

**Dear Prof Chris Le, the Editors-in-Chief of the Journal of Biogeosciences**

**Dear Dr Park, Dr Sarma, Dr Abril and Dr Butman, the Editors of the Special issues**
Human impacts on carbon fluxes in Asian river systems

Please find enclosed the revised manuscript: "**_$CO_2$ partial pressure and $CO_2$ emissions from the lower Red River (Vietnam)_**" to be submitted as an Original Research Article to the _Biogeosciences_. All co-authors know and agree with the contents of the revised manuscript and declare no competing financial interests.

We would like to thank you, the editors and the reviewers for the helpful comments for improving the quality of the manuscript. As required, we have highlighted all the changes in blue and provide point by point replies to the reviewers' comments. We do hope that this now fully revised manuscript is suitable for publication in your journal.

Thank you very much for your kind consideration.
Sincerely yours,
Dr Thi Phuong Quynh LE on behalf of the authors.

Dr. Thi Phuong Quynh LE,
Laboratory of Environmental Chemistry,
Institute of Natural Products Chemistry,
Vietnam Academy of Science and Technology,
18-Hoang Quoc Viet, Cau Giay, Hanoi, Vietnam
Tel.: 84-24-37916621        Fax: 84-24-37564390,

E-mail: quynhltp@yahoo.com    or quynhltp@gmail.com

**Dear Authors,**

**Please note that I do not see a final version of the manuscript but it appears you have addressed the concerns of the reviewers quite well. A decision will be made once a final version of the manuscript is submitted and can be properly reviewed.**

**Sincerely,**

**David Butman**

This paper provides some important quantification of CO2 concentration, evasion rates, and temporal and spatial heterogeneity in an understudied Southeast Asian river system. Given the lack of data available on these systems, the concentration data presented in the paper is valuable on its own.

I am concerned, however, at the large discrepancy between calculated and measured CO2 given the lack of a reasonable explanation aside from calculation error. Furthermore, along with what the other reviewer wrote, the reliance on wind-speed as the only determinant factor for k600 is subject to large errors in flux estimation. Knowing this, it is nearly impossible to assess how well correlated CO2 flux is with any of the environmental parameters used in the multi-variate analysis at the end of the paper. I think if the authors figured out why their calculated values are off, used a more broadly accepted model to estimate k600 (or better yet, measured it directly), and simply presented the concentration and flux measurements from the Red River, it would be a valuable contribution to the literature.

*Thank you very much for the helpful comments. We revised the paper taking into account your comments concerning k600. k600 now is calculated from the formula proposed by Raymond et al (2012), based on different variables such as river discharge, water velocity, slope. The values of k600 are now considered more realistic and then CO₂ flux evasion was recalculated.*

Specific Comments:

-How exactly does the data presented in this manuscript relate to anthropogenic impacts? The authors suggest that the Red River is "strongly" affected by human activities but do not provide results that suggest humans have altered the amount or way by which CO2 is evaded from the river. The

"influence of dams" based simply on observing higher concentrations at that site is unconvincing – there is no direct evidence. Likewise, the ascribed influence of human population is equally weak.

*Thank you for the comments. We revised the discussion about the human impact on $pCO_2$ and $fCO_2$ of the Red river in the section "4.2 Spatial variations of $pCO_2$ and $fCO_2$ outgassing in pages 11- 12. In this section, separated factors such as dam impoundment, population density and land-use were discussed.*

- There are more accurate ways to calculate k600 than from wind speed. As the other commenter suggested, instantaneous discharge, flow hydraulics, and even channel slope may provide better or more robust ways to model k600 compared to wind speed.

The explanation that wind speed is driving the diel signature in CO2 flux is circular. Wind speed is how the flux was calculated in the first place, so higher winds during the day will always yield a "higher flux" of CO2 during the day. CO2 might have a completely different diel pattern if modelled with something other than wind speed.

It's hard to evaluate the seasonal variation of CO2 fluxes generated from only wind speed and concentration

*Thank you very much for the helpful comments. As mentioned above, k600 value was determined by the method proposed by Raymond et al (2012), then the $CO_2$ outgassing fluxes were recalculated and discussed for their seasonal and spatial variations.*

*K600 was calculated as presented in the section "2.5 $CO_2$ fluxes determination" , page 5-6:*
*"In this study, $k_{600}$ was calculated using the equation from Raymond et al. (2012) based on stream velocity (V, in m s $^{-1}$ ), slope (S, unitless), depth (D, in meters) and discharge (Q, in $m^3 s^{-1}$), as follow:*

$$k_{600} = 4725 \pm 445 \; x \; (V \; x \; S)^{\,0.86 \pm 0.016} \; x \; Q^{-0.14 \pm 0.012} \; x \; D^{\,0.66 \pm 0.029} \quad Eq.\ (2)$$

- Although the authors suggest that their direct and calculated pCO2 values were well correlated, they do not seem to correspond very well at all. The y-axis in Figure 3 makes it impossible to appreciate the noise in this relationship. More importantly, there is no plausible explanation as to why the slope is not close to 1, but rather that calculated pCO2 was nearly 1/5 of that measured directly by their equilibrator.

*We apologize for the errors of $pCO_2$ calculation values in the previous version of the manuscript. Values were corrected in the revised manuscript (see table 2). Figure 2 (fig 3 old) was also corrected. A difference between $pCO_2$ calculated and measured was found and discussed in the revised manuscript page 7.*

- No specific reasoning given as to why temperature was positively correlated with pCO2 concentration except enhanced weathering rates. Did the authors consider instream respiration?

*This is possibly related to enhanced decomposition rates of organic matter.*

-What are the +/- values in any of the tables? Standard deviation?

*They are standard deviation. We added the information in the table captions.*

- There seems to be overall very little diel signature in any of the water chemistry data presented in the paper. All of the differences seem to fit within the error bars of each "average" measurement.

*Yes, we fully agree with the reviewer, and the discussion concerning day-night variations (first paragraph of the Discussion) was rewritten. $pCO_2$ differences between night and day were really low, most probably because of low temperature difference and low photosynthetic activity due to the turbidity of the Red River.*

Technical Corrections:

17: I would avoid the use of the word "good" when describing a river system

*The word "good" was replaced by "representative"*

19: Not sure what is meant by "carbon dynamic"

*The sentence was revised "This study aims to quantify the spatial and seasonal variability of $CO_2$ partial pressure and $CO_2$ emission of the lower Red River system"*

21-22: Do you mean "relative" rather than "in contrast"?

*"Relative" was replaced as suggested*

42-43: Unclear sentence summarizing Raymond 2013

*The sentence was revised "Raymond et al. (2013) estimated a global evasion rate of 2.1 Pg C $yr^{-1}$ from inland waters, and that global hot spots in stream and rivers which occupy only 20 % of the global land surface represented 70 % of the emission. They emphasised that further studies are needed for identifying the mechanisms controlling $CO_2$ evasion at a global scale."*

113: How was alkalinity measured? There are no methods detailing this

*The method for alkalinity determination was added in the revised manuscript (see page 4)*

277: Discussion mixed in with results, difficult to follow

*Results and discussion are now better separated, as suggested.*

286: More discussion in the results section

*This was separated between results and discussion as suggested*

346: Results being presented in the "Discussion"

*This was separated between results and discussion*

359: Results being presented in the Discussion

*Results were added in the "Discussion" in page 12.*

**Anonymous Referee #2**

General comments:

The authors reported new data on CO2 partial pressure and CO2 evasion from the lower Red River in Vietnam. This paper also provides useful water chemistry data of the river system. Considering that river systems in the Southeast Asia are underrepresented in the global budget of riverine carbon fluxes despite their large river discharge and carbon loads, this study could provide valuable datasets. However, the paper can be improved further by (1) strengthening the estimates of CO2 evasions, (2) providing detailed discussion on the observed patterns, and (3) reorganizing the paragraphs (e.g. some paragraphs in results fit to discussion, and vice versa). I would also suggest that the paper receive a thorough editing for grammar and clarity by the authors. Specific comments are below, which the authors can consider when revising the manuscript.

*Thank you very much for the comments. We revised the manuscript and checked the grammar as suggested.*

Specific comments:

Lines 43–47: These are confusing because the first one (2.7 Pg C yr-1) includes riverine carbon transport, mineralization, and deposition, while the second one ("a lower value" is for $CO_2$ evasion only. I don't think the estimate of $CO_2$ evasion (2.1 Pg C yr-1) from inland waters by Raymond et al. (2013) is a lower value than the previous estimate. Lauerwald et al. (2015) provided a lower estimate, though.

*The paragraph was revised*

*"Natural hydrological processes and biogeochemistry of many rivers in the world have suffered from the influences of climate change and human activities in their drainage basins. Riverine carbon fluxes and outgassing are important parts of the carbon exchange among terrestrial, oceanic and atmospheric environment. Rivers and streams not only transfer various forms of carbon (dissolved and particulate) to oceans, but also evade a significant amount of carbon to the atmosphere (Battin et al., 2009; Richey et al., 2002). Due to $CO_2$ evasion, the flux of carbon that leaves the terrestrial biosphere through global fluvial network was suggested to be twice larger than the amount that ultimately reaches the coastal ocean (Bauer et al., 2013; Regnier et al., 2013). Raymond et al. (2013) estimated a global evasion rate of 2.1 Pg C $yr^{-1}$ from inland waters, and that global hot spots in stream and rivers which occupy only 20 % of the global land surface represented 70 % of the emission. They emphasised*

*that further studies are needed for identifying the mechanisms controlling $CO_2$ evasion at a global scale".*

Lines 125–126: Alkalinity measurement is critical for the calculation of pCO2 and CO2 evasion. Please provide more detailed information on how the alkalinity was measured and double checked. Was there a difference in alkalinity between filtered and unfiltered samples? It seems the turbidity can go up sometimes and I wonder how this could influence the alkalinity of filtered water.

*The method for alkalinity determination was added in the revised manuscript in page 4.*
*" Total alkalinity of the hourly samples was immediately determined on non-filtered water samples (30 ml water sample) in situ by titration method with 0.01M HCl (APHA, 1995). For each sample, triplicates were titrated and the analytical error was below 3 %".*

*We also tried to test the difference in alkalinity between filtered and unfiltered samples at the Hanoi site in March 2018 and found that the alkalinity values were not so different in dry season when suspended solids was not so high .*

Lines 138–: Have you measured $CO_2$ flux directly from the surface of the water and compared that with the calculated values? (e.g. Duc et al., 2013, Environmental Science & Technology, 47, 968-975)

*Yes, we measured $CO_2$ flux directly from the surface of the water using a floating chamber method. We obtained an average $CO_2$ flux of 189 $mmol.m^{-2}.d^{-1}$ (with values ranging from 37.8 – 492.1 $mmol.m^{-2}.day^{-1}$). These values were lower than the values calculated from $pCO_2$ measured using equilibrator method and the equation provided by Frankignoulle et al. (2001) and Raymond and Cole (2011). This result was surprising since it is considered that when the water flow is higher than 0.2 $m\ s^{-1}$, the presence of the chamber induces artificial turbulence that results in elevated $CO_2$ fluxes.*

Lines 143–: I agree with the other reviewers that using wind speed as a component in the equation may not appropriate. Detailed explanation is needed on why the equation was chosen.

*As suggested by reviewers, we calculated k600 by other method proposed by Raymond et al (2012), basing on different variables such as river discharge, water velocity, depth and slope. The values of k600 now are ameliorated and then $CO_2$ flux evasion was recalculated.*

*K600 was calculated as presented in the section "2.5 $CO_2$ fluxes determination" , page 5-6:*

*"In this study, $k_{600}$ was calculated using the equation from Raymond et al. (2012) based on stream velocity (V, in m s $^{-1}$ ), slope (S, unitless), depth (D, in meters) and discharge (Q, in $m^3 s^{-1}$), as follow:*

*$k_{600} = 4725 \pm 445 \; x \; (V \; x \; S)^{0.86 \pm 0.016} \; x \; Q^{-0.14 \pm 0.012} \; x \; D^{0.66 \pm 0.029}$   Eq. (2)*

Lines 256–269: Detailed explanation is needed in "discussion" on why there is such a large difference in the measured and calculated $pCO_2$.

*We apologize for the errors of $pCO_2$ calculation. Values were corrected in the revised manuscript (see table 2). However, a difference between $pCO_2$ calculated and measured was found. An explanation concerning this difference was added in the manuscript revised in page 7 -8.*

*"3.3. Comparisons of the $pCO_2$ results obtained by the two methods*

*$pCO_2$ along the lower Red River (Vietnam) in the dry and the wet seasons were determined by two methods: i) direct measurements using an equilibrator connected to an IRGA, ii) calculated from pH and alkalinity using the $CO_2$-SYS® software. The direct $pCO_2$ measurements gave slightly higher values than the calculated ones (Table 2), but the values of two methods were similar and presented the same trend of spatial and seasonal variations ($R^2$ = 0.77, Fig. 2; Table 2). Lower values of the calculated $pCO_2$ in this study may be caused by the analytical errors in pH or under-estimation of total alkalinity. Similarly, the $CO_2$ outgassing rates which were calculated from measured $pCO_2$ from equilibrator were higher than the ones derived from the calculated $pCO_2$ from $CO_2$-SYS, however they are in the same orders and have similar variation trends (Table 3, Fig 2)"*

Lines 299–312: This paragraph would belong to discussion rather than results. Discussion: The results and discussion are mixed.

*We revised the paragraph*

Lines 400: Are the differences statistically significant?

*Yes. The results from t-test showed that:*

- *Seasonal (dry –wet) variation : difference appeared for all variables (p< 0.05)*
- *Day-night variation: difference for pH (p< 0.05) but no clear difference for other variables*

*The results from t-test and ANOVA test showed that*

*Spatial variation (5 sites) : temperature was not different within 5 sites. Other variables were significant different (p< 0.05).*

*We added the difference statistics significant through the revised ms.*

Lines 411–422: This sentence is too long. Extracting only essential information would be better than this long sentence.

*This sentence is now revised and moved to the section " 4.2 Spatial variations of pCO2 and fCO2 outgassing » in page 10 - 11*

Tables: Is the "+/-" standard error or standard deviation? Please clarify it.

*It's standard deviation. We added the information in the table captions*

Figures: Is the error bar standard error or standard deviation? Please clarify it.

*It is the standard deviation. We added the information in the figure captions*

Figure captions needs to more detailed description of the figures including explanations on legends.

*Figure captions were revised as suggested, and thus more detailed.*

**Anonymous Referee #3**

General Comments:

The manuscript entitled "CO2 partial pressure and CO2 emissions from the lower Red River (Vietnam)" by Le et al provides new and important data on CO2 in the Lower Red River. The results are a good contribution for the understanding of the role of Asian rivers for the global carbon evasion from freshwater ecosystems. However, there are many technical problems that need to be address to increase the quality of the paper. I agree with the other reviewers that the estimate of the CO2 evasion is the main problem in the paper and must be reworked. The figures and tables are clear and well designed. However, I suggested some changes in the number of figures and tables. In summary, the authors should review the flux calculation (see specific comments below); consider k600 calculations based on turbulence generated by water flow and; reorganize Results and Discussion sections. Specific comments and suggestions are below.

*We thank you very much for the helpful comments and suggestions.*

Specific comments:

66 – This sentence suggests that there is a "limited knowledge" about carbon flux. Does it mean that there are published information about carbon flux in the Red River, or not?

*Yes, there are some published information about carbon fluxes for a peri-urban area in the Red River Delta (Trinh et al., 2012) and (Nguyen et al., 2018). We revised the sentence and add the references:*

*"However, there is a lack of data concerning $CO_2$ outgassing and carbon budget of the lower Red River (Trinh et al., 2012, Nguyen et al., 2018)."*

79 – In the figure 1, the Ba Lat station does not seem to be located about 50 km from the sea. Please clarify.

*We apologize for the error. The Ba Lat site is 13 km from the sea. The information was corrected in the text page 3.*

Some references in the introduction are missing (e.g. lines 48, 56)

*References were added in the revised manuscript*

86 – I suggest removing the sentence "see below for the detail river discharge in 2014" and insert the correct reference - figure, table etc.

*The sentence was removed, the word "table 1" was replaced*

109 – The sentence "All solutions used for. . ." is unnecessary.

*This sentence was deleted*

151 – What is the "regulator"? You mean pump or a flux regulator? Please, clarify.

*It was changed into "a flow regulator".*

162 – By convention k600 is the gas transfer velocity normalized to a Schmidt number of 600. And k is the gas transfer velocity. "Sc" should not be the "Schmidt number, normalized to Schmidt number of 600". "Sc" should be the Schmidt number of a given gas (CO2 in this case) at a given temperature. Please, clarify the calculation and check if the calculation is correct.

*We revised the paper by recalculating the k600 proposed by (Raymond et al., 2012). Thus, Sc was removed from the new calculation.*

*K600 was calculated as presented in the section "2.5 $CO_2$ fluxes determination" , page 5-6:*

*"In this study, $k_{600}$ was calculated using the equation from Raymond et al. (2012) based on stream velocity (V, in m s$^{-1}$ ), slope (S, unitless), depth (D, in meters) and discharge (Q, in m$^3$ s$^{-1}$), as follow:*

*$k_{600} = 4725 \pm 445 \times (V \times S)^{0.86 \pm 0.016} \times Q^{-0.14 \pm 0.012} \times D^{0.66 \pm 0.029}$   Eq. (2)*

166 – I wonder why the authors did not considered the temperature variation measured by HOBO sensor to calculate "a"? Were temperature records not made at the same time? In addition, conductivity is much higher in Ba Lat station suggesting differences in salinity among stations. If "a" is a constant (line 167), the salinity variation seems not to be considered in the calculation.

*We calculated the $fCO_2$ at 5 sited of the Red River with different $\alpha$ values : $\alpha = 3.941 \ 10^{-2} \ mol.L^{-1}.atm^{-1}$ at 24 °C for dry season and $\alpha = 3.138 \ 10^{-2} \ mol.L^{-1}.atm^{-1}$ at 27 °C for rainy season. The average flux of $CO_2$ outgassing was 525.7 $mmol.m^{-2}.d^{-1}$. Then, this value was very closed with the one calculated from the constant $\alpha = 3.4 \ 10^{-2} \ mol.L^{-1}.atm^{-1}$ we used for both rainy and dry seasons of the Red River in this study, which resulted in the $fCO_2$ average of the whole Red River 530 $mmol.m^{-2}.d^{-1}$.*

*An explanation was added in page 6*

*"$\alpha$ is the solubility coefficient of $CO_2$ for given temperature and salinity (Weiss, 1974) (mol $L^{-1}$ $atm^{-1}$). In this case, $\alpha = 0.034 \ mol \ L^{-1} \ atm^{-1}$. In this study, salinity variations were low, except for the Ba Lat station. Temperature did not change a lot. We checked the influence of different $\alpha$ values in the dry ($\alpha = 3.941 \ 10^{-2} \ mol \ L^{-1} \ atm^{-1}$ at 24 °C) and the wet season ($\alpha = 3.138 \ 10^{-2} \ mol \ L^{-1} \ atm^{-1}$ at 27 °C) at the 5 sites and compared with the constant $\alpha$ value of 0.034 mol $L^{-1}$ $atm^{-1}$. "*

170 – Please, check the units of each element of the equation. The flux was calculated in $\mu$mol m-2 s-1 but the k, a and pCO2 air is cm h-1, mol L-1 atm-1 and ppm, respectively. Please reorganize the section 2.4.1 for clarification.

*Thank you for the comments. The flux is now calculated from different variables taking into account different units. We added the explanation for converting the final unit of flux calculation in mmol $m^{-2}$ $d^{-1}$ in the section 2.4 $CO_2$ fluxes determination in page 5*

183 – If k600 values are closely related to flow velocity and channel gradient, why author used k600 values calculated from wind speed? The authors should present a convincing explanation or review the calculation and values.

*Thank you for the comment. We now calculate the k600 by the formula proposed by Raymond et al (2012) which is based on river flow velocity, river discharge, depth, slope in pages 5 and 6:*

*"$k_{600}$, was gas transfer velocity of $CO_2$ or piston velocity (cm $h^{-1}$). Some studies indicate that $k_{600}$ values are closely related to flow velocity and channel gradient for rivers (Alin et al., 2011). In this study, $k_{600}$ was calculated using the equation from Raymond et al. (2012) based on stream velocity (V, in m $s^{-1}$), slope (S, unitless), depth (D, in meters) and discharge (Q, in $m^3 s^{-1}$), as follow:*

$$k_{600} = 4725 \pm 445 \ x \ (V \ x \ S)^{\ 0.86 \pm 0.016} \ x \ Q^{-0.14 \pm 0.012} \ x \ D^{\ 0.66 \pm 0.029} \quad Eq. \ (2)"$$

220 – The same results are presented in Table 2 and Figure 2. Please, remove the table or the figure.

*Figure 2 was removed as suggested.*

223 – I suggest add Alkalinity values in the table or figure 2.

*Alkalinity values are presented in the table 2, in the column named "TAlk".*

257 – Change "monsoon" to "wet".

*It was changed*

263-269 – Abril et al., 2015 discuss overestimation of pCO2 using total alkalinity in acidic, organic-rich freshwaters. However, if this water condition is not the case of Red River, the find of Abril is not an explanation for the opposite (lower calculate values when compared with the measured values). These values should be similar or, please, provide a clear and convincing explanation.

*Thank you for the suggestion. We removed the sentence from the revised manuscript.*

276 – Discussion inside the Results section. Results and Discussion are mixed in several parts of these sections. I suggest re-write and reorganize these sections.

*Thank you for the suggestion. We re-writed and reorganized the Results and Discussion sections*

Table 1 – Please double check the station location. I plot the station locations and there are some inconsistences (e.g. Ba Lat is located in the ocean and not in the Red River).

*There was a mistake for the location of the Ba Lat site. This was corrected in table 1 and in the text. Other sites were also checked.*

Table 2 and Figure 2 – Why the values after +/- in table 2 and the error bar in Figure 2 are different? What does they mean?

*We checked the table 2. The figure 2 was removed in the revised manuscript.*

Figure 6 – What the gray and dark circles mean. Dry and wet season? Insert these information in the figure caption.

*The figure 4 demonstrates the Relationship between environmental variables and $pCO_2$ at the five sites of the lower Red River in the wet (Sept) and the dry season (Nov) 2014. The figure and caption were revised.*

**Anonymous Referee #4**

GENERAL COMMENTS

Le et al. report a valuable and potentially interesting data-set of CO2 data measured with an equilibrator at several stations of the Red River. This is a valuable data-set as CO2 data directly measured are lacking worldwide, and in particular in sub-tropical and tropical environments.

However, the paper suffers from a poor writing (English phrasing and syntax) that absolutely needs to be improved.

Also, the presentation and discussion of the results are extremely convoluted. The authors make a list of numerous possible hypothesis but do not really provide a convincing interpretation of the data (a clear and solid "story-line").

Some of the comparisons in the discussion are out of scope and irrelevant such as comparisons with mangroves, beaver ponds, sea-ice and the Southern Ocean.

*Thank you for the comments and suggestions. We revised the manuscript in taking into account all comments/suggestions from the reviewers.*

MAJOR COMMENTS

The discussion is extremely convoluted and goes in all sorts of directions but nothing really conclusive comes out of it. By looking at the pCO2 plot in Figure 2, we can conclude:

- Differences between night and day are not statistically different whatever the site. The authors should try to explain this by comparing with other \*river\* sites where night-day pCO2 differences have been reported. The low phytoplankton biomass as indicated by chlorophyll-a content and the low differences in night-day temperature that induce low variations in CO2 solubility probably explain the low daily variations in pCO2. Anyway the low daily variations of pCO2 are an interesting aspect of the paper that deserves a longer discussion. Possibly make a table with studies that have shown daily variations of pCO2 with other variables such as average Chlorophyll-a content or POC, daily changes in temperature, . . .

*Thank you very much for the suggestion. We revised the text for discussing the temperature influence on $pCO_2$ and $fCO_2$ of the Red River in page 9, section "**4.1 Temporal variations of $pCO_2$ and $CO_2$ fluxes of the lower Red River**" as suggested.*

*For example, we revise the text in page 9:*

*"Concerning the lower Red River, water temperature did not show clear variation between the day and the night. In addition, low Chl-a concentrations were measured, from 0.5 to 3.1 µg $L^{-1}$, probably as a result of the high turbidity limiting light penetration in the water column. Thus, phytoplankton activity had a low influence on C dynamic in the lower Red River system. Consequently, there are no clear variations of $pCO_2$ and $CO_2$ fluxes between the day and the night time at the different stations along the lower Red River"*

- Overall the spatial gradients (differences among sites) are very low, except for Hoa Binh. Probably that all sites are located very close and correspond to similar sized rivers with similar catchment characteristics (land cover + lithology) so that the spatial differences are low. An explanation for the higher values of Hoa Binh is needed. Is it because it's downstream of a dam? This would make sense based on existing literature (Sinnamary river, cf paper by Guérin et al.). Maybe it's mentioned somewhere in the text, but I missed the information being distracted by all of the other marginal bits and pieces of discussion.

*Thank you for the suggestion. We revised the discussion about the influence of the dam in page 10 and 11, section "Influence of dams on $pCO_2$ and $CO_2$ emission"*

*"**4.2.4 Influence of dams on $pCO_2$ and $CO_2$ emission***
*Previously, reservoirs were suggested to decrease riverine $pCO_2$ due to increased residence times and autotrophic production (Wang et al., 2007). However, Lauerward et al., (2015) found a low negative correlation between them. Abril et al., (2005) noted that intense mineralization of organic matter (OM) originating from the reservoir was possibly a significant source for $pCO_2$ value in downstream river. In addition, the influence of the dam on the gas transfer velocity and then $CO_2$ outgassing flux in the river downstream of the dam was also demonstrated in the study of the Sinnamary River (Guérin et al., 2007). In the present study, in the upstream part, $pCO_2$ ranged from 964 ppm (at Yen Bai) to 3,830 ppm (at Hoa Binh), being highest at the Hoa Binh site where the lowest pH values were measured. Higher $k_{600}$ values (from 63 to 68 cm $h^{-1}$) were also observed at the Hoa Binh and Vu Quang sites. Noted that the Hoa Binh site is situated downstream a series of reservoirs, which have been constructed in both Chinese and Vietnamese parts including two large dams Hoa Binh (in 1989) and Son La (in 2010). The Vu Quang site is located in the downstream of a series of reservoirs, including two important Thac Ba (in 1970) and Tuyen Quang (in 2010). Previous studies emphasized that these dams have impacted water and sediment discharges downstream (Ha and Vu 2012; Ngo et al. 2014; Lu et al. 2015) with significant sediment deposition being observed in the reservoirs (Dang et al. 2010; Vinh et al. 2014; Lu et al. 2015). Thus, the higher $pCO_2$ measured at these sites (average value of 3129 ± 32 ppm) may reflect the increased decomposition of OM and/or the water perturbation due to dam construction, especially for the Da River. The impact of dams on downstream $pCO_2$ may be less for the Lo and the Thao Rivers (average values of 1395 ± 63 ppm and 993 ± 14 ppm, respectively), where less numbers and less size (only small and medium) of dams/reservoirs were built up in their upstream parts. Thus, the high $pCO_2$ measured at these stations may reflect the increased decomposition of OM and/or the water perturbation due to the large dam construction."*

- The seasonal variations are comparatively very small (except for Hoa Binh). This is an intriguing result that deserves being explored. The authors should compile in a table studies that report seasonal cycles of pCO2 (report the min-max of pCO2) and other relevant variables such as ratio of max/min of discharge, seasonal changes (minC2 max) of POC and DOC. It might be useful to show a plot with the full seasonal cycle of

freshwater discharge and indicate the two sampling periods. This would allow readers to situate the samplings on the hydrograph.

*A table with the values of pCO$_2$, POC, DOC, Chl a, SS and water discharge in dry and wet season was added in the in the Supplementary Material.*

*A figure of daily river discharges at the outlets of the Thao, Da and Lo tributaries and the main axe of the Red River at Hanoi and SonTay was added in the Supplementary Material.*

- The fact that the city of Hanoi does not seem to influence markedly the O2 and pCO2 levels is also intriguing and deserves some discussion. Was the station located within the city itself or slightly downstream ? Could this be due to the fact that freshwater discharge is relatively important (2000-3000 m3/s ?). A tabular comparison of studies that have shown the influence of cities on river CO2 might be useful. In rivers, the main driver of the gas transfer velocity is turbulence generated by flow. I suggest that the authors use flow velocity data (that are available at gauging stations) and use the equations of Raymond et al. (2012) to compute the gas transfer velocity and fluxes (equation N 5 of Table 2 of Raymond et al. (2012) is recommended).

*Thank you very much for the suggestion. We now calculate the k600 by using the equation of (Raymond et al 2012) as suggested.*

*Hanoi station locates within the city itself and at this site, the river has not received yet the wastewater discharge of the whole city. This may explain that the city of Hanoi does not seem to influence markedly the O$_2$ and pCO$_2$ levels. We revised the section "Influence of population density on pCO$_2$ and CO$_2$ emission" in page 12 as followings:*

*"4.2.5 Influence of population density on pCO$_2$ and CO$_2$ emission*
*Previous studies demonstrated very high value of pCO$_2$ in river estuaries as a result of different human activities. For instance, pCO$_2$ up to ~25,000 ppm was measured in the Rhine estuary (Kempe, 1982) or up to ~15,200 ppm in the Scheldt estuaries due to high discharge of pollutants (Borges and Frankignoulle, 2002).*
*     Concerning the Red River, from the upstream to the downstream part of the main axe, pCO$_2$ together with CO$_2$ outgassing flux slightly increased from Yen Bai (993 ± 14 ppm and 364.9 ± 10.3 mmol m$^{-2}$ d$^{-1}$ respectively) to Hanoi (1,275 ± 17 ppm and 304 ± 7.3 mmol m$^{-2}$ d$^{-1}$), whatever the season. However, it is worth to note that the Hanoi station was located within the city itself and at this station, the river has not yet received the wastewater discharge of the whole city. Consequently, the Hanoi*

*station in this study may not reflect the influence of whole city, with probably lower $O_2$ and higher $pCO_2$ levels as observed for other urban rivers in the Red River Delta (Trinh et al., 2007; 2009; 2012)"*

For all of the spatial and seasonal comparisons, some firm statistical testing is required (t-test, ANOVA, . . .).

*We now checked all of the spatial and seasonal comparisons by statistical testing from t-test and ANOVA, in page 6 for the method and throughout the ms.*

*"Student t-test was used to test the difference of variables values between the two different times (the wet and the dry) and (the day and the night), whereas ANOVA was used to test the difference of variables within stations on the measured mean variables. Probabilities (p) were determined and a p value of < 0.05 was considered to be significant"*

SPECIFIC COMMENTS

L29-30: could also be due direct inputs of CO2 from soils or wetlands.

*Yes, it is a possibility, and it was added*

L48-51: this is a very complicated way to state that C fluxes from rivers depend on lithology and land cover.

*This sentence was revised "Riverine carbon concentrations and $CO_2$ outgassing from rivers are impacted by both natural and human factors (Liu et al., 2016; Liu et al., 2017)."*

L59: improve instead of "precise"

*It was changed*

L75: Please justify the choice of the 5 stations. Also, it needs to be clearly explained that one of the stations is influenced by seawater intrusion. Are the other stations affected by tidal wave propagation ?

*The paragraph was revised as followings:*

*"Five stations were studied in the lower Red River (Vietnam): Yen Bai station (at the outlet of the Thao river); Hoa Binh station (after Son La and Hoa Binh reservoirs, at the outlet of the Da River); Vu Quang (at the outlet of the Lo River); Hanoi and Ba Lat stations (in the main course of the Red River downstream). The three stations Yen Bai, Vu Quang and Hoa Binh are representative for three main tributaries (Thao, Da and Lo) of the upstream Red River whereas the Ha Noi station is representative for the main course Red River after confluence of three main tributaries. Within 5 stations observed, only the Ba Lat station which is located in the Red River mouth, about 13 km from the sea is influenced by seawater intrusion (Fig 1)."*

L94: provide information on land cover, lithology, and other relevant catchment characteristics for the studied rivers.

*Some information concerning geomorphology, lithology or other river characteristics of the studied rivers were added in the text (page 3) as suggested.*

*"The Delta is located in a very flat and low land, with an elevation ranging from 0.4 to 12 m above sea level (Nguyen Ngoc Sinh et al., 1995). Previous studies showed the difference of lithology in the three upstream tributaries: Paleozoic sedimentary rocks (55.5%), Mesozoic silicic rocks (18.0%) and Mesozoic carbonated rocks (16.7%) dominate in the Thao basin, whereas Paleozoic sedimentary rocks (85.3%) and Mesozoic carbonated rocks (14.7%) cover the Da river basin, and the Lo is composed of Mesozoic silicic rocks (21.5%) and Paleozoic sedimentary rocks (72.7%). The delta area is totally covered by alluvial deposits (100%) (Moon et al., 2007; Le et al., 2007). "*

L107: replace "sensor" by CTD

*It is not a CTD. It is a YSI probe with different sensors. So we revised the sentence*

*"Physico-chemical parameters were automatically recorded every minute during 24h for each sampling campaign: pH, turbidity, salinity, chlorophyll a by a YSI6920 multiparameters probe (YSI, USA)"*

L110: "all data must be entered on the documents" ? What does this mean ?

*This sentence was deleted*

L113: total "alkalinity"

*It was corrected*

L117-126: provide precision and accuracy of all the measured variables. Everywhere in the ms (text and tables) pCO2 values should not be given at the tenth of ppm, given that with an equilibrator precision of pCO2 measurements is typically of +/-1 ppm and accuracy of pCO2 measurements is typically of +/- 3-5 ppm (at best).

*Precision and accuracy of the measured variables were added in the section 2.2 Sampling procedures and analysis*

L125: what was the volume of sample water? What acid and concentration was used ? How was the end-point determined? What titrator was used? The authors discuss the differences of pCO2 measured directly and calculated from pH and total alkalinity (TA). However, for this discussion to be meaningful it is necessary to have an idea of the quality of the pH and TA measurements, and this is only possible if analytical techniques are described in detail. This is lacking for TA here, and elsewhere the authors need to provide information on the pH electrode calibration. Type of buffers, frequency of calibration, etc.

*The method for alkalinity determination was added in page 4*

*"Total alkalinity of the hourly samples was immediately determined on non-filtered water samples (30ml water sample) in situ by titration method with 0.01M HCl (APHA, 1995). For each sample, triplicates were titrated and the analytical error was below 3 %."*

*pH electrode was calibrated using standard solutions (pH = 4.01 and pH = 6.88, Merck) and the pH precision and accuracy was ± 0.01.*

TA data need to be expressed in $\mu$mol/L (as by convention). This is the unit of the TA as input variable for the CO2sys program. So why express the data in mg/L ? Further the data expressed per mass (mg/L) instead of per moles are extremely confusing. Is it mg of C ? Or mg of H+ ? (based on the conventional definition of TA as the quantity

(number of moles) of protons to titrate bases in one kilogram (or L) of water). Should it be mg of C, then the TA for the first line of Table 2 (105 mg/L) would be 8750 $\mu$mol/L. In this case the computation of pCO2 gives 3670 ppm for a pH of 8.2 and 26.4 C (and not 270 ppm as stated). Should it be mg of CaCO3, then the TA for the first line of Table 2 (105 mg/L) would be 1050 $\mu$mol/L. In this case the computation of pCO2 gives 440 ppm for a pH of 8.2 and 26.4 C (and not 270 ppm as stated). Should it be mg of CO3, then the TA for the first line of Table 2 (105 mg/L) would be 1750 $\mu$mol/L. In this case the computation of pCO2 gives 733 ppm for a pH of 8.2 and 26.4 C (and not 270 ppm as stated).

*The result from TA analysis is in mg/l $CaCO_3$ as units. During calculating pCO2 by CO2-SYS, when needed, it was converted into $\mu$muol/l or mol/l.*

*In the revised version we re-calculated pCO2 values*

L140: Abbreviation IRGA not defined.

*It was added in the text "a portable InfraRed Gas Analyser (IRGA) (Licor 820, Licor®, USA)"*

L144: replace "balanced" by equilibrated.

*It was replaced*

L156: This equation was established in the 1970's well before the paper of Raymond and Cole.

*This sentence was deleted for the revised version*

L160: replace "function" by parameterization.

*This sentence was deleted for the revised version.*

*L167: solubility changes with temperature, why did you state a constant value ?*

*We calculated the $fCO_2$ at 5 sited of the Red River with different $\alpha$ values : $\alpha = 3.941 \ 10^{-2}$ mol.$L^{-1}$.atm$^{-1}$ at 24 $^o$C for dry season and $\alpha = 3.138 \ 10^{-2}$ mol.$L^{-1}$.atm$^{-1}$ at 27 $^o$C for rainy season. The average flux of $CO_2$ outgassing was 525.7 mmol.$m^{-2}$.$d^{-1}$. Then, this value was very closed with the one calculated from the constant $\alpha = 3.4 \ 10^{-2}$ mol.$L^{-1}$.atm$^{-1}$ we used for both rainy and dry seasons of the Red River in this study, which resulted in the $fCO_2$ average of the whole Red River 530 mmol.$m^{-2}$.$d^{-1}$.*

*An explanation was added in page 6*

*"$\alpha$ is the solubility coefficient of $CO_2$ for given temperature and salinity (Weiss, 1974) (mol $L^{-1}$ atm$^{-1}$). In this case, $\alpha = 0.034$ mol $L^{-1}$ atm$^{-1}$. In this study, salinity variations were low, except for the Ba Lat station. Temperature did not change a lot. We checked the influence of different $\alpha$ values in the dry ($\alpha = 3.941 \ 10^{-2}$ mol $L^{-1}$ atm$^{-1}$ at 24 $^o$C) and the wet season ($\alpha = 3.138 \ 10^{-2}$ mol $L^{-1}$ atm$^{-1}$ at 27 $^o$C) at the 5 sites and compared with the constant $\alpha$ value of 0.034 mol $L^{-1}$ atm$^{-1}$"*

L174-190: I suggest that this section is removed, as this topic has been discussed at length in several dedicated papers.

*This section was deleted as suggested*

L195: The speciation of DIC was established decades before the papers of Cai et al. 2008 and Sun et al. 2010). The paper of Park (1969) formalised the equations to compute the speciation from all of the possible combinations.

*The reference Park (1969) was added*

L203: The calculated pCO2 is about 5 times lower than the pCO2 measured directly, this can hardly be considered as 'similar results' as stated.

*We apologize for the error of values $pCO_2$ calculated in the table 2. The text was also revised for discussion about the difference between $pCO_2$ measured and calculated in page 7-8, as followings:*

*"**3.3. Comparisons of the $pCO_2$ results obtained by the two methods**
$pCO_2$ along the lower Red River (Vietnam) in the dry and the wet seasons were determined by two methods: i) direct measurements using an equilibrator connected to an IRGA, ii) calculated from pH and alkalinity using the $CO_2$-SYS$^®$ software. The direct $pCO_2$ measurements gave slightly higher values than the calculated ones (Table 2), but the values of two methods were similar and presented the same*

*trend of spatial and seasonal variations ($R^2$ = 0.77, Fig. 2; Table 2). Lower values of the calculated $pCO_2$ in this study may be caused by the analytical errors in pH or under-estimation of total alkalinity. Similarly, the $CO_2$ outgassing rates which were calculated from measured $pCO_2$ from equilibrator were higher than the ones derived from the calculated $pCO_2$ from $CO_2$-SYS, however they are in the same orders and have similar variation trends (Table 3, Fig 2).*

*"*

L230-235: Present and discuss either salinity or conductivity but not both as they provide the same information.

*We revised the sentence as followings:*

*"Conductivity followed the same trend of the salinity variation which were around 0.2±0.0 mS $cm^{-1}$ for 4 upstream sites and higher values were found for the Ba Lat site, especially in dry season (up to 6.6±3.4 mS $cm^{-1}$)($p < 0.05$)" in page 7.*

L233: Statistical test needed.

*Statistical test was added throughout the revised ms as suggested*

L245-251: Statistical test needed.

*Statistical test was added throughout the revised ms as suggested*

L260: The "good" correlation is of marginal interest what's relevant is that the pCO2 values from the two methods differ by a factor of 5

*Again, the text was also revised for discussion about the difference between $pCO_2$ measured and calculated, as followings in page 7 -8*

*"**3.3. Comparisons of the $pCO_2$ results obtained by the two methods***

*$pCO_2$ along the lower Red River (Vietnam) in the dry and the wet seasons were determined by two methods: i) direct measurements using an equilibrator connected to an IRGA, ii) calculated from pH and alkalinity using the $CO_2$-SYS® software. The direct $pCO_2$ measurements gave slightly higher values than the calculated ones (Table 2), but the values of two methods were similar and presented the same trend of spatial and seasonal variations ($R^2$ = 0.77, Fig. 2; Table 2). Lower values of the calculated $pCO_2$ in this study may be caused by the analytical errors in pH or under-estimation of total alkalinity. Similarly, the $CO_2$ outgassing rates which were calculated from measured $pCO_2$ from equilibrator were higher than the ones derived from the calculated $pCO_2$ from $CO_2$-SYS, however they are in the same orders and have similar variation trends (Table 3, Fig 2).*

L261-269: This result is extremely intriguing because the sampled rivers are within the "acceptable" range of applicability of the computation of pCO2 from pH and TA, with low DOC and high pH values. The under-estimation of the computed pCO2 could be due to a bias in the measurements of pH and TA, which is not possible to evaluate given the lack of information in the material and methods. Anyway, the under-estimation of TA and/or over-estimation of pH could explain why the computed pCO2 is very low compared to directly measured pCO2.

*The method for TA and pH determination was added in the revised manuscript in page 4. We may under-estimate TA resulting the lower computed pCO2 compared to directly measured pCO2. A paragraph was added to explain the difference of pCO2 computed and $pCO_2$ measured in page 7 -8*

*"3.3. Comparisons of the $pCO_2$ results obtained by the two methods*

*$pCO_2$ along the lower Red River (Vietnam) in the dry and the wet seasons were determined by two methods: i) direct measurements using an equilibrator connected to an IRGA, ii) calculated from pH and alkalinity using the $CO_2$-SYS® software. The direct $pCO_2$ measurements gave slightly higher values than the calculated ones (Table 2), but the values of two methods were similar and presented the same trend of spatial and seasonal variations ($R^2$ = 0.77, Fig. 2; Table 2). Lower values of the calculated $pCO_2$ in this study may be caused by the analytical errors in pH or under-estimation of total alkalinity. Similarly, the $CO_2$ outgassing rates which were calculated from measured $pCO_2$ from equilibrator were higher than the ones derived from the calculated $pCO_2$ from $CO_2$-SYS, however they are in the same orders and have similar variation trends (Table 3, Fig 2).*

L271-298: This discussion is of marginal interest because it depends on the way the gas transfer velocity was computed from a parameterisation as function of wind speed. pCO2 did not change markedly from night to day, but wind speed was higher during the day than the night. This is somewhat trivial, and if the authors re-compute the fluxes with a parameterization as function of flow, this day-night difference will be erased.

*Thank you for the comments. We remove this previous discussion since we recalculated the k600 and flux of $CO_2$ outgassing.*

*K600 was calculated as presented in the section "2.5 $CO_2$ fluxes determination" , page 5-6: "In this study, $k_{600}$ was calculated using the equation from Raymond et al. (2012) based on stream velocity (V, in m s$^{-1}$ ), slope (S, unitless), depth (D, in meters) and discharge (Q, in m$^3$ s$^{-1}$), as follow:*

$$k_{600} = 4725 \pm 445 \ x \ (V \ x \ S)^{0.86 \pm 0.016} \ x \ Q^{-0.14 \pm 0.012} \ x \ D^{0.66 \pm 0.029} \quad Eq. \ (2)$$

L298: Regarding the Ho et al. (2016) study, a note of caution is needed. This study used wind speeds from a sonic anemometer above the mangrove forest (obviously higher wind speeds that the level of the river below the canopy) and located 4 km from the coastline while the tracer injection point was located 12 km away from the coastline. For both these reasons, wind speed data used to build the Ho et al. (2016) relationship is over-estimated, meaning that the relationship itself is not reliable, and the role of currents in generating turbulence (and driving the gas transfer velocity) probably underestimated.

*Thank you for the comments. We remove this part of the discussion since we now recalculate the k600 and flux of $CO_2$ outgassing of the Red River using a different equation.*

*K600 was calculated as presented in the section "2.5 $CO_2$ fluxes determination", page 5-6:*

*"In this study, $k_{600}$ was calculated using the equation from Raymond et al. (2012) based on stream velocity (V, in m s$^{-1}$), slope (S, unitless), depth (D, in meters) and discharge (Q, in m$^3$ s$^{-1}$), as follow:*

*$k_{600} = 4725 \pm 445$ x $(V$ x $S)^{0.86 \pm 0.016}$ x $Q^{-0.14 \pm 0.012}$ x $D^{0.66 \pm 0.029}$   Eq. (2)*

L299-312: In this discussion there's a mix of studies in lakes and mangroves, which makes little sense when discussing CO2 dynamics in rivers.

*Thank you for the suggestion. We revised the paragraph and removed all reference studies unnecessary in lakes and mangroves throughout the revised ms.*

L302: Roulet et al (1997) report on beaver ponds. This is a very specific environment that is not very relevant for comparison with Red river.

*The reference Roulet et al (1997) was removed as suggested.*

L321: references are needed to back this statement.

*The reference (Richey et al 2002) was added in the revised ms.*

L323: or (in addition) inputs of CO2 from wetlands.

*It was added as suggested*

L335: Chanda study deals with mangroves and Takahashi with oceans. This is irrelevant (and out of scope) to discuss CO2 dynamics in rivers.

*The references were removed.*

L338: Comparison with Southern Ocean is irrelevant (and out of scope).

*The sentence was removed*

L346: Why did you not include freshwater discharge in the PCA and the correlation matrix analysis ?

*The PCA and the correlation matrix took into account river discharge.*

L359: Statistical test needed.

*Statistical test was added throughout the revised ms.*

L397: The cited references deal with Artic Ocean and sea-ice, not with Pearl River.

*This was removed from the revised ms.*

L401: Statistical test needed.

*Statistical test was added throughout the revised ms.*

L407-408: The city of Ha Noi does seem to influence DOC and POC but there seems to be little effect on O2 and pCO2. This is intriguing.

*We revised the discussion in page 12, in the section*
*"4.2.5 Influence of population density on $pCO_2$ and $CO_2$ emission*
*Previous studies demonstrated very high value of $pCO_2$ in river estuaries as a result of different human activities. For instance, $pCO_2$ up to ~25,000 ppm was measured in the Rhine estuary (Kempe, 1982) or*

*up to ~15,200 ppm in the Scheldt estuaries due to high discharge of pollutants (Borges and Frankignoulle, 2002).*

*Concerning the Red River, from the upstream to the downstream part of the main axe, $pCO_2$ together with $CO_2$ outgassing flux slightly increased from Yen Bai (993 ± 14 ppm and 364.9 ± 10.3 mmol $m^{-2}$ $đ^{-1}$ respectively) to Hanoi (1,275 ± 17 ppm and 304 ± 7.3 mmol $m^{-2}$ $đ^{-1}$), whatever the season. However, it is worth to note that the Hanoi station was located within the city itself and at this station, the river has not yet received the wastewater discharge of the whole city. Consequently, the Hanoi station in this study may not reflect the influence of whole city, with probably lower $O_2$ and higher $pCO_2$ levels as observed for other urban rivers in the Red River Delta (Trinh et al., 2007; 2009; 2012). "*

L410: some of the cited "rivers" are in fact mangrove brackish systems (irrelevant and out of scope).

*We checked the cited "rivers" in this paragraph*

L426-429: the authors provide no evidence to back up this hypothesis that is mainly speculation.

*The sentence was removed from the revised manuscript.*

L427: Elsewhere in the discussion the authors say that the Hoa Binh site has high pCO2 because downstream of a dam, and here the authors say the low pCO2 values in the Red River is also due dams. This cannot work both ways. Should the authors want to explore why pCO2 values are lower in the Red River than other rivers worldwide comparing levels of DOC and POC with other rivers might be useful. Also, some studies have provided hypothesis to explain large-scale variations of pCO2 across river catchments such as productivity, population density, temperature, . . . e.g. the Lauerwald study that is cited by the authors.

*The sentence was removed from the revised manuscript. The discussion about different factors that impacted on $pCO_2$ and $fCO_2$ of the Red River was re-written and re-organized through the revised manuscript.*

L434-451: This section provides a summary of the paper (and duplicates the abstract) and does not provide a real conclusion.

*The conclusion was revised*

L565: typo in "Costs"

*The reference was removed from the revised manuscript*

L598: Nathalie is the first name not the family name.

*The first name and family name of the authors in this reference were corrected*

L616: Patricia is the first name not the family name.

*The reference was removed from the revised manuscript.*

L618 and L650: This reference appears twice.

*The reference was removed from the revised manuscript.*

Table 3: There's no need to provide the air-water CO2 fluxes derived from the pCO2 calculated from pH and TA. Since calculated pCO2 is under-estimated compared to directly measured pCO2, this is also the case for the fluxes, in a very predictable and trivial way.

*The flux of $CO_2$ outgassing derived from the $pCO_2$ calculated from pH and TA was removed from the table 3.*

In tables and figures is given in the turbidity of the probe in NTU, when in fact authors measured TSM. Why not show also (or instead) the TSM data ?

Also, it might be useful to show the %POC data instead (or in addition to) POC, since POC follows closely TSM, but %POC gives some info on the origin of POC.

*The data on TSM and % POC were added in the table SM1*

**J. Crawford**

jtcrawford@usgs.gov

This paper documents the chemical conditions and concentrations of dissolved carbon dioxide in the Red River system of Vietnam. The data contribute to the "database" of concentration values for the globe, with one goal of further constraining the CO2 source strength of inland waters. Therefore, the data are valuable on their own, especially given that they fill in geographic gaps for SE Asia. The main criticism of this paper is the use of a wind-driven gas exchange model. While criticism of gas exchange models are prevalent within the community of researchers, this example is especially problematic as it relies on unlikely drivers of turbulence (and thus gas exchange) in riverine systems. There is some evidence that gas exchange is enhanced by some wind patterns in very large rivers, however, gas exchange in rivers is not considered to be a major driver. Rather, it is turbulence generated by water flow that drives gas exchange rates in these systems. Therefore, the CO2 emission estimates are not only biased, as recognized by the authors, but are likely to be highly inaccurate due to the model selected. It is hard to believe the results without some other line of confirmation. In addition to the criticism of the estimates of gas exchange, I did not find the discussion points to be well supported by the data especially given the limited time and geographic scope of the measurements. There is simply not enough evidence to support any of the inferred drivers of CO2 variability in this river system.

*Thank you for the comments. We revised the manuscript in taking into account all comments from reviewers.*

Specific Comments:

48: what references support plate tectonics as major drivers of carbon fluxes in this system?

*We added the information in page 3*
*"The Delta is located in a very flat and low land, with an elevation ranging from 0.4 to 12 m above sea level (Nguyen Ngoc Sinh et al., 1995). Previous studies showed the difference of lithology in the three upstream tributaries: Paleozoic sedimentary rocks (55.5%), Mesozoic silicic rocks (18.0%) and Mesozoic carbonated rocks (16.7%) dominate in the Thao basin, whereas Paleozoic sedimentary rocks (85.3%) and Mesozoic carbonated rocks (14.7%) cover the Da river basin, and the Lo is composed of Mesozoic silicic rocks (21.5%) and Paleozoic sedimentary rocks (72.7%). The delta area is totally covered by alluvial deposits (100%) (Moon et al., 2007; Le et al., 2007). "*

53: are changes in sediments the hypothesized drivers of changing carbon fluxes in this study?

*Yes. The information concerning changing of suspended solids of the Red River was added in page 12, in the section "Influence of dams on pCO2 and CO2 emission*

*" Noted that the Hoa Binh site is situated downstream a series of reservoirs, which have been constructed in both Chinese and Vietnamese parts including two large dams Hoa Binh (in 1989) and Son La (in 2010). The Vu Quang site is located in the downstreamof a series of reservoirs, including two important Thac Ba (in 1970) and Tuyen Quang (in 2010). Previous studies emphasized that these dams have impacted water and sediment discharges downstream (Ha and Vu 2012; Ngo et al. 2014; Lu et al. 2015) with significant sediment deposition being observed in the reservoirs (Dang et al. 2010; Vinh et al. 2014; Lu et al. 2015).*

184: if exchange is less related to wind, then what is the justification for using this model in the present study?

*Thank you for the suggestion. But we now revise the ms by recalculating k600 proposed by Raymond et al 2012*

*$K600$ was calculated as presented in the section "2.5 $CO_2$ fluxes determination" , page 5-6:*

*"In this study, $k_{600}$ was calculated using the equation from Raymond et al. (2012) based on stream velocity (V, in m s$^{-1}$ ), slope (S, unitless), depth (D, in meters) and discharge (Q, in m$^3$ s$^{-1}$), as follow:*

*$k_{600} = 4725 \pm 445 \times (V \times S)^{0.86 \pm 0.016} \times Q^{-0.14 \pm 0.012} \times D^{0.66 \pm 0.029}$   Eq. (2)*

214: such low temperature variability leads to skepticism of this environmental paramenter being a significant driver of CO2 variability. In addition, the broad conclusion here is that water chemistry seems to be quite stable over time.

*Thank you for the suggestion. The test by ANOVA and t-test results showed that no clear day-night variation but clear seasonal (dry-wet) variation of temperature was found at 5 sites. Other variables including $pCO_2$, organic matters ... showed seasonal variation.*

*So, we revised the section of temporal variation of pCO2 and CO2 flux in page 9 -10*

273: a lack of CO2 diel variability, but a finding of diel exchange variability, is a direct function of the model. This diel variability in fluxes then, is simply due to changes in wind which I do not believe are likely drivers of gas exchange in most river systems.

*Again, we recalculated the k600 by Raymond et al (2012), that lead to the change in CO2 flux. We re-write the discussion.*

276: this section reads more like discussion than results

*We removed this section in the revised ms.*

346: in contrast, this opening paragraph of the discussion most likely belongs in the results section of the manuscript

*Thank you for the suggestion. In the revised ms, this paragraph is in the section Results/ 3.4. Relations between pCO2 and water chemistry variables in page 8 -9.*

359: what part of the study design allows for a significant investigation on the role of dams and gas exchange?

*We added the paragraph for describing the 5 sites observed in page 3*
*"Five stations were studied along the lower Red River (Vietnam): Yen Bai station (at the outlet of the Thao River); Hoa Binh station (after Son La and Hoa Binh reservoirs, at the outlet of the Da River); Vu Quang (at the outlet of the Lo River); Hanoi and Ba Lat stations (in the main course of the Red River downstream). The three stations Yen Bai, Vu Quang and Hoa Binh are representative for water quality of the three main tributaries (Thao, Da and Lo) of the upstream Red River, whereas the Hanoi station is representative for the main course Red River after confluence of three main tributaries. Only the Ba Lat station, which is located at the Red River mouth (about 13 km from the sea) is influenced by seawater intrusion (Fig 1). A more detailed description of the river characteristics of the Thao, Da, Lo and the main branch of the Red River can be found in (Le et al., 2007)"*

401: paragraph is too speculative

*The paragraph was revised in page 13 (line 451)*

449: but the temperature variation was very small. How much could this have possibly contributed to the variation in CO2 exchange?

*Thank you for the suggestion. As we mention above, we recalculate the k600 and fCO2. So the results now were represented and synthetized.*

*The discussion concerning day-night variations was rewritten. $pCO_2$ differences between night and day were really low, most probably because of low temperature difference and low photosynthetic activity due to the turbidity of the Red River.*

*The conclusion was also revised.*